# Gradient descent GAN optimization is locally stable

**Vaishnavh Nagarajan**
Computer Science Department
Carnegie-Mellon University
Pittsburgh, PA 15213
vaishnavh@cs.cmu.edu

**J. Zico Kolter**
Computer Science Department
Carnegie-Mellon University
Pittsburgh, PA 15213
zkolter@cs.cmu.edu

## Abstract

Despite the growing prominence of generative adversarial networks (GANs), optimization in GANs is still a poorly understood topic. In this paper, we analyze the "gradient descent" form of GAN optimization, i.e., the natural setting where we simultaneously take small gradient steps in both generator and discriminator parameters. We show that even though GAN optimization does *not* correspond to a convex-concave game (even for simple parameterizations), under proper conditions, equilibrium points of this optimization procedure are still *locally asymptotically stable* for the traditional GAN formulation. On the other hand, we show that the recently proposed Wasserstein GAN can have non-convergent limit cycles near equilibrium. Motivated by this stability analysis, we propose an additional regularization term for gradient descent GAN updates, which *is* able to guarantee local stability for both the WGAN and the traditional GAN, and also shows practical promise in speeding up convergence and addressing mode collapse.

## 1 Introduction

Since their introduction a few years ago, Generative Adversarial Networks (GANs) [Goodfellow et al., 2014] have gained prominence as one of the most widely used methods for training deep generative models. GANs have been successfully deployed for tasks such as photo super-resolution, object generation, video prediction, language modeling, vocal synthesis, and semi-supervised learning, amongst many others [Ledig et al., 2017, Wu et al., 2016, Mathieu et al., 2016, Nguyen et al., 2017, Denton et al., 2015, Im et al., 2016].

At the core of the GAN methodology is the idea of jointly training two networks: a generator network, meant to produce samples from some distribution (that ideally will mimic examples from the data distribution), and a discriminator network, which attempts to differentiate between samples from the data distribution and the ones produced by the generator. This problem is typically written as a min-max optimization problem of the following form:

$$\min_{G} \max_{D} \quad \left( \mathbb{E}_{x \sim p_{\text{data}}}[\log D(x)] + \mathbb{E}_{z \sim p_{\text{latent}}}[\log(1 - D(G(z)))] \right). \tag{1}$$

For the purposes of this paper, we will shortly consider a more general form of the optimization problem, which also includes the recent Wasserstein GAN (WGAN) [Arjovsky et al., 2017] formulation.

Despite their prominence, the actual task of optimizing GANs remains a challenging problem, both from a theoretical and a practical standpoint. Although the original GAN paper included some analysis on the convergence properties of the approach [Goodfellow et al., 2014], it assumed that updates occurred in pure function space, allowed arbitrarily powerful generator and discriminator networks, and modeled the resulting optimization objective as a convex-concave game, therefore yielding well-defined global convergence properties. Furthermore, this analysis assumed that the discriminator network is fully optimized between generator updates, an assumption that does not mirror the practice of GAN optimization. Indeed, in practice, there exist a number of well-documented failure modes for GANs such as mode collapse or vanishing gradient problems.

**Our contributions.** In this paper, we consider the "gradient descent" formulation of GAN optimization, the setting where both the generator and the discriminator are updated simultaneously via simple (stochastic) gradient updates; that is, there are no inner and outer optimization loops, and neither the generator nor the discriminator are assumed to be optimized to convergence. Despite the fact that, as we show, this does *not* correspond to a convex-concave optimization problem (even for simple linear generator and discriminator representations), we show that:

> Under suitable conditions on the representational powers of the discriminator and the generator, the resulting GAN dynamical system *is* locally exponentially stable.

That is, for some region around an equilibrium point of the updates, the gradient updates will converge to this equilibrium point at an exponential rate. Interestingly, our conditions can be satisfied by the traditional GAN but *not* by the WGAN, and we indeed show that WGANs can have non-convergent limit cycles in the gradient descent case.

Our theoretical analysis also suggests a natural method for regularizing GAN updates by adding an additional regularization term on the norm of the discriminator gradient. We show that the addition of this term leads to locally exponentially stable equilibria for all classes of GANs, including WGANs. The additional penalty is highly related to (but also notably different from) recent proposals for practical GAN optimization, such as the unrolled GAN [Metz et al., 2017] and the improved Wasserstein GAN training [Gulrajani et al., 2017]. In practice, the approach is simple to implement, and preliminary experiments show that it helps avert mode collapse and leads to faster convergence.

## 2   Background and related work

**GAN optimization and theory.**   Although the theoretical analysis of GANs has been far outpaced by their practical application, there have been some notable results in recent years, in addition to the aforementioned work in the original GAN paper. For the most part, this work is entirely complementary to our own, and studies a very different set of questions. Arjovsky and Bottou [2017] provide important insights into *instability* that arises when the supports of the generated distribution and the true distribution are disjoint. In contrast, in this paper we delve into an equally important question of whether the updates are stable even *when* the generator is in fact very close to the true distribution (and we answer in the affirmative). Arora et al. [2017], on the other hand, explore questions relating to the sample complexity and expressivity of the GAN architecture, and their relation to the existence of an equilibrium point. However, it is still unknown as to whether, given that an equilibrium exists, the GAN update procedure will converge locally.

From a more practical standpoint, there have been a number of papers that address the topic of optimization in GANs. Several methods have been proposed that introduce new objectives or architectures for improving the (practical and theoretical) stability of GAN optimization [Arjovsky et al., 2017, Poole et al., 2016]. A wide variety of optimization heuristics and architectures have also been proposed to address challenges such as mode collapse [Salimans et al., 2016, Metz et al., 2017, Che et al., 2017, Radford et al., 2016]. Our own proposed regularization term falls under this same category, and hopefully provides some context for understanding some of these methods. Specifically, our regularization term (motivated by stability analysis) captures a degree of "foresight" of the generator in the optimization procedure, similar to the unrolled GANs procedure [Metz et al., 2017]. Indeed, we show that our gradient penalty is closely related to 1-unrolled GANs, but also provides more flexibility in leveraging this foresight. Finally, gradient-based regularization has been explored for GANs, with one of the most recent works being that of Gulrajani et al. [2017], though their penalty is on the discriminator rather than the generator as in our case.

Finally, there are several works that have concurrently addressed similar issues as this paper. Of particular similarity to the methodology we propose here are the works by Roth et al. [2017] and Mescheder et al. [2017]. The first of these two presents a stabilizing regularizer that is based on a gradient norm, where the gradient is calculated with respect to the datapoints. Our regularizer on the other hand is based on the norm of a gradient calculated with respect to the parameters. Our approach has some strong similarities with that of the second work noted above; however, the authors there do not establish or disprove stability, and instead note the presence of zero eigenvalues (which we will treat in some depth) as a motivation for their alternative optimization method. Thus, we feel the works as a whole are quite complementary, and signify the growing interest in GAN optimization issues.

**Stochastic approximation algorithms and analysis of nonlinear systems.** The technical tools we use to analyze the GAN optimization dynamics in this paper come from the fields of stochastic approximation algorithms and the analysis of nonlinear differential equations – notably the "ODE method" for analyzing convergence properties of dynamical systems [Borkar and Meyn, 2000, Kushner and Yin, 2003]. Consider a general stochastic process driven by the updates $\boldsymbol{\theta}_{t+1} = \boldsymbol{\theta}_t + \alpha_t(h(\boldsymbol{\theta}_t) + \epsilon_t)$ for vector $\boldsymbol{\theta}_t \in \mathbb{R}^n$, step size $\alpha_t > 0$, function $h : \mathbb{R}^n \to \mathbb{R}^n$ and a martingale difference sequence $\epsilon_t$.[1] Under fairly general conditions, namely: 1) bounded second moments of $\epsilon_t$, 2) Lipschitz continuity of $h$, and 3) summable but not square-summable step sizes, the stochastic approximation algorithm converges to an equilibrium point of the (deterministic) ordinary differential equation $\dot{\boldsymbol{\theta}}(t) = h(\boldsymbol{\theta}(t))$.

Thus, to understand stability of the stochastic approximation algorithm, it suffices to understand the stability and convergence of the deterministic differential equation. Though such analysis is typically used to show global asymptotic convergence of the stochastic approximation algorithm to an equilibrium point (assuming the related ODE also is globally asymptotically stable), it can also be used to analyze the *local* asymptotic stability properties of the stochastic approximation algorithm around equilibrium points.[2] This is the technique we follow throughout this entire work, though for brevity we will focus entirely on the analysis of the continuous time ordinary differential equation, and appeal to these standard results to imply similar properties regarding the discrete updates.

Given the above consideration, our focus will be on proving stability of the dynamical system around equilbrium points, i.e. points $\boldsymbol{\theta}^\star$ for which $h(\boldsymbol{\theta}^\star) = 0$.[3] Specifically, we appeal to the well known *linearization theorem* [Khalil, 1996, Sec 4.3], which states that if the Jacobian of the dynamical system $\mathbf{J} = \partial h(\theta)/\partial \theta|_{\theta=\theta^\star}$ evaluated at an equilibrium point is Hurwitz (has all strictly negative eigenvalues, $\mathrm{Re}(\lambda_i(\mathbf{J})) < 0$, $\forall i = 1, \ldots, n$), then the ODE will converge to $\theta^\star$ for some non-empty region around $\theta^\star$, at an exponential rate. This means that the system is locally asymptotically stable, or more precisely, locally exponentially stable (see Definition A.1 in Appendix A).

Thus, an important contribution of this paper is a proof of this seemingly simple fact: under some conditions, *the Jacobian of the dynamical system given by the GAN update is a Hurwitz matrix at an equilibrium* (or, if there are zero-eigenvalues, if they correspond to a subspace of equilibria, the system is still asymptotically stable). While this is a trivial property to show for convex-concave games, the fact that the GAN is *not* convex-concave leads to a substantially more challenging analysis.

In addition to this, we provide an analysis that is based on Lyapunov's stability theorem (described in Appendix A). The crux of the idea is that to prove convergence it is sufficient to identify a non-negative "energy" function for the linearized system which always decreases with time (specifically, the energy function will be a distance from the equilibrium, or from the subspace of equilibria). Most importantly, this analysis provides insights into the dynamics that lead to GAN convergence.

## 3  GAN optimization dynamics

This section comprises the main results of this paper, showing that under proper conditions the gradient descent updates for GANs (that is, updating both the generator and discriminator locally and simultaneously) is locally exponentially stable around "good" equilibrium points (where "good" will be defined shortly). This requires that the GAN loss be strictly concave, which is not the case for WGANs, and we indeed show that the updates for WGANs can cycle indefinitely. This leads us to propose a simple regularization term that *is* able to guarantee exponential stability for *any* concave GAN loss, including the WGAN, rather than requiring strict concavity.

## 3.1 The generalized GAN setting

For the remainder of the paper, we consider a slightly more general formulation of the GAN optimization problem than the one presented earlier, given by the following min/max problem:

$$\min_{G} \max_{D} \quad V(G, D) = (\mathbb{E}_{x \sim p_{\text{data}}}[f(D(x))] + \mathbb{E}_{z \sim p_{\text{latent}}}[f(-D(G(z)))]) \tag{2}$$

where $G : \mathcal{Z} \to \mathcal{X}$ is the generator network, which maps from the latent space $\mathcal{Z}$ to the input space $\mathcal{X}$; $D : \mathcal{X} \to \mathbb{R}$ is the discriminator network, which maps from the input space to a classification of the example as real or synthetic; and $f : \mathbb{R} \to \mathbb{R}$ is a concave function. We can recover the traditional GAN formulation [Goodfellow et al., 2014] by taking $f$ to be the (negated) logistic loss $f(x) = -\log(1 + \exp(-x))$; note that this convention slightly differs from the standard formulation in that in this case the discriminator outputs the real-valued "logits" and the loss function would implicitly scale this to a probability. We can recover the Wasserstein GAN by simply taking $f(x) = x$.

Assuming the generator and discriminator networks to be parameterized by some set of parameters, $\boldsymbol{\theta}_D$ and $\boldsymbol{\theta}_G$ respectively, we analyze the simple stochastic gradient descent approach to solving this optimization problem. That is, we take simultaneous gradient steps in both $\boldsymbol{\theta_D}$ and $\boldsymbol{\theta_G}$, which in our "ODE method" analysis leads to the following differential equation:

$$\dot{\boldsymbol{\theta}}_{\mathbf{D}} = \nabla_{\boldsymbol{\theta_D}} V(\boldsymbol{\theta_G}, \boldsymbol{\theta_D}), \quad \dot{\boldsymbol{\theta}}_{\mathbf{G}} := \nabla_{\boldsymbol{\theta_G}} V(\boldsymbol{\theta_G}, \boldsymbol{\theta_D}). \tag{3}$$

**A note on alternative updates.** Rather than updating both the generator and discriminator according to the min-max problem above, Goodfellow et al. [2014] also proposed a modified update for just the generator that minimizes a different objective, $V'(G, D) = -\mathbb{E}_{z \sim p_{\text{latent}}}[f(D(G(z)))]$ (the negative sign is pulled out from inside $f$). In fact, all the analyses we consider in this paper apply equally to this case (or any convex combination of both updates), as the ODE of the update equations have the same Jacobians at equilibrium.

## 3.2 Why is proving stability hard for GANs?

Before presenting our main results, we first highlight why understanding the local stability of GANs is non-trivial, even when the generator and discriminator have simple forms. As stated above, GAN optimization consists of a min-max game, and gradient descent algorithms will converge if the game is convex-concave – the objective must be convex in the term being minimized and concave in the term being maximized. Indeed, this was a crucial assumption in the convergence proof in the original GAN paper. However, for virtually any parameterization of the real GAN generator and discriminator, even if both representations are *linear*, the GAN objective will not be a convex-concave game:

**Proposition 3.1.** *The GAN objective in Equation 2 can be a concave-concave objective, i.e., concave with respect to both the discriminator and generator parameters, for a large part of the discriminator space, including regions arbitrarily close to the equilibrium.*

To see why, consider a simple GAN over 1-dimensional data and latent space with linear generator and discriminator, i.e. $D(x) = \theta_D x + \theta_D'$ and $G(z) = \theta_G z + \theta_G'$. Then the GAN objective is:

$$V(G, D) = \mathbb{E}_{x \sim p_{\text{data}}}[f(\theta_D x + \theta_D')] + \mathbb{E}_{z \sim p_{\text{latent}}}[f(-\theta_D(\theta_G z + \theta_G') - \theta_D')].$$

Because $f$ is concave, by inspection we can see that $V$ is concave in $\theta_D$ and $\theta_D'$; but it is *also* concave (not convex) in $\theta_G$ and $\theta_G'$, for the same reason. Thus, the optimization involves *concave minimization*, which in general is a difficult problem. To prove that this is not a peculiarity of the above linear discriminator system, in Appendix B, we show similar observations for a more general parametrization, and also for the case where $f''(x) = 0$ (which happens in the case of WGANs).

Thus, a major question remains as to whether or not GAN optimization is stable at all (most concave maximization is not). Indeed, there are several well-known properties of GAN optimization that may make it seem as though gradient descent optimization may *not* work in theory. For instance, it is well-known that at the optimal location $p_g = p_{\text{data}}$, the optimal discriminator will output zero on all examples, which in turn means that *any* generator distribution will be optimal for this generator. This would seem to imply that the system can not be stable around such an equilibrium.

However, as we will show, gradient descent GAN optimization *is* locally asymptotically stable, even for natural parameterizations of generator-discriminator pairs (which still make up concave-concave optimization problems). Furthermore, at equilibrium, although the zero-discriminator property means that the generator is not stable "independently", the joint dynamical system of generator and discriminator *is* locally asymptotically stable around certain equilibrium points.

### 3.3 Local stability of general GAN systems

This section contains our first technical result, establishing that GANs are locally stable under proper local conditions. Although the proofs are deferred to the appendix, the elements that we do emphasize here are the conditions that we identified for local stability to hold. Indeed, because the proof rests on these conditions (some of which are fairly strong), we want to highlight them as much as possible, as they themselves also convey valuable intuition as to what is required for GAN convergence.

To formalize our conditions, we denote the support of a distribution with probability density function (p.d.f) $p$ by $\mathrm{supp}(p)$ and the p.d.f of the generator $\boldsymbol{\theta}_{\mathbf{G}}$ by $p_{\boldsymbol{\theta}_{\mathbf{G}}}$. Let $B_{\epsilon}(\cdot)$ denote the Euclidean $L_2$-ball of radius of $\epsilon$. Let $\lambda_{\max}(\cdot)$ and $\lambda_{\min}^{(+)}(\cdot)$ denote the largest and the smallest non-zero eigenvalues of a non-zero positive semidefinite matrix. Let $\mathsf{Col}(\cdot)$ and $\mathsf{Null}(\cdot)$ denote the column space and null space of a matrix respectively. Finally, we define two key matrices that will be integral to our analyses:

$$\mathbf{K}_{DD} \triangleq \mathbb{E}_{p_{\mathrm{data}}}[\nabla_{\boldsymbol{\theta}_{\mathbf{D}}} D_{\boldsymbol{\theta}_{\mathbf{D}}}(x)\nabla_{\boldsymbol{\theta}_{\mathbf{D}}}^T D_{\boldsymbol{\theta}_{\mathbf{D}}}(x)]\big|_{\boldsymbol{\theta}_{\mathbf{D}}^{\star}} , \quad \mathbf{K}_{DG} \triangleq \int_{\mathcal{X}} \nabla_{\boldsymbol{\theta}_{\mathbf{D}}} D_{\boldsymbol{\theta}_{\mathbf{D}}}(x)\nabla_{\boldsymbol{\theta}_{\mathbf{G}}}^T p_{\theta_G}(x)dx\bigg|_{(\boldsymbol{\theta}_{\mathbf{D}}^{\star},\boldsymbol{\theta}_{\mathbf{G}}^{\star})} .$$

Here, the matrices are evaluated at an equilibrium point $(\boldsymbol{\theta}_{\mathbf{D}}^{\star}, \boldsymbol{\theta}_{\mathbf{G}}^{\star})$ which we will characterize shortly. The significance of these terms is that, as we will see, $\mathbf{K}_{DD}$ is proportional to the Hessian of the GAN objective with respect to the discriminator parameters at equilibrium, and $\mathbf{K}_{DG}$ is proportional to the off-diagonal term in this Hessian, corresponding to the discriminator and generator parameters. These matrices also occur in similar positions in the Jacobian of the system at equilibrium.

We now discuss conditions under which we can guarantee exponential stability. All our conditions are imposed on both $(\boldsymbol{\theta}_{\mathbf{D}}^{\star}, \boldsymbol{\theta}_{\mathbf{G}}^{\star})$ and all equilibria in a small neighborhood around it, though we do not state this explicitly in every assumption. First, we define the "good" equilibria we care about as those that correspond to a generator which matches the true distribution and a discriminator that is identically zero on the support of this distribution. As described next, implicitly, this also assumes that the discriminator and generator representations are powerful enough to guarantee that there are no "bad" equilibria in a local neighborhood of this equilibrium.

**Assumption I.** $p_{\boldsymbol{\theta}_{\mathbf{G}}^{\star}} = p_{\mathrm{data}}$ and $D_{\boldsymbol{\theta}_{\mathbf{D}}^{\star}}(x) = 0, \forall x \in \mathrm{supp}(p_{\mathrm{data}})$.

The assumption that the generator matches the true distribution is a rather strong assumption, as it limits us to the "realizable" case, where the generator is capable of creating the underlying data distribution. Furthermore, this means the discriminator is (locally) powerful enough that for any other generator distribution it is not at equilibrium (i.e., discriminator updates are non-zero). Since we do not typically expect this to be the case, we also provide an alternative non-realizable assumption below that is also sufficient for our results, i.e., the system is still stable. In both the realizable and non-realizable cases the requirement of an all-zero discriminator remains. This implicitly requires even the generator representation be (locally) rich enough so that when the discriminator is not identically zero, the generator is not at equilibrium (i.e., generator updates are non-zero). Finally, note that these conditions do not disallow bad equilibria outside of this neighborhood, which may potentially even be unstable.

**Assumption I.** (**Non-realizable**) The discriminator is *linear* in its parameters $\boldsymbol{\theta}_{\mathbf{D}}$ and furthermore, for any equilibrium point $(\boldsymbol{\theta}_{\mathbf{D}}^{\star}, \boldsymbol{\theta}_{\mathbf{G}}^{\star})$, $D_{\boldsymbol{\theta}_{\mathbf{D}}^{\star}}(x) = 0, \forall x \in \mathrm{supp}(p_{\mathrm{data}}) \cup \mathrm{supp}(p_{\boldsymbol{\theta}_{\mathbf{G}}^{\star}})$.

This alternative assumption is largely a weakening of Assumption I, as the condition on the discriminator remains, but there is no requirement that the generator give rise to the true distribution. However, the requirement that the discriminator be linear in the parameters (*not* in its input) is an additional restriction that seems unavoidable in this case for technical reasons. Further, note that the fact that $D_{\boldsymbol{\theta}_{\mathbf{D}}^{\star}}(x) = 0$ and that the generator/discriminator are both at equilibrium, still means that although it may be that $p_{\boldsymbol{\theta}_{\mathbf{G}}^{\star}} \neq p_{\mathrm{data}}$, these distributions are (locally) indistinguishable as far as the discriminator is concerned. Indeed, this is a nice characterization of "good" equilibria that the discriminator cannot differentiate between the real and generated samples.

Our goal next is to identify strong curvature conditions that can be imposed on the objective $V$ (or a function related to the objective), though only locally at equilibrium. First, we will require that the objective is strongly concave in the discriminator parameter space at equilibrium (note that it is concave by default). However, on the other hand, we cannot require the objective to be strongly convex in the generator parameter space as we saw that the objective is not convex-concave even in the nicest scenario, even arbitrarily close to equilbrium. Instead, we identify another convex function, namely

*the magnitude of the update on the equilibrium discriminator*, i.e., $\| \nabla_{\boldsymbol{\theta}_{\mathbf{D}}} V(\boldsymbol{\theta}_D, \boldsymbol{\theta}_G)|_{\boldsymbol{\theta}_D = \boldsymbol{\theta}_D^\star} \|^2$, and require that to be strongly convex in the generator space at equilibrium. Since these strong curvature assumptions will allow only systems with a locally unique equilibrium, we will state them in a relaxed form that accommodates a local subspace of equilibria. Furthermore, we will state these assumptions in two parts, first as a condition on $f$ and second as a condition on the parameter space.

First, the condition on $f$ is straightforward, making it necessary that the loss $f$ be concave at $0$; as we will show, when this condition is not met, there need not be local asymptotic convergence.

**Assumption II.** The function $f$ satisfies $f''(0) < 0$, and $f'(0) \neq 0$.

Next, to state conditions on the parameter space while also allowing systems with multiple equilibria locally, we first define the following property for a function, say $g$, at a specific point in its domain: along any direction, either the second derivative of $g$ must be non-zero or *all* derivatives must be zero. For example, at the origin, $g(x, y) = x^2 + x^2 y^2$ is flat along $y$, and along any other direction at an angle $\alpha \neq 0$ with the $y$ axis, the second derivative is $2 \sin^2 \alpha$. For the GAN system, we will require this property, formalized in Property I, for two convex functions whose Hessians are proportional to $\mathbf{K}_{DD}$ and $\mathbf{K}_{DG}^T \mathbf{K}_{DG}$. We provide more intuition for these functions below.

**Property I.** $g : \Theta \to \mathbb{R}$ satisfies Property I at $\boldsymbol{\theta}^\star \in \Theta$ if for any $\boldsymbol{\theta} \in \mathsf{Null}(\nabla_{\boldsymbol{\theta}}^2 g(\boldsymbol{\theta})|_{\boldsymbol{\theta}^\star})$, the function is locally constant along $\boldsymbol{\theta}$ at $\boldsymbol{\theta}^\star$, i.e., $\exists \epsilon > 0$ such that for all $\epsilon' \in (-\epsilon, \epsilon)$, $g(\boldsymbol{\theta}^\star) = g(\boldsymbol{\theta}^\star + \epsilon' \boldsymbol{\theta})$.

**Assumption III.** At an equilibrium $(\boldsymbol{\theta}_{\mathbf{D}}^\star, \boldsymbol{\theta}_{\mathbf{G}}^\star)$, the functions $\mathbb{E}_{p_{\text{data}}}[D_{\boldsymbol{\theta}_{\mathbf{D}}}^2(x)]$ and $\left\| \mathbb{E}_{p_{\text{data}}}[\nabla_{\boldsymbol{\theta}_{\mathbf{D}}} D_{\boldsymbol{\theta}_{\mathbf{D}}}(x)] - \mathbb{E}_{p_{\boldsymbol{\theta}_{\mathbf{G}}}}[\nabla_{\boldsymbol{\theta}_{\mathbf{D}}} D_{\boldsymbol{\theta}_{\mathbf{D}}}(x)] \right\|^2 \Big|_{\boldsymbol{\theta}_{\mathbf{D}} = \boldsymbol{\theta}_{\mathbf{D}}^\star}$ must satisfy Property I in the discriminator and generator space respectively.

Here is an intuitive explanation of what these two non-negative functions represent and how they relate to the objective. The first function is a function of $\boldsymbol{\theta}_{\mathbf{D}}$ which measures how far $\boldsymbol{\theta}_{\mathbf{D}}$ is from an all-zero state, and the second is a function of $\boldsymbol{\theta}_{\mathbf{G}}$ which measures how far $\boldsymbol{\theta}_{\mathbf{G}}$ is from the true distribution; at equilibrium these functions are zero. We will see later that given $f''(0) < 0$, the curvature of the first function at $\boldsymbol{\theta}_{\mathbf{D}}^\star$ is representative of the curvature of $V(\boldsymbol{\theta}_{\mathbf{D}}, \boldsymbol{\theta}_{\mathbf{G}}^\star)$ in the discriminator space; similarly, given $f'(0) \neq 0$ the curvature of the second function at $\boldsymbol{\theta}_{\mathbf{G}}^\star$ is representative of the curvature of *the magnitude of the discriminator update on $\boldsymbol{\theta}_D^\star$* in the generator space. The intuition behind why this particular relation holds is that, when $\boldsymbol{\theta}_{\mathbf{G}}$ moves away from the true distribution, while the second function in Assumption III increases, $\boldsymbol{\theta}_{\mathbf{D}}^\star$ also becomes more suboptimal for that generator; as a result, the magnitude of update on $\boldsymbol{\theta}_{\mathbf{D}}^\star$ increases too. Note that we show in Lemma C.2 that the Hessian of the two functions in Assumption III in the discriminator and the generator space respectively are proportional to $\mathbf{K}_{DD}$ and $\mathbf{K}_{DG}^T \mathbf{K}_{DG}$.

The above relations involving the two functions and the GAN objective, together with Assumption III, basically allow us to consider systems with reasonable strong curvature properties, while also allowing many equilibria in a local neighborhood in a specific sense. In particular, if the curvature of the first function is flat along a direction $\mathbf{u}$ (which also means that $\mathbf{K}_{DD}\mathbf{u} = 0$) we can perturb $\boldsymbol{\theta}_{\mathbf{D}}^\star$ slightly along $\mathbf{u}$ and still have an 'equilibrium discriminator' as defined in Assumption I, i.e., $\forall x \in \text{supp}(p_{\boldsymbol{\theta}_{\mathbf{G}}^\star}), D_{\boldsymbol{\theta}_{\mathbf{D}}}(x) = 0$. Similarly, for any direction $\mathbf{v}$ along which the curvature of the second function is flat (i.e., $\mathbf{K}_{DG}\mathbf{v} = 0$), we can perturb $\boldsymbol{\theta}_{\mathbf{G}}^\star$ slightly along that direction such that $\boldsymbol{\theta}_{\mathbf{G}}$ remains an 'equilibrium generator' as defined in Assumption I, i.e., $p_{\theta_G} = p_{\text{data}}$. We prove this formally in Lemma C.2. Perturbations along any other directions do not yield equilibria because then, either $\boldsymbol{\theta}_D$ is no longer in an all-zero state or $\boldsymbol{\theta}_G$ does not match the true distribution. Thus, we consider a setup where the rank deficiencies of $\mathbf{K}_{DD}$ and $\mathbf{K}_{DG}^T \mathbf{K}_{DG}$ if any, correspond to equivalent equilibria (which typically exist for neural networks, though in practice they may not correspond to 'linear' perturbations as modeled here).

Our final assumption is on the supports of the true and generated distributions: we require that all the generators in a sufficiently small neighborhood of the equilibrium have distributions with the same support as the true distribution. Following this, we briefly discuss a relaxation of this assumption.

**Assumption IV.** $\exists \epsilon_G > 0$ such that $\forall \boldsymbol{\theta}_{\mathbf{G}} \in B_{\epsilon_G}(\boldsymbol{\theta}_{\mathbf{G}}^\star), \text{supp}(p_{\boldsymbol{\theta}_{\mathbf{G}}}) = \text{supp}(p_{\text{data}})$.

This may typically hold if the support covers the whole space $\mathcal{X}$; but when the true distribution has support in some smaller disjoint parts of the space $\mathcal{X}$, nearby generators may correspond to slightly

displaced versions of this distribution with a different support. For the latter scenario, we show in Appendix C.1 that local exponential stability holds under a certain smoothness condition on the discriminator. Specifically, we require that $D_{\theta_D^\star}(\cdot)$ be zero not only on the support of $\theta_G^\star$ but also on the support of small perturbations of $\theta_G^\star$ as otherwise the generator will not be at equilibrium. (Additionally, we also require this property from the discriminators that lie within a small perturbation of $\theta_D^\star$ in the null space of $\mathbf{K}_{DD}$ so that they correspond to equilibrium discriminators.) We note that while this relaxed assumption accounts for a larger class of examples, it is still strong in that it also restricts us from certain simple systems. Due to space constraints, we state and discuss the implications of this assumption in greater detail in Appendix C.1.

We now state our main result.

**Theorem 3.1.** *The dynamical system defined by the GAN objective in Equation 2 and the updates in Equation 3 is locally exponentially stable with respect to an equilibrium point $(\theta_D^\star, \theta_G^\star)$ when the Assumptions I, II, III, IV hold for $(\theta_D^\star, \theta_G^\star)$ and other equilibria in a small neighborhood around it. Furthermore, the rate of convergence is governed only by the eigenvalues $\lambda$ of the Jacobian $\mathbf{J}$ of the system at equilibrium with a strict negative real part upper bounded as:*

- *If $\mathrm{Im}(\lambda) = 0$, then $\mathrm{Re}(\lambda) \leq \frac{2f''(0)f'^2(0)\lambda_{\min}^{(+)}(\mathbf{K}_{DD})\lambda_{\min}^{(+)}(\mathbf{K}_{DG}^T\mathbf{K}_{DG})}{4f''^2(0)\lambda_{\min}^{(+)}(\mathbf{K}_{DD})\lambda_{\max}(\mathbf{K}_{DD})+f'(0)^2\lambda_{\min}^{(+)}(\mathbf{K}_{DG}^T\mathbf{K}_{DG})}$*

- *If $\mathrm{Im}(\lambda) \neq 0$, then $\mathrm{Re}(\lambda) \leq f''(0)\lambda_{\min}^{(+)}(\mathbf{K}_{DD})$*

The vast majority of our proofs are deferred to the appendix, but we briefly describe the intuition here. It is straightforward to show that the Jacobian $\mathbf{J}$ of the system at equilibrium can be written as:

$$\mathbf{J} = \begin{bmatrix} \mathbf{J}_{DD} & \mathbf{J}_{DG} \\ -\mathbf{J}_{DG}^T & \mathbf{J}_{GG} \end{bmatrix} = \begin{bmatrix} 2f''(0)\mathbf{K}_{DD} & f'(0)\mathbf{K}_{DG} \\ -f'(0)\mathbf{K}_{DG}^T & 0 \end{bmatrix}.$$

Recall that we wish to show this is Hurwitz. First note that $\mathbf{J}_{DD}$ (the Hessian of the objective with respect to the discriminator) is negative semi-definite if and only if $f''(0) < 0$. Next, a crucial observation is that $\mathbf{J}_{GG} = 0$ i.e, the Hessian term w.r.t. the generator vanishes because for the all-zero discriminator, all generators result in the same objective value. Fortunately, this means *at equilibrium* we do not have non-convexity in $\theta_G$ precluding local stability. Then, we make use of the crucial Lemma G.2 we prove in the appendix, showing that any matrix of the form $\begin{bmatrix} -\mathbf{Q} & \mathbf{P}; & -\mathbf{P}^T & 0 \end{bmatrix}$ is Hurwitz provided that $-\mathbf{Q}$ is strictly negative definite and $\mathbf{P}$ has full column rank.

However, this property holds only when $\mathbf{K}_{DD}$ is positive definite and $\mathbf{K}_{DG}$ is full column rank. Now, if $\mathbf{K}_{DD}$ or $\mathbf{K}_{DG}$ do not have this property, recall that the rank deficiency is due to a subspace of equilibria around $(\theta_D^\star, \theta_G^\star)$. Consequently, we can analyze the stability of the system projected to an subspace orthogonal to these equilibria (Theorem A.4). Additionally, we also prove stability using Lyapunov's stability (Theorem A.1) by showing that the squared $L_2$ distance to the subspace of equilibria always either decreases or only instantaneously remains constant.

**Additional results.** In order to illustrate our assumptions in Theorem 3.1, in Appendix D we consider a simple GAN that learns a multi-dimensional Gaussian using a quadratic discriminator and a linear generator. In a similar set up, in Appendix E, we consider the case where $f(x) = x$, i.e., the Wasserstein GAN, and so $f''(x) = 0$, and we show that the system can perennially cycle around an equilibrium point without converging. A simple two-dimensional example is visualized in Section 4. Thus, *gradient descent WGAN optimization is not necessarily asymptotically stable.*

### 3.4 Stabilizing optimization via gradient-based regularization

Motivated by the considerations above, in this section we propose a regularization penalty for the generator update, which uses a term based upon the gradient of the discriminator. Crucially, the regularization term does *not* change the parameter values at the equilibrium point, and at the same time enhances the local stability of the optimization procedure, both in theory and practice. Although these update equations do require that we differentiate with respect to a function of another gradient term, such "double backprop" terms (see e.g., Drucker and Le Cun [1992]) are easily computed by modern automatic differentiation tools. Specifically, we propose the regularized update

$$\theta_{\mathbf{G}} := \theta_{\mathbf{G}} - \alpha\nabla_{\theta_{\mathbf{G}}}\left(V(D_{\theta_{\mathbf{D}}}, G_{\theta_{\mathbf{G}}}) + \eta\|\nabla_{\theta_{\mathbf{D}}}V(D_{\theta_{\mathbf{D}}}, G_{\theta_{\mathbf{G}}})\|^2\right). \tag{4}$$

**Local Stability** The intuition of this regularizer is perhaps most easily understood by considering how it changes the Jacobian at equilibrium (though there are other means of motivating the update as well, discussed further in Appendix F.2). In the Jacobian of the new update, although there are now non-antisymmetric diagonal blocks, the block diagonal terms are now negative definite:

$$\begin{bmatrix} \mathbf{J}_{DD} & \mathbf{J}_{DG} \\ -\mathbf{J}_{DG}^T(\mathbf{I} + 2\eta\mathbf{J}_{DD}) & -2\eta\mathbf{J}_{DG}^T\mathbf{J}_{DG} \end{bmatrix}.$$

As we show below in Theorem 3.2 (proved in Appendix F), as long as we choose $\eta$ small enough so that $I + 2\eta\mathbf{J}_{DD} \succeq 0$, this guarantees the updates are locally asymptotically stable for any concave $f$. In addition to stability properties, this regularization term also addresses a well known failure state in GANs called *mode collapse*, by lending more "foresight" to the generator. The way our updates provide this foresight is very similar to the unrolled updates proposed in Metz et al. [2017], although our regularization is much simpler and provides more flexibility to leverage the foresight. In practice, we see that our method can be as powerful as the more complex and slower 10-unrolled GANs. We discuss this and other intuitive ways of motivating our regularizer in Appendix F.

**Theorem 3.2.** *The dynamical system defined by the GAN objective in Equation 2 and the updates in Equation 4, is locally exponentially stable at the equilibrium, under the same conditions as in Theorem 3.1, if $\eta < \frac{1}{2\lambda_{\max}(-\mathbf{J}_{DD})}$. Further, under appropriate conditions similar to these, the WGAN system is locally exponentially stable at the equilibrium for any $\eta$. The rate of convergence for the WGAN is governed only by the eigenvalues $\lambda$ of the Jacobian at equilibrium with a strict negative real part upper bounded as:*

- *If $\mathrm{Im}(\lambda) = 0$, then $\mathrm{Re}(\lambda) \leq -\frac{2f'^2(0)\eta\lambda_{\min}^{(+)}(\mathbf{K}_{DG}^T\mathbf{K}_{DG})}{4f'^2(0)\eta^2\lambda_{\max}(\mathbf{K}_{DG}^T\mathbf{K}_{DG})+1}$*

- *If $\mathrm{Im}(\lambda) \neq 0$, then $\mathrm{Re}(\lambda) \leq -\eta f'^2(0)\lambda_{\min}^{(+)}(\mathbf{K}_{DG}^T\mathbf{K}_{DG})$.*

# 4 Experimental results

We very briefly present experimental results that demonstrate that our regularization term also has substantial practical promise.[4] In Figure 1, we compare our gradient regularization to 10-unrolled GANs on the same architecture and dataset (a mixture of eight Gaussians) as in Metz et al. [2017]. Our system quickly spreads out all the points instead of first exploring only a few modes and then redistributing its mass over all the modes gradually. Note that the conventional GAN updates are known to enter mode collapse for this setup. We see similar results (see Figure 2 here, and Figure 4 in the Appendix for a more detailed figure) in the case of a stacked MNIST dataset using a DCGAN [Radford et al., 2016], i.e., three random digits from MNIST are stacked together so as to create a distribution over 1000 modes. Finally, Figure 3 presents streamline plots for a 2D system where both the true and the latent distribution is uniform over $[-1, 1]$ and the discriminator is $D(x) = w_2x^2$ while the generator is $G(z) = az$. Observe that while the WGAN system goes in orbits as expected, the original GAN system converges. With our updates, both these systems converge quickly to the true equilibrium.

# 5 Conclusion

In this paper, we presented a theoretical analysis of the local asymptotic stability of GAN optimization under proper conditions. We further showed that the recently proposed WGAN is *not* asymptotically stable under the same conditions, but we introduced a gradient-based regularizer which stabilizes both traditional GANs and the WGANs, and can improve convergence speed in practice.

The results here provide substantial insight into the nature of GAN optimization, perhaps even offering some clues as to why these methods have worked so well *despite* not being convex-concave. However, we also emphasize that there are substantial limitations to the analysis, and directions for future work. Perhaps most notably, the analysis here only provides an understanding of what happens

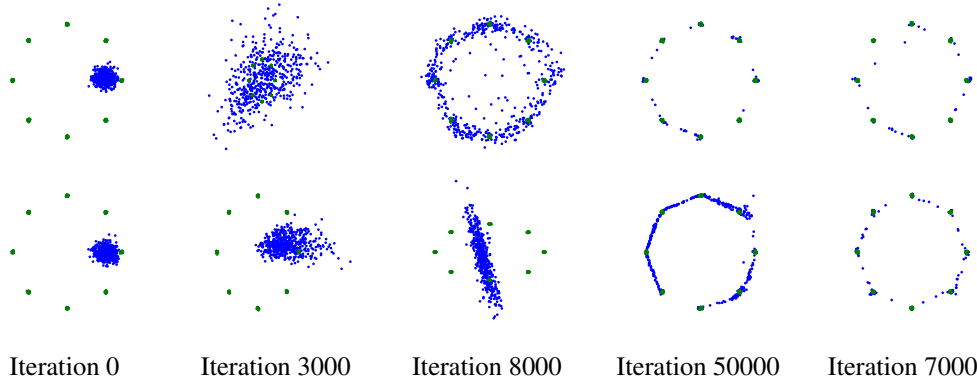

Iteration 0      Iteration 3000      Iteration 8000      Iteration 50000      Iteration 70000

Figure 1: Gradient regularized GAN, $\eta = 0.5$ (top row) vs. 10-unrolled with $\eta = 10^{-4}$ (bottom row).

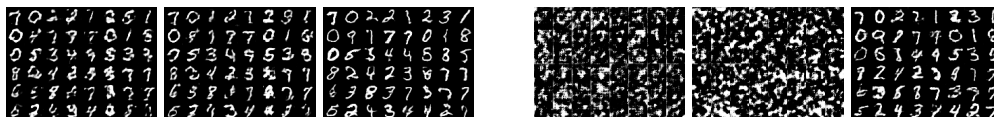

Figure 2: Gradient regularized (left) and traditional (right) DCGAN architectures on stacked MNIST examples, after 1, 4 and 20 epochs.

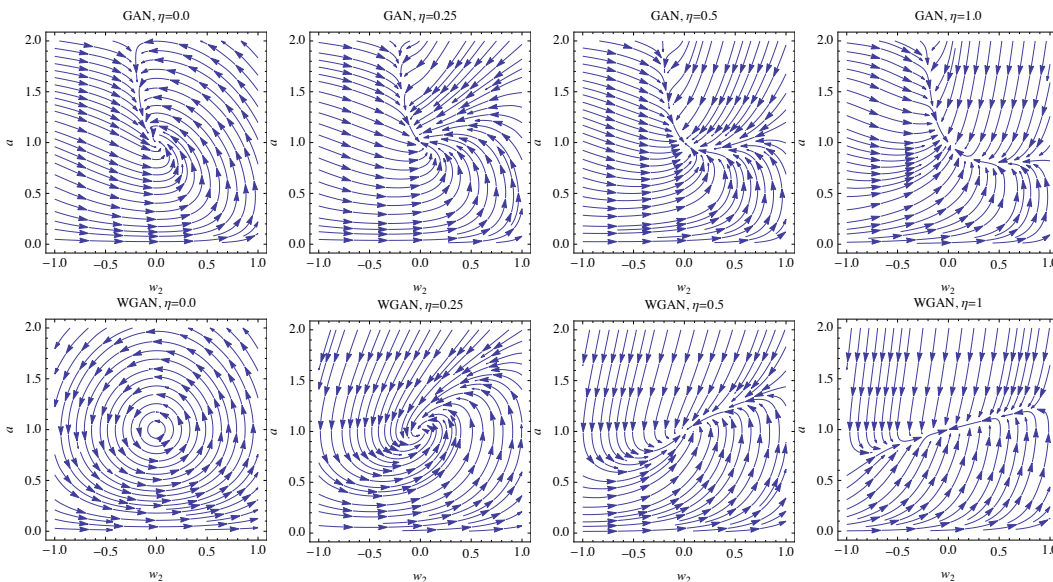

Figure 3: Streamline plots around the equilibrium $(0, 1)$ for the conventional GAN (top) and the WGAN (bottom) for $\eta = 0$ (vanilla updates) and $\eta = 0.25, 0.5, 1$ (left to right).

locally, close to an equilibrium point. For non-convex architectures this may be all that is possible, but it seems plausible that much stronger *global* convergence results could hold for simple settings like the linear quadratic GAN (indeed, as the streamline plots show, we observe this in practice for simple domains). Second, the analysis here does not show the equilibrium points necessarily exist, but only illustrates convergence if there do exist points that satisfy certain criteria: the existence question has been addressed by previous work [Arora et al., 2017], but much more analysis remains to be done here. GANs are rapidly becoming a cornerstone of deep learning methods, and the theoretical and practical understanding of these methods will prove crucial in moving the field forward.

**Acknowledgements.** We thank Lars Mescheder for pointing out a missing condition in the relaxed version of Assumption IV (see Appendix C.1) in earlier versions of this manuscript.

## Footnotes

[1] Stochastic gradient descent on an objective $f(\theta)$ can be expressed in this framework as $h(\boldsymbol{\theta}) = \nabla_{\boldsymbol{\theta}} f(\boldsymbol{\theta})$.

[2] Note that the local analysis does *not* show that the stochastic approximation algorithm will necessarily converge to an equilibrium point, but still provides a valuable characterization of how the algorithm will behave around these points.

[3] Note that this is a slightly different usage of the term equilibrium as typically used in the GAN literature, where it refers to a Nash equilibrium of the min-max optimization problem. These two definitions (assuming we mean just a local Nash equilibrium) are equivalent for the ODE corresponding to the min-max game, but we use the dynamical systems meaning throughout this paper, that is, any point where the gradient update is zero.

[4]We provide an implementation of this technique at `https://github.com/locuslab/gradient_regularized_gan`

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
