[Supplementary Material]

# Appendix

| Epoch 1 | Epoch 2 | Epoch 4 | Epoch 8 | Epoch 16 | Epoch 20 |

Figure 4: Gradient regularized GAN with $\eta = 5 \times 10^{-6}$ vs. traditional GAN

## A  Preliminaries

In this section, we present preliminaries from non-linear systems theory [Khalil, 1996]. In particular, we formally define local stability of dynamic systems, and then present an important theorem that helps us study stability of non-linear systems. Finally, we present a modification of this result that will be crucial in proving stability of GANs under our assumptions.

Consider a system consisting of variables $\boldsymbol{\theta} \in \mathbb{R}^n$ whose time derivative is defined by $h(\boldsymbol{\theta})$ i.e.,

$$\dot{\boldsymbol{\theta}} = h(\theta). \tag{5}$$

Without loss of generality let the origin be an equilibrium point of this sytem. That is, $h(\mathbf{0}) = \mathbf{0}$. Let $\boldsymbol{\theta}(t)$ denote the state of the system at some time $t$. Then, we have the following definition of local stability:

**Definition A.1 (Stability).** (Definition 4.1 from Khalil [1996]) The origin of the system in Equation 5 is

- stable if for each $\epsilon > 0$, there is $\delta = \delta(\epsilon) > 0$ such that

$$\|\boldsymbol{\theta}(0)\| < \delta \implies \|\boldsymbol{\theta}(t)\| < \epsilon, \ \forall t \geq 0.$$

- unstable if not stable.

- asymptotically stable if it is stable and $\delta > 0$ can be chosen such that

$$\|\boldsymbol{\theta}(0)\| < \delta \implies \lim_{t \to \infty} \theta(t) = 0$$

- exponentially stable if it is asymptotically stable and $\delta, k, \lambda > 0$ can be chosen such that

$$\|\boldsymbol{\theta}(0)\| < \delta \implies \|\boldsymbol{\theta}(t)\| \le k\|\boldsymbol{\theta}(0)\| \exp(-\lambda t)$$

The system is stable if for any chosen ball around the equilibrium (of radius $\epsilon$), one can initialize the system anywhere within a sufficiently small ball around the equilibrium (of radius $\delta(\epsilon)$) such that the system always stays within the $\epsilon$ ball. Note that such a system may either converge to equilibrium or orbit around equilibrium perennially within the $\epsilon$ ball. In contrast, a system is unstable if there are initializations that are arbitrarily close to the equilibrium which can escape the $\epsilon$-ball. Finally, asymptotic stability is a stronger notion of stability, which implies that there is a region around the equilibrium such that any initialization within that region will converge to the equilibrium (in the limit $t \to \infty$). For example, as we saw, GANs are always stable; however, WGANs are stable but not asymptotically stable.

**Extension to multiple equilibria.** Note that since a GAN system might have multiple arbitrarily close equilibria, or a subspace of equilibria, we will define asymptotic stability to imply convergence to any of the equilibria in the neighborhood of a considered equilibrium. That is, $\lim_{t \to \infty} \theta(t) = \boldsymbol{\theta}^\star$ where $\boldsymbol{\theta}^\star$ is either the considered equilibrium point at the origin or any other equilibrium point that is within some small neighborhood around origin.

We now present Lyapunov's stability theorem which is used to prove locally asymptotic stability of a given system. The basic idea is that a system is asymptotically stable if we can find a scalar "energy" function $V(\boldsymbol{\theta})$ (also called a Lyapunov function) that i) is *positive definite* which means, $V(\boldsymbol{\theta})$ positive everywhere and zero at the equilibrium ii) its time derivative $\dot{V}(\boldsymbol{\theta})$ is strictly negative around the equilibrium.

**Theorem A.1** (**Lyapunov function**). *(Theorem 4.1 from Khalil [1996]) Let $B_\epsilon(\mathbf{0})$ be a small region around the origin of the system in Equation 5. Let $V : B_\epsilon(\mathbf{0}) \to \mathbb{R}$ be a continuously differentiable function such that*

- *it is* positive definite *i.e.,* $V(\mathbf{0}) = 0$ *and* $V(\boldsymbol{\theta}) > 0$ *for* $\boldsymbol{\theta} \in B_\epsilon(0) - \{\mathbf{0}\}$

- $\dot{V}(\boldsymbol{\theta}) \le 0$ *for* $\boldsymbol{\theta} \in B_\epsilon(0) - \{0\}$

*Then, the origin is stable. Moreover, if*

$$\dot{V}(\boldsymbol{\theta}) < 0, \forall \boldsymbol{\theta} \in B_\epsilon(0) - \{\mathbf{0}\}$$

*then the origin is asymptotically stable.*

We next present an important tool that simplifies the study of stability of non-linear systems. The result is that one can "linearize" any non-linear system near an equilibrium and analyze the stability of the linearized system to comment on the local stability of the original system.

**Theorem A.2** (**Linearization**). *(Theorem 4.5 from Khalil [1996]) Let $\mathbf{J}$ be the Jacobian of the system in Equation 5 at its origin i.e.,*

$$\mathbf{J} = \left. \frac{\partial h(\boldsymbol{\theta})}{\partial \boldsymbol{\theta}} \right|_{\boldsymbol{\theta} = \mathbf{0}}.$$

*Then,*

- *The origin is locally exponentially stable if $\mathbf{J}$ is Hurwitz i.e., $\mathrm{Re}(\lambda) < 0$ for all eigenvalues $\lambda$ of $\mathbf{J}$.*

- *The origin is unstable if $\mathrm{Re}(\lambda) > 0$ for all eigenvalues $\lambda$ of $\mathbf{J}$.*

The key idea in the proof for this result is that the system can be written as $h(\boldsymbol{\theta}) = \mathbf{J}\boldsymbol{\theta} + g_1(\boldsymbol{\theta})$, where $g_1(\boldsymbol{\theta})$, the remainder of the linear approximation is bounded as $\|g_1(\boldsymbol{\theta})\| \le O(\|\boldsymbol{\theta}\|^2)$ sufficiently

close to equilibrium. Now, it turns out that when $\mathbf{J}$ is Hurwitz, one can find a quadratic Lyapunov function for the original system whose rate of decrease is also quadratic in $\boldsymbol{\theta}$. Since, $\|g_1(\boldsymbol{\theta})\|$ is only a quadratic remainder term, one can show that the remainder term only adds a cubic term to the change in the Lyapunov function. This is however smaller than a quadratic change near the equilibrium, and therefore the quadratic Lyapunov function for the linearized system works as a Lyapunov function for the original system too.

In all our analyses, we will linearize our system and show that the Jacobian is Hurwitz. However, it is often useful to identify the quadratic Lyapunov function for the (linearized) system. Unfortunately, for some of the Jacobians we will encounter, it is hard to come up with a quadratic Lyapunov function that always strictly decreases. Instead, we will identify a function that either strictly decreases or sometimes remains constant but only instantenously. While Lyapunov's stability theorem does not help us conclude anything about stability for this case, the following corollary of LaSalle's theorem (we do not state the theorem here) is sufficient to prove asymptotic stability in this case.

**Theorem A.3** (**Corollary of LaSalle's invariance principle**, Corollary 4.1 from Khalil [1996]). *Let $B_\epsilon(\mathbf{0})$ be a small region around an equilibrium $0$ of the system in Equation 5. Let $V : B_\epsilon(\mathbf{0}) \to \mathbb{R}$ be a continuously differentiable function such that*

- *$V(\boldsymbol{\theta}) = 0$ if and only if $\dot{\boldsymbol{\theta}} = 0$ and $V(\boldsymbol{\theta}) > 0$ for $\boldsymbol{\theta} \in B_\epsilon(0) - \{\mathbf{0}\}$ such that $\dot{\boldsymbol{\theta}} \neq 0$ .*

- *$\dot{V}(\boldsymbol{\theta}) \leq 0$ for $\boldsymbol{\theta} \in B_\epsilon(\mathbf{0}) - \{0\}$*

- *Let $S = \{\boldsymbol{\theta} \in B_\epsilon(\mathbf{0}) \,|\, \dot{V}(\boldsymbol{\theta}) = 0\}$. There is no trajectory that identically stays in $S$ except for the trajectories at equilibrium points.*

*then the system is locally asymptotically stable with respect to $\mathbf{0}$ and other equilibria in its neighborhood.*

Finally, we prove an extension of the linearization theorem that helps us deal with analyzing the stability of a special kind of non-linear systems, specifically those with multiple equilibria in a local neighborhood of a considered equilibrium. The theorem, though inuitively follows from the original linearization theorem itself, is not a standard theorem in non-linear systems, to the best of our knowledge.

Formally, we consider a case where the system consists of two sets of parameters $\boldsymbol{\theta}$ and $\boldsymbol{\gamma}$ such that from the equilibrium, any small perturbation along $\boldsymbol{\gamma}$ preserves the equilibrium. We show that it is enough to show that the Jacobian with respect to $\boldsymbol{\theta}$ is Hurwitz to prove stability.

**Theorem A.4.** *Consider a non-linear system of parameters $(\boldsymbol{\theta}, \boldsymbol{\gamma})$,*

$$\dot{\boldsymbol{\theta}} = h_1(\boldsymbol{\theta}, \boldsymbol{\gamma}), \dot{\boldsymbol{\gamma}} = h_2(\boldsymbol{\theta}, \boldsymbol{\gamma}) \tag{6}$$

*with an equilibrium point at the origin. Let there exist $\epsilon$ such that for any $\boldsymbol{\gamma} \in B_\epsilon(\mathbf{0})$, $(\mathbf{0}, \boldsymbol{\gamma})$ is an equilibrium point. Then, if*

$$\mathbf{J} = \left. \frac{\partial h_1(\boldsymbol{\theta}, \boldsymbol{\gamma})}{\partial \boldsymbol{\theta}} \right|_{(\mathbf{0}, \mathbf{0})} \tag{7}$$

*is a Hurwitz matrix, the non-linear system in Equation 6 is exponentially stable.*

*Proof.* The proof for this statement is quite similar to the proof of the original theorem for linearization. The high level idea is that if $\mathbf{J}$ is exponentially stable, then there exists a quadratic Lyapunov function that is always decreasing for the system $\dot{\boldsymbol{\theta}} = \mathbf{J}\boldsymbol{\theta}$. Then, we show that the same quadratic function works for the original non-linear system too in a small neighborhood around equilibrium for which the non-linear remainder terms are sufficiently small. In particular, we show that $\boldsymbol{\theta}$ converges to zero, and $\boldsymbol{\gamma}$ converges to a value less than $\epsilon$.

A subtle point however, is that this quadratic function would decrease only when it is within a particular neighborhood of $\boldsymbol{\theta}$ around origin, and also a particular neighborhood of $\boldsymbol{\gamma}$ around origin. However, within this neighborhood, say $\mathcal{S}$, we can only guarantee that $\boldsymbol{\theta}$ exponentially approaches the origin; $\boldsymbol{\gamma}$ might move away from the $\epsilon$-neighborhood around origin, and if it does, the system may exit $\mathcal{S}$ and the system may not even converge! We carefully overcome this, by first identifying

$\mathcal{S}$, and then identifying a smaller space within $\mathcal{S}$ where $\boldsymbol{\gamma}$ does not vary too much over the course of convergence, so that the system stays within $\mathcal{S}$ forever – until convergence.

To identify $\mathcal{S}$, let,

$$h_1(\theta, \gamma) = \mathbf{J}\boldsymbol{\theta} + g_1(\boldsymbol{\theta}, \boldsymbol{\gamma}).$$

The first crucial step is to show that for any constant $c > 0$, for a sufficiently small neighborhood around the equilibrium, we will have $\|g_1(\boldsymbol{\theta}, \boldsymbol{\gamma})\| \leq c\|\boldsymbol{\theta}\|$. To show this, consider the Taylor series expansion for the remainder $g_1(\boldsymbol{\theta}, \boldsymbol{\gamma}) = h_1(\boldsymbol{\theta}, \boldsymbol{\gamma}) - \mathbf{J}\boldsymbol{\theta}$ around equilibrium. Clearly, the expansion would not have a constant term because $h_1(\mathbf{0}, \mathbf{0}) = 0$. It would not have a linear term in $\boldsymbol{\theta}$ because that is accounted for already. Finally, it will not have any term that is purely a function of $\boldsymbol{\gamma}$, because $h_1(\mathbf{0}, \boldsymbol{\gamma}) = 0$ in a small neighborhood around equilibrium (since $(\mathbf{0}, \boldsymbol{\gamma})$ are all equilibria). Therefore, we can write:

$$g_1(\boldsymbol{\theta}, \boldsymbol{\gamma}) = \boldsymbol{\theta} g_2(\boldsymbol{\gamma}) + g_3(\boldsymbol{\theta}, \boldsymbol{\gamma})$$

where $g_2(\boldsymbol{\gamma})$ only consists of linear or higher degree terms in $\boldsymbol{\gamma}$ and $g_3(\boldsymbol{\theta}, \boldsymbol{\gamma})$ consists only of terms that are quadratic or higher degree terms in $\boldsymbol{\theta}$ (and any arbitrary degree of $\boldsymbol{\gamma}$). Therefore, we have that:

$$\lim_{\boldsymbol{\gamma} \to \mathbf{0}} g_2(\boldsymbol{\gamma}) = 0, \quad \lim_{\boldsymbol{\theta} \to \mathbf{0}} \frac{g_3(\boldsymbol{\theta}, \boldsymbol{\gamma})}{\|\boldsymbol{\theta}\|} = 0$$

Then, for an arbitrarily chosen small constant $c$, for a sufficiently close neighborhood around the equilibrium, we can say that $\|g_2(\boldsymbol{\gamma})\| \leq c/2$ and $\|g_3(\boldsymbol{\theta}, \boldsymbol{\gamma})\| \leq c\|\boldsymbol{\theta}\|/2$. Thus,

$$\|g_1(\boldsymbol{\theta}, \boldsymbol{\gamma})\| \leq \|\boldsymbol{\theta}\|\|g_2(\boldsymbol{\gamma})\| + \|g_3(\boldsymbol{\theta}, \boldsymbol{\gamma})\| \leq c\|\boldsymbol{\theta}\|$$

We will use this property soon for a cleverly chosen value of $c$. Now, by Theorem 4.6 in Khalil [1996], we have that for any positive definite symmetric matrix $\mathbf{Q}$, there exists a positive definite matrix $\mathbf{P}$ such that $\mathbf{J}^T\mathbf{P} + \mathbf{J}\mathbf{P} = -\mathbf{Q}$. Then, if we choose $V(\boldsymbol{\theta}) = \boldsymbol{\theta}^T\mathbf{P}\boldsymbol{\theta}$ as the quadratic Lyapunov function for the linearized system $\dot{\boldsymbol{\theta}} = \mathbf{J}\boldsymbol{\theta}$, the rate of its decrease is given by $\dot{V}(\boldsymbol{\theta}) = -\boldsymbol{\theta}^T\mathbf{Q}\boldsymbol{\theta}$ which is negative at all points except at $\boldsymbol{\theta} = 0$.

Now, if we use the same Lyapunov function for the whole system as $V(\boldsymbol{\theta}, \boldsymbol{\gamma}) = \boldsymbol{\theta}^T\mathbf{P}\boldsymbol{\theta}$, the rate of its decrease near the origin would be $\dot{V}(\boldsymbol{\theta}, \boldsymbol{\gamma}) = -\boldsymbol{\theta}^T\mathbf{Q}\boldsymbol{\theta} + 2\boldsymbol{\theta}^T\mathbf{P}g_1(\boldsymbol{\theta}, \boldsymbol{\gamma})$. If we choose a sufficiently small neighborhood such that for $c = \frac{1}{4\|\mathbf{P}\|_F}\lambda_{\min}(\mathbf{Q})$, $\|g_1(\boldsymbol{\theta}, \boldsymbol{\gamma})\| \leq c\|\boldsymbol{\theta}\|$, then we have that,

$$\dot{V}(\boldsymbol{\theta}, \boldsymbol{\gamma}) \leq -\lambda_{\min}(\mathbf{Q})\|\boldsymbol{\theta}\|^2 + \frac{2}{4\|\mathbf{P}\|_F}\lambda_{\min}(\mathbf{Q})\|\mathbf{P}\|_F\|\boldsymbol{\theta}\|^2 = -\frac{1}{2}\lambda_{\min}(\mathbf{Q})\|\boldsymbol{\theta}\|^2 < 0$$

Now, as long as we ensure that the trajectory of the system remains in the neighborhood around origin for which $|g_1(\boldsymbol{\theta}, \boldsymbol{\gamma})\| \leq \frac{1}{4\|\mathbf{P}\|_F}\lambda_{\min}(\mathbf{Q})\|\boldsymbol{\theta}\|$ and $\|\boldsymbol{\gamma}\| < \epsilon$, this system would then exponentially converge to one of the equilibria near origin. Let us call this neighborhood $\mathcal{S}$ i.e., within this neighborhood of $\boldsymbol{\gamma}$ and $\boldsymbol{\theta}$, the Lyapunov function strictly decreases for the non-linear system.

This brings us to the second crucial part of this proof, which is to ensure that we always stay in $\mathcal{S}$. Let $\mathcal{S}$ contain a ball of radius $d$. We will show that for sufficiently close initializations which are within a ball of radius $d/2$, the displacement of $\boldsymbol{\gamma}$ is at most $d/2$. Since $\boldsymbol{\theta}$ only approaches origin, this means that the system never exited $\mathcal{S}$.

To bound how much $\boldsymbol{\gamma}$ changes with time, let us consider the Taylor series expansion of $h_2(\boldsymbol{\theta}, \boldsymbol{\gamma})$. First of all, there is no constant term. Next, there is no term that is purely a function of $\boldsymbol{\gamma}$ because $h_2(\mathbf{0}, \boldsymbol{\gamma}) = 0$. Then, we can say that:

$$h_2(\boldsymbol{\theta}, \boldsymbol{\gamma}) = g_4(\boldsymbol{\theta}, \boldsymbol{\gamma})\boldsymbol{\theta}$$

Since $g_4(\mathbf{0}, \mathbf{0})$ is finite, in a small neighborhood around equilibrium, there exists a fixed constant $c'$ such that $\|g_4(\boldsymbol{\theta}, \boldsymbol{\gamma})\|_2 \leq c'$. Then, $h_2(\boldsymbol{\theta}, \boldsymbol{\gamma}) \leq c'\|\boldsymbol{\theta}\|$.

Figure 5: **Illustration of Theorem A.4**: $\mathcal{S}$ is the neighborhood within which $\boldsymbol{\theta}$ converges exponentially to $\mathbf{0}$ to a point on the $\boldsymbol{\gamma}$-axis which corresponds to an equilibrium. However, all initializations within $\mathcal{S}$ may not preserve the trajectory within $\mathcal{S}$ due to a lack of guarantee on how $\boldsymbol{\gamma}$ behaves – as illustrated by the dashed trajectory. We identify a smaller ball within $\mathcal{S}$ such that for any intitialization within that ball, $\boldsymbol{\gamma}$ is well-behaved and consqently ensures exponential convergence of $\boldsymbol{\theta}$.

Now, if the trajectory indeed always remained in $\mathcal{S}$, we know that $\|\boldsymbol{\theta}(t)\| = \|\boldsymbol{\theta}(0)\| \exp\left(-c''t\right)$ for some constant $c'' > 0$. Assume we initialize $\boldsymbol{\theta}(0)$ within a radius of $\frac{c''d}{2c'}$. The rate at which $\boldsymbol{\gamma}$ changes at any point is,

$$\|\dot{\boldsymbol{\gamma}}\| \leq c'\|\boldsymbol{\theta}(0)\| \exp\left(-c''t\right)$$

Then, the maximum displacement in $\boldsymbol{\gamma}$ can be,

$$\int_{t=0}^{\infty} c'\|\boldsymbol{\theta}(0)\| \exp\left(-c''t\right) dt = c'\frac{\|\boldsymbol{\theta}(0)\|}{c''} \leq \frac{d}{2}$$

Thus, the trajectory always lies in $\mathcal{S}$, which implies exponential convergence along $\boldsymbol{\theta}$ to a point where $\boldsymbol{\theta} = 0$ and $\|\boldsymbol{\gamma}\| < \epsilon$. Thus the system exponentially converges to an equilibrium.

$\square$

# B  GANs are not concave-convex near equilibrium

In this section, we consider a more general system than the one considered in the main paper to demonstrate that GANs are not concave-convex near equilibrium. In particular, consider the following discriminator and generator pair learning a distribution in 1-D:

$$D_{\mathbf{w}}(x) = \sum_{i=0}^{d_D} w_i x^i$$

$$G_{\mathbf{a}}(z) = \sum_{j=0}^{d_G} a_j z^j$$

where $d_D \geq 1$ and $d_G \geq 1$. Let the distribution to be learned be arbitrary. Let the latent distribution be the standard normal. Then, the gradient of the objective with respect to the generator parameters is:

$$\frac{\partial V(G, D)}{\partial a_j} = -\mathbb{E}_{z \sim \mathcal{N}(0,1)} \left[ f'\left( -\sum_{i=0}^{d_D} w_i (G_{\mathbf{a}}(z))^i \right) \cdot \left( \sum_{i=1}^{d_D} i w_i (G_{\mathbf{a}}(z))^{i-1} \right) \cdot z^j \right]$$

The second derivative is,

$$\frac{\partial^2 V(G, D)}{\partial a_j^2} = -\mathbb{E}_{z \sim \mathcal{N}(0,1)} \left[ f'\left(-\sum_{i=0}^{d_D} w_i(G_{\mathbf{a}}(z))^i\right) \cdot \left(\sum_{i=2}^{d_D} i(i-1)w_i(G_{\mathbf{a}}(z))^{i-2}\right) \cdot z^{2j} \right]$$

$$+ \mathbb{E}_{z \sim \mathcal{N}(0,1)} \left[ f''\left(-\sum_{i=0}^{d_D} w_i(G_{\mathbf{a}}(z))^i\right) \cdot \left(\left(\sum_{i=1}^{d_D} iw_i(G_{\mathbf{a}}(z))^{i-1}\right) \cdot z^j\right)^2 \right]$$

Now, consider the case where $f''(x) < 0$. For points in the discriminator parameter space where $w_1 \neq 0$ but $w_i = 0$ for all $i \neq 1$, the term above simplifies to the following when $j \neq 1$:

$$\mathbb{E}_{z \sim \mathcal{N}(0,1)} \left[ f''\left(-w_1(G_{\mathbf{a}}(z))\right) \cdot \left(w_1 z^j\right)^2 \right]$$

which is clearly negative, i.e., the objective is concave in most of the generator parameters, and this holds for parameters arbitrarily close to the all-zero discriminator parameter (as $w_1 \rightarrow 0$).

On the other hand, consider the case where $f''(x) = 0$ for all $x \in \mathbb{R}$. Then, if $d_D > 2$, we can consider $w_2 \neq 0$ while $w_i = 0$ for all $i \neq 2$. In this case, the second derivative simplifies to:

$$-\mathbb{E}_{z \sim \mathcal{N}(0,1)} \left[ f'\left(-w_2(G_{\mathbf{a}}^2(z))\right) 2w_2 z^{2j} \right].$$

If $f'(x) > 0$ for all $x$ (which is true in the case of WGANs), then in the region $w_2 > 0$ the above term is negative, i.e., the GAN objective is concave in terms of the generator parameters.

## C  Local exponential stability of GANs

In this section, we provide the full proof for our result about the local stability of GANs through the following lemmas. First, we derive the Jacobian at equilibrium.

**Lemma C.1.** *For the dynamical system defined by the GAN objective in Equation 2 and the updates in Equation 3, the Jacobian at an equilibrium point* $(\theta_{\mathbf{D}}^\star, \theta_{\mathbf{G}}^\star)$, *under the Assumptions I and IV is:*

$$\mathbf{J} = \begin{bmatrix} \mathbf{J}_{DD} & \mathbf{J}_{DG} \\ -\mathbf{J}_{DG}^T & \mathbf{J}_{GG} \end{bmatrix} = \begin{bmatrix} 2f''(0)\mathbf{K}_{DD} & f'(0)\mathbf{K}_{DG} \\ -f'(0)\mathbf{K}_{DG}^T & 0 \end{bmatrix}$$

*where*

$$\mathbf{K}_{DD} \triangleq \mathbb{E}_{p_{\text{data}}}[(\nabla_{\theta_{\mathbf{D}}} D_{\theta_{\mathbf{D}}}(x))(\nabla_{\theta_{\mathbf{D}}} D_{\theta_{\mathbf{D}}}(x))^T]|_{\theta_{\mathbf{D}}^\star} \succeq 0$$

*and*

$$\mathbf{K}_{DG} \triangleq \int_{\mathcal{X}} \nabla_{\theta_{\mathbf{D}}} D_{\theta_{\mathbf{D}}}(x) \nabla_{\theta_{\mathbf{G}}}^T p_{\theta_{\mathbf{G}}}(x) dx \Big|_{\theta_{\mathbf{D}} = \theta_{\mathbf{D}}^\star, \theta_{\mathbf{G}} = \theta_{\mathbf{G}}^\star}$$

*Proof.* To derive the Jacobian, we begin with a subtly different algebraic form of the GAN objective in Equation 2 by replacing the term $\mathbb{E}_{z \sim p_{\text{latent}}}[f(-D_{\theta_{\mathbf{D}}}(G_{\theta_{\mathbf{G}}}(z)))]$ with $\mathbb{E}_{p_{\theta_{\mathbf{G}}}}[f(-D_{\theta_{\mathbf{D}}}(x))] = \int_{\mathcal{X}} p_{\theta_{\mathbf{G}}}(x)f(-D_{\theta_{\mathbf{D}}}(x))$. Effectively, we separate the discriminator and the generator's effects in this term. This is crucial because we will proceed with all of our analysis in this form. Observe that the system then becomes,

$$V(D_{\theta_{\mathbf{D}}}, G_{\theta_{\mathbf{G}}}) = \mathbb{E}_{p_{\text{data}}}[f(D_{\theta_{\mathbf{D}}}(x))] + \mathbb{E}_{p_{\theta_{\mathbf{G}}}}[f(-D_{\theta_{\mathbf{D}}}(x))]$$

$$\dot{\theta}_{\mathbf{D}} = \mathbb{E}_{p_{\text{data}}}[f'(D_{\theta_{\mathbf{D}}}(x))\nabla_{\theta_{\mathbf{D}}} D_{\theta_{\mathbf{D}}}(x)] - \mathbb{E}_{p_{\theta_{\mathbf{G}}}}[f'(-D_{\theta_{\mathbf{D}}}(x))\nabla_{\theta_{\mathbf{D}}} D_{\theta_{\mathbf{D}}}(x)]$$

$$\dot{\theta}_{\mathbf{G}} = -\int_{\mathcal{X}} \nabla_{\theta_{\mathbf{G}}} p_{\theta_{\mathbf{G}}}(x)f(-D_{\theta_{\mathbf{D}}}(x))dx$$

Throughout this paper we will use the notation $\nabla^T(\cdot)$ to denote the row vector corresponding to the gradient that is being computed. Now, let $n_D$ be the number of discriminator parameters and $n_G$ the

number of generator parameters. Then the first $n_D \times n_D$ block in $\mathbf{J}$, which we will denote by $\mathbf{J}_{DD}$ is:

$$
\begin{aligned}
\mathbf{J}_{DD} &\triangleq \nabla^2_{\boldsymbol{\theta}_{\mathbf{D}}} V(G_{\boldsymbol{\theta}_{\mathbf{G}}}, D_{\boldsymbol{\theta}_{\mathbf{D}}})\big|_{(\boldsymbol{\theta}_{\mathbf{D}}^{\star}, \boldsymbol{\theta}_{\mathbf{G}}^{\star})} = \frac{\partial \dot{\boldsymbol{\theta}}_{\mathbf{D}}}{\partial \boldsymbol{\theta}_{\mathbf{D}}}\bigg|_{\boldsymbol{\theta}_{\mathbf{D}} = \boldsymbol{\theta}_{\mathbf{D}}^{\star}, \boldsymbol{\theta}_{\mathbf{G}} = \boldsymbol{\theta}_{\mathbf{G}}^{\star}} = \frac{\partial \left.\dot{\boldsymbol{\theta}}_{\mathbf{D}}\right|_{\boldsymbol{\theta}_{\mathbf{G}} = \boldsymbol{\theta}_{\mathbf{G}}^{\star}}}{\partial \boldsymbol{\theta}_{\mathbf{D}}}\bigg|_{\boldsymbol{\theta}_{\mathbf{D}} = \boldsymbol{\theta}_{\mathbf{D}}^{\star}} \\
&= \frac{\partial \left( \mathbb{E}_{p_{\text{data}}}[f'(D_{\boldsymbol{\theta}_{\mathbf{D}}}(x))\nabla_{\boldsymbol{\theta}_{\mathbf{D}}} D_{\boldsymbol{\theta}_{\mathbf{D}}}(x)] - \mathbb{E}_{p_{\text{data}}}[f'(-D_{\boldsymbol{\theta}_{\mathbf{D}}}(x))\nabla_{\boldsymbol{\theta}_{\mathbf{D}}} D_{\boldsymbol{\theta}_{\mathbf{D}}}(x)] \right)}{\partial \boldsymbol{\theta}_{\mathbf{D}}}\bigg|_{\boldsymbol{\theta}_{\mathbf{D}} = \boldsymbol{\theta}_{\mathbf{D}}^{\star}} \\
&= \left( \mathbb{E}_{p_{\text{data}}}\left[ f''(D_{\boldsymbol{\theta}_{\mathbf{D}}}(x))\nabla_{\boldsymbol{\theta}_{\mathbf{D}}} D_{\boldsymbol{\theta}_{\mathbf{D}}}(x)\nabla^T_{\boldsymbol{\theta}_{\mathbf{D}}} D_{\boldsymbol{\theta}_{\mathbf{D}}}(x) \right] + \mathbb{E}_{p_{\text{data}}}\left[ f'(D_{\boldsymbol{\theta}_{\mathbf{D}}}(x))\nabla^2_{\boldsymbol{\theta}_{\mathbf{D}}} D_{\boldsymbol{\theta}_{\mathbf{D}}}(x) \right] \right)\big|_{\boldsymbol{\theta}_{\mathbf{D}} = \boldsymbol{\theta}_{\mathbf{D}}^{\star}} \\
&\quad + \left( \mathbb{E}_{p_{\text{data}}}\left[ f''(-D_{\boldsymbol{\theta}_{\mathbf{D}}}(x))\nabla_{\boldsymbol{\theta}_{\mathbf{D}}} D_{\boldsymbol{\theta}_{\mathbf{D}}}(x)\nabla^T_{\boldsymbol{\theta}_{\mathbf{D}}} D_{\boldsymbol{\theta}_{\mathbf{D}}}(x) \right] - \mathbb{E}_{p_{\text{data}}}\left[ f'(-D_{\boldsymbol{\theta}_{\mathbf{D}}}(x))\nabla^2_{\boldsymbol{\theta}_{\mathbf{D}}} D_{\boldsymbol{\theta}_{\mathbf{D}}}(x) \right] \right)\big|_{\boldsymbol{\theta}_{\mathbf{D}} = \boldsymbol{\theta}_{\mathbf{D}}^{\star}} \\
&= \left( \mathbb{E}_{p_{\text{data}}}\left[ f''(0)\nabla_{\boldsymbol{\theta}_{\mathbf{D}}} D_{\boldsymbol{\theta}_{\mathbf{D}}}(x)\nabla^T_{\boldsymbol{\theta}_{\mathbf{D}}} D_{\boldsymbol{\theta}_{\mathbf{D}}}(x) \right] + \mathbb{E}_{p_{\text{data}}}\left[ f'(0)\nabla^2_{\boldsymbol{\theta}_{\mathbf{D}}} D_{\boldsymbol{\theta}_{\mathbf{D}}}(x) \right] \right)\big|_{\boldsymbol{\theta}_{\mathbf{D}} = \boldsymbol{\theta}_{\mathbf{D}}^{\star}} \\
&\quad + \left( \mathbb{E}_{p_{\text{data}}}\left[ f''(0)\nabla_{\boldsymbol{\theta}_{\mathbf{D}}} D_{\boldsymbol{\theta}_{\mathbf{D}}}(x)\nabla^T_{\boldsymbol{\theta}_{\mathbf{D}}} D_{\boldsymbol{\theta}_{\mathbf{D}}}(x) \right] - \mathbb{E}_{p_{\text{data}}}\left[ f'(0)\nabla^2_{\boldsymbol{\theta}_{\mathbf{D}}} D_{\boldsymbol{\theta}_{\mathbf{D}}}(x) \right] \right)\big|_{\boldsymbol{\theta}_{\mathbf{D}} = \boldsymbol{\theta}_{\mathbf{D}}^{\star}} \\
&= 2f''(0)\, \mathbb{E}_{p_{\text{data}}}\left[ \nabla_{\boldsymbol{\theta}_{\mathbf{D}}} D_{\boldsymbol{\theta}_{\mathbf{D}}}(x)\nabla^T_{\boldsymbol{\theta}_{\mathbf{D}}} D_{\boldsymbol{\theta}_{\mathbf{D}}}(x) \right]\big|_{\boldsymbol{\theta}_{\mathbf{D}} = \boldsymbol{\theta}_{\mathbf{D}}^{\star}}
\end{aligned}
$$

The subsequent $n_D \times n_G$ matrix, which we will denote by $\mathbf{J}_{DG}$ is:

$$
\begin{aligned}
\mathbf{J}_{DG} &\triangleq \frac{\partial \nabla_{\boldsymbol{\theta}_{\mathbf{D}}} V(G_{\boldsymbol{\theta}_{\mathbf{G}}}, D_{\boldsymbol{\theta}_{\mathbf{D}}})}{\partial \boldsymbol{\theta}_{\mathbf{G}}}\bigg|_{(\boldsymbol{\theta}_{\mathbf{D}}^{\star}, \boldsymbol{\theta}_{\mathbf{G}}^{\star})} = \frac{\partial \dot{\boldsymbol{\theta}}_{\mathbf{D}}}{\partial \boldsymbol{\theta}_{\mathbf{G}}}\bigg|_{\boldsymbol{\theta}_{\mathbf{D}} = \boldsymbol{\theta}_{\mathbf{D}}^{\star}, \boldsymbol{\theta}_{\mathbf{G}} = \boldsymbol{\theta}_{\mathbf{G}}^{\star}} = \frac{\partial \left.\dot{\boldsymbol{\theta}}_{\mathbf{D}}\right|_{\boldsymbol{\theta}_{\mathbf{D}} = \boldsymbol{\theta}_{\mathbf{D}}^{\star}}}{\partial \boldsymbol{\theta}_{\mathbf{G}}}\bigg|_{\boldsymbol{\theta}_{\mathbf{G}} = \boldsymbol{\theta}_{\mathbf{G}}^{\star}} \\
&= \frac{\partial}{\partial \boldsymbol{\theta}_{\mathbf{G}}} \mathbb{E}_{p_{\boldsymbol{\theta}_{\mathbf{G}}}}[f'(0)\nabla_{\boldsymbol{\theta}_{\mathbf{D}}} D_{\boldsymbol{\theta}_{\mathbf{D}}}(x)]\bigg|_{\boldsymbol{\theta}_{\mathbf{D}} = \boldsymbol{\theta}_{\mathbf{D}}^{\star}, \boldsymbol{\theta}_{\mathbf{G}} = \boldsymbol{\theta}_{\mathbf{G}}^{\star}} \\
&= f'(0) \int_{\mathcal{X}} \nabla_{\boldsymbol{\theta}_{\mathbf{D}}} D_{\boldsymbol{\theta}_{\mathbf{D}}}(x)\nabla^T_{\boldsymbol{\theta}_{\mathbf{G}}} p_{\boldsymbol{\theta}_{\mathbf{G}}}(x) dx\bigg|_{\boldsymbol{\theta}_{\mathbf{D}} = \boldsymbol{\theta}_{\mathbf{D}}^{\star}, \boldsymbol{\theta}_{\mathbf{G}} = \boldsymbol{\theta}_{\mathbf{G}}^{\star}} = f'(0)\mathbf{K}_{DG}
\end{aligned}
$$

It is easy to see that the lower $n_G \times n_D$ matrix is $-\mathbf{J}_{DG}^T$:

$$
\begin{aligned}
\frac{\partial \dot{\boldsymbol{\theta}}_{\mathbf{G}}}{\partial \boldsymbol{\theta}_{\mathbf{D}}}\bigg|_{\boldsymbol{\theta}_{\mathbf{D}} = \boldsymbol{\theta}_{\mathbf{D}}^{\star}, \boldsymbol{\theta}_{\mathbf{G}} = \boldsymbol{\theta}_{\mathbf{G}}^{\star}} &= \frac{\partial \left.\dot{\boldsymbol{\theta}}_{\mathbf{G}}\right|_{\boldsymbol{\theta}_{\mathbf{G}} = \boldsymbol{\theta}_{\mathbf{G}}^{\star}}}{\partial \boldsymbol{\theta}_{\mathbf{D}}}\bigg|_{\boldsymbol{\theta}_{\mathbf{D}} = \boldsymbol{\theta}_{\mathbf{D}}^{\star}} \\
&= -\frac{\partial}{\partial \boldsymbol{\theta}_{\mathbf{D}}} \int_{\mathcal{X}} f(D_{\boldsymbol{\theta}_{\mathbf{D}}}(x))\nabla_{\boldsymbol{\theta}_{\mathbf{G}}} p_{\boldsymbol{\theta}_{\mathbf{G}}}(x) dx\bigg|_{\boldsymbol{\theta}_{\mathbf{D}} = \boldsymbol{\theta}_{\mathbf{D}}^{\star}, \boldsymbol{\theta}_{\mathbf{G}} = \boldsymbol{\theta}_{\mathbf{G}}^{\star}} = -\mathbf{J}_{DG}^T
\end{aligned}
$$

Furthermore, the lower $n_G \times n_G$ matrix $\mathbf{J}_{GG}$ turns out to be zero. Here, we will use an implication of Assumption IV. More specifically, generators $\boldsymbol{\theta}_{\mathbf{G}}$ that are within a sufficiently small radius $\epsilon_G$ around the equilibrium have the same support and therefore i) $D_{\boldsymbol{\theta}_{\mathbf{D}}^{\star}}(x) = 0$ for $x$ in this support. Furthermore for all generators within a radius $\epsilon_G/2$, any perturbation of the generator is not going to change the support, and therefore ii) $\nabla_{\boldsymbol{\theta}_{\mathbf{G}}} p_{\boldsymbol{\theta}_{\mathbf{G}}}(x) = 0$ for $x$ that is not in this support. [5]

Now, to show that $\mathbf{J}_{GG}$ is zero, we take any vector $\mathbf{v}$ that is a perturbation in the generator space and show that $\mathbf{v}^T \mathbf{J}_{GG} = 0$. Here, we will use the limit definition of the derivative along a particular direction $\mathbf{v}$.

$$\mathbf{v}^T \left.\frac{\partial \dot{\boldsymbol{\theta}}_{\mathbf{G}}}{\partial \boldsymbol{\theta}_{\mathbf{G}}}\right|_{\boldsymbol{\theta}_{\mathbf{D}}=\boldsymbol{\theta}_{\mathbf{D}}^\star, \boldsymbol{\theta}_{\mathbf{G}}=\boldsymbol{\theta}_{\mathbf{G}}^\star} = \mathbf{v}^T \left.\frac{\partial \dot{\boldsymbol{\theta}}_{\mathbf{G}}}{\partial \boldsymbol{\theta}_{\mathbf{G}}}\right|_{\boldsymbol{\theta}_{\mathbf{D}}=\boldsymbol{\theta}_{\mathbf{D}}^\star}\Bigg|_{\boldsymbol{\theta}_{\mathbf{G}}=\boldsymbol{\theta}_{\mathbf{G}}^\star} = -\lim_{\substack{\boldsymbol{\theta}_{\mathbf{G}}-\boldsymbol{\theta}_{\mathbf{G}}^\star=\epsilon\mathbf{v}\\\epsilon\to 0}} \frac{\int_{\mathcal{X}} f(-D_{\boldsymbol{\theta}_{\mathbf{D}}^\star}(x)) \overbrace{\nabla^T_{\boldsymbol{\theta}_{\mathbf{G}}} p_{\boldsymbol{\theta}_{\mathbf{G}}}(x)}^{0 \text{ for } x \notin \mathrm{supp}(p_{\boldsymbol{\theta}_{\mathbf{G}}^\star})}\, dx}{\epsilon}$$

$$= -\lim_{\substack{\boldsymbol{\theta}_{\mathbf{G}}-\boldsymbol{\theta}_{\mathbf{G}}^\star=\epsilon\mathbf{v}\\\epsilon\to 0}} \frac{\int_{\mathrm{supp}(p_{\boldsymbol{\theta}_{\mathbf{G}}^\star})} f(-\overbrace{D_{\boldsymbol{\theta}_{\mathbf{D}}^\star}(x)}^{0})\nabla^T_{\boldsymbol{\theta}_{\mathbf{G}}} p_{\boldsymbol{\theta}_{\mathbf{G}}}(x)dx}{\epsilon}$$

$$= -f(0) \lim_{\substack{\boldsymbol{\theta}_{\mathbf{G}}-\boldsymbol{\theta}_{\mathbf{G}}^\star=\epsilon\mathbf{v}\\\epsilon\to 0}} \frac{\nabla^T_{\boldsymbol{\theta}_{\mathbf{G}}}\int_{\mathrm{supp}(p_{\boldsymbol{\theta}_{\mathbf{G}}^\star})} p_{\boldsymbol{\theta}_{\mathbf{G}}}(x)dx}{\epsilon}$$

$$= -f(0) \lim_{\substack{\boldsymbol{\theta}_{\mathbf{G}}-\boldsymbol{\theta}_{\mathbf{G}}^\star=\epsilon\mathbf{v}\\\epsilon\to 0}} \frac{\nabla^T_{\boldsymbol{\theta}_{\mathbf{G}}} 1}{\epsilon} = 0$$

$\square$

To prove that the system is stable we will need to show that this matrix is Hurwitz. We show later in Lemma G.2 that when i) $\mathbf{J}_{DD} \prec 0$ and furthermore ii) $\mathbf{J}_{DG}$ is full column rank, then $\mathbf{J}$ is indeed Hurwitz. However from $f''(0) < 0$, we only have that $\mathbf{J}_{DD} \preceq 0$. For these two conditions to be met, we will need $\mathbf{K}_{DD}$ and $\mathbf{K}_{DG}^T \mathbf{K}_{DG}$ to be full rank, which you may recall from our discussion in the main paper below Assumption III, is met only when there is a unique equilibrim locally.

Now, we show why this is the case – by establishing a relation between the matrices $\mathbf{K}_{DD}$ and $\mathbf{K}_{DG}$ and the curvature of functions in Assumption III – and further show how the null spaces of these matrices correspond to a subspace of equilibria. Then, we show in Lemma C.3, how to consider a rotation of the system and project to a space that is orthogonal to this subspace of equilibria. Then from the Theorem A.4 that we have proved in Appendix A, it is sufficient to show that the Jacobian of the projected system is Hurwitz.

In the following discussion, we will use the term "equilibrium discriminator" to denote a discriminator that is identically zero on the support and "equilibrium generator" to denote a generator that matches the true distribution, as defined in Assumption I. Note that for an equilibrium discriminator, the generator updates are zero and vice versa for an equilibrium generator.

**Lemma C.2.** *For the dynamical system defined by the GAN objective in Equation 2 and the updates in Equation 3, under Assumptions I and III, there exists $\epsilon_D, \epsilon_G > 0$ such that for all $\epsilon'_D \leq \epsilon_D$ and $\epsilon'_G \leq \epsilon_G$, and for any unit vectors $\mathbf{u} \in \mathsf{Null}(\mathbf{K}_{DD}), \mathbf{v} \in \mathsf{Null}(\mathbf{K}_{DG})$, $(\boldsymbol{\theta}_{\mathbf{D}}^\star + \epsilon'_D \mathbf{u}, \boldsymbol{\theta}_{\mathbf{G}}^\star + \epsilon'_G \mathbf{v})$ is an equilibrium point as defined in Assumption I.*

*Proof.* Note that $2\mathbf{K}_{DD}$ is the Hessian of the function $\mathbb{E}_{p_{\mathrm{data}}}[D_{\boldsymbol{\theta}_{\mathbf{D}}}^2(x)]$ at equilibrium:

$$\nabla^2_{\boldsymbol{\theta}_{\mathbf{D}}}\mathbb{E}_{p_{\mathrm{data}}}[D_{\boldsymbol{\theta}_{\mathbf{D}}}^2(x)]\big|_{\boldsymbol{\theta}_{\mathbf{D}}^\star} = 2\left.\frac{\partial \mathbb{E}_{p_{\mathrm{data}}}[D_{\boldsymbol{\theta}_{\mathbf{D}}}(x)\nabla_{\boldsymbol{\theta}_{\mathbf{D}}} D(x)]}{\partial \boldsymbol{\theta}_{\mathbf{D}}}\right|_{\boldsymbol{\theta}_{\mathbf{D}}^\star}$$

$$= 2\left(\mathbb{E}_{p_{\mathrm{data}}}[\nabla_{\boldsymbol{\theta}_{\mathbf{D}}} D_{\boldsymbol{\theta}_{\mathbf{D}}}(x)\nabla^T_{\boldsymbol{\theta}_{\mathbf{D}}} D(x)] + \mathbb{E}_{p_{\mathrm{data}}}[\underbrace{D_{\boldsymbol{\theta}_{\mathbf{D}}}(x)}_{0 \text{ at eqbm}} \nabla^2_{\boldsymbol{\theta}_{\mathbf{D}}} D(x)]\right)\Bigg|_{\boldsymbol{\theta}_{\mathbf{D}}^\star}$$

$$= 2\left(\mathbb{E}_{p_{\mathrm{data}}}[\nabla_{\boldsymbol{\theta}_{\mathbf{D}}} D_{\boldsymbol{\theta}_{\mathbf{D}}}(x)\nabla^T_{\boldsymbol{\theta}_{\mathbf{D}}} D(x)]\right)\big|_{\boldsymbol{\theta}_{\mathbf{D}}^\star} = 2\mathbf{K}_{DD}$$

Then, by Assumption III, $\mathbb{E}_{p_{\mathrm{data}}}[D_{\boldsymbol{\theta}_{\mathbf{D}}}^2(x)]$ is locally constant along any unit vector $\mathbf{u} \in \mathsf{Null}(\mathbf{K}_{DD})$. That is, for sufficiently small $\epsilon$, if $\boldsymbol{\theta}_{\mathbf{D}} = \boldsymbol{\theta}_{\mathbf{D}}^\star + \epsilon\mathbf{u}$, $\mathbb{E}_{p_{\mathrm{data}}}[D_{\boldsymbol{\theta}_{\mathbf{D}}}^2(x)]$ equals the value of the function at

equilibrium, which is 0 because $D_{\boldsymbol{\theta}_{\mathbf{D}}^\star(x)} = 0$ (according to Assumption I). Thus, we can conclude that for all $x$ in the support of $p_{\text{data}}$, $D_{\boldsymbol{\theta}_{\mathbf{D}}}(x) = 0$. Then, the generator update is zero, because

$$\dot{\boldsymbol{\theta}}_{\mathbf{G}} = -f(0) \int_{\text{supp}(p_{\text{data}})} \nabla_{\boldsymbol{\theta}_{\mathbf{G}}} p_{\boldsymbol{\theta}_{\mathbf{G}}}(x)dx = -f(0)\nabla_{\boldsymbol{\theta}_{\mathbf{G}}} \int_{\text{supp}(p_{\text{data}})} p_{\boldsymbol{\theta}_{\mathbf{G}}}(x)dx = -f(0)\nabla_{\boldsymbol{\theta}_{\mathbf{G}}} 1 = 0.$$

In other words, $\boldsymbol{\theta}_{\mathbf{D}}$ is an equilibrium discriminator which when paired with any generator results in zero updates on the generator.

Similarly, $2\mathbf{K}_{DG}^T \mathbf{K}_{DG}$ is the Hessian of the function $\left\| \mathbb{E}_{p_{\text{data}}}[\nabla_{\boldsymbol{\theta}_{\mathbf{D}}} D_{\boldsymbol{\theta}_{\mathbf{D}}}(x)] - \mathbb{E}_{p_{\boldsymbol{\theta}_{\mathbf{G}}}}[\nabla_{\boldsymbol{\theta}_{\mathbf{D}}} D_{\boldsymbol{\theta}_{\mathbf{D}}}(x)] \right\|^2$ at equilibrium:

$$\nabla_{\boldsymbol{\theta}_{\mathbf{G}}} \left\| \mathbb{E}_{p_{\text{data}}}[\nabla_{\boldsymbol{\theta}_{\mathbf{D}}} D_{\boldsymbol{\theta}_{\mathbf{D}}}(x)] - \mathbb{E}_{p_{\boldsymbol{\theta}_{\mathbf{G}}}}[\nabla_{\boldsymbol{\theta}_{\mathbf{D}}} D_{\boldsymbol{\theta}_{\mathbf{D}}}(x)] \right\|^2$$

$$= -2\left( \int_{\mathcal{X}} \nabla_{\boldsymbol{\theta}_{\mathbf{G}}} p_{\boldsymbol{\theta}_{\mathbf{G}}}(x) \nabla_{\boldsymbol{\theta}_{\mathbf{D}}} D_{\boldsymbol{\theta}_{\mathbf{D}}}(x)dx \right) \left( \mathbb{E}_{p_{\text{data}}}[\nabla_{\boldsymbol{\theta}_{\mathbf{D}}} D_{\boldsymbol{\theta}_{\mathbf{D}}}(x)] - \mathbb{E}_{p_{\boldsymbol{\theta}_{\mathbf{G}}}}[\nabla_{\boldsymbol{\theta}_{\mathbf{D}}} D_{\boldsymbol{\theta}_{\mathbf{D}}}(x)] \right)$$

$$\implies \nabla^2_{\boldsymbol{\theta}_{\mathbf{G}}} \left\| \mathbb{E}_{p_{\text{data}}}[\nabla_{\boldsymbol{\theta}_{\mathbf{D}}} D_{\boldsymbol{\theta}_{\mathbf{D}}}(x)] - \mathbb{E}_{p_{\boldsymbol{\theta}_{\mathbf{G}}}}[\nabla_{\boldsymbol{\theta}_{\mathbf{D}}} D_{\boldsymbol{\theta}_{\mathbf{D}}}(x)] \right\|^2 \Big|_{\boldsymbol{\theta}_{\mathbf{D}}^\star, \boldsymbol{\theta}_{\mathbf{G}}^\star}$$

$$= 2\left( \int_{\mathcal{X}} \nabla_{\boldsymbol{\theta}_{\mathbf{G}}} p_{\boldsymbol{\theta}_{\mathbf{G}}}(x) \nabla_{\boldsymbol{\theta}_{\mathbf{D}}} D_{\boldsymbol{\theta}_{\mathbf{D}}}(x)dx \right) \left( \int_{\mathcal{X}} \nabla_{\boldsymbol{\theta}_{\mathbf{G}}} p_{\boldsymbol{\theta}_{\mathbf{G}}}(x) \nabla_{\boldsymbol{\theta}_{\mathbf{D}}} D_{\boldsymbol{\theta}_{\mathbf{D}}}(x)dx \right)^T \Big|_{\boldsymbol{\theta}_{\mathbf{D}}^\star, \boldsymbol{\theta}_{\mathbf{G}}^\star}$$

$$- 2\left( \underbrace{\mathbb{E}_{p_{\text{data}}}[\nabla_{\boldsymbol{\theta}_{\mathbf{D}}} D_{\boldsymbol{\theta}_{\mathbf{D}}}(x)] - \mathbb{E}_{p_{\boldsymbol{\theta}_{\mathbf{G}}}}[\nabla_{\boldsymbol{\theta}_{\mathbf{D}}} D_{\boldsymbol{\theta}_{\mathbf{D}}}(x)]}_{0 \text{ at eqbm}} \right)^T \int_{\mathcal{X}} \nabla_{\boldsymbol{\theta}_{\mathbf{D}}} D_{\boldsymbol{\theta}_{\mathbf{D}}}(x) \nabla^2_{\boldsymbol{\theta}_{\mathbf{G}}} p_{\boldsymbol{\theta}_{\mathbf{G}}}(x)dx \Big|_{\boldsymbol{\theta}_{\mathbf{D}}^\star, \boldsymbol{\theta}_{\mathbf{G}}^\star}$$

$$= 2\mathbf{K}_{DG}^T \mathbf{K}_{DG}$$

Then, by Assumption III, $\left\| \mathbb{E}_{p_{\text{data}}}[\nabla_{\boldsymbol{\theta}_{\mathbf{D}}} D_{\boldsymbol{\theta}_{\mathbf{D}}}(x)] - \mathbb{E}_{p_{\boldsymbol{\theta}_{\mathbf{G}}}}[\nabla_{\boldsymbol{\theta}_{\mathbf{D}}} D_{\boldsymbol{\theta}_{\mathbf{D}}}(x)] \right\|^2$ is locally constant along any unit vector $\mathbf{v} \in \mathsf{Null}(\mathbf{K}_{DG})$. That is, for sufficiently small $\epsilon'$, if $\boldsymbol{\theta}_{\mathbf{G}} = \boldsymbol{\theta}_{\mathbf{G}}^\star + \epsilon'\mathbf{v}$, $\left\| \mathbb{E}_{p_{\text{data}}}[\nabla_{\boldsymbol{\theta}_{\mathbf{D}}} D_{\boldsymbol{\theta}_{\mathbf{D}}}(x)] - \mathbb{E}_{p_{\boldsymbol{\theta}_{\mathbf{G}}}}[\nabla_{\boldsymbol{\theta}_{\mathbf{D}}} D_{\boldsymbol{\theta}_{\mathbf{D}}}(x)] \right\|^2$ equals the value of the function at equilibrium, which is 0 because $p_{\boldsymbol{\theta}_{\mathbf{G}}^\star} = p_{\text{data}}$ (according to Assumption I).

Now, we can't immediately conclude that $\boldsymbol{\theta}_G$ corresponds to the true distribution. To show that, we first note that that at $(\boldsymbol{\theta}_{\mathbf{D}}^\star, \boldsymbol{\theta}_{\mathbf{G}})$, the discriminator update, whose magnitude is equal to $|f'(0)| \cdot \left\| \mathbb{E}_{p_{\text{data}}}[\nabla_{\boldsymbol{\theta}_{\mathbf{D}}} D_{\boldsymbol{\theta}_{\mathbf{D}}}(x)] - \mathbb{E}_{p_{\boldsymbol{\theta}_{\mathbf{G}}}}[\nabla_{\boldsymbol{\theta}_{\mathbf{D}}} D_{\boldsymbol{\theta}_{\mathbf{D}}}(x)] \right\|$, is zero. However, as we have seen at $\boldsymbol{\theta}_{\mathbf{D}}^\star$ the generator update is zero too. Therefore, $(\boldsymbol{\theta}_{\mathbf{D}}^\star, \boldsymbol{\theta}_{\mathbf{G}})$ is an equilibrium point (both updates are zero) and from Assumption I we can conclude that $p_{\boldsymbol{\theta}_{\mathbf{G}}} = p_{\text{data}}$. Thus, $\boldsymbol{\theta}_{\mathbf{G}}$ is an equilibrium generator i.e., when paired with any equilibrium discriminator, the discriminator updates are zero.

In summary, for all slight perturbations along $\mathbf{u} \in \mathsf{Null}(\mathbf{K}_{DD})$, $\mathbf{v} \in \mathsf{Null}(\mathbf{K}_{DG})$ we have established that the discriminator and generator individually satisfy the requirements of an equilibrium discriminator and generator pair, and therefore the system is itself in equilibrium for these perturbations. $\quad\square$

Now, we show how to rotate and project the system to get a Hurwitz Jacobian matrix.

**Lemma C.3.** *For the dynamical system defined by the GAN objective in Equation 2 and the updates in Equation 3, consider the eigenvalue decompositions* $\mathbf{K}_{DD} = \mathbf{U}_{\mathbf{D}} \boldsymbol{\Lambda}_{\mathbf{D}} \mathbf{U}_{\mathbf{D}}{}^T$ *and* $\mathbf{K}_{DG}^T \mathbf{K}_{DG} = \mathbf{U}_{\mathbf{G}} \boldsymbol{\Lambda}_{\mathbf{G}} \mathbf{U}_{\mathbf{G}}{}^T$. *Let* $\mathbf{U}_{\mathbf{D}} = [\mathbf{T}_D^T, \mathbf{T}_D'^T]$ *and* $\mathbf{U}_{\mathbf{G}} = [\mathbf{T}_G^T, \mathbf{T}_G'^T]$ *such that* $\mathsf{Col}(\mathbf{T}_D'^T) = \mathsf{Null}(\mathbf{K}_{DD})$ *and* $\mathsf{Col}(\mathbf{T}_G'^T) = \mathsf{Null}(\mathbf{K}_{DG})$. *Consider the projections,* $\boldsymbol{\gamma}_{\mathbf{D}} = \mathbf{T}_D \boldsymbol{\theta}_D$ *and* $\boldsymbol{\gamma}_{\mathbf{G}} = \mathbf{T}_G \boldsymbol{\theta}_G$. *Then, the block in the Jacobian at equilibrium that corresponds to the projected system has the form:*

$$\mathbf{J}' = \begin{bmatrix} \mathbf{J}'_{DD} & \mathbf{J}'_{DG} \\ -\mathbf{J}'^T_{DG} & 0 \end{bmatrix} = \begin{bmatrix} 2f''(0)\mathbf{T}_D \mathbf{K}_{DD} \mathbf{T}_D^T & f'(0)\mathbf{T}_D \mathbf{K}_{DG} \mathbf{T}_G^T \\ -f'(0)\mathbf{T}_G \mathbf{K}_{DG}^T \mathbf{T}_D^T & 0 \end{bmatrix}$$

*Under Assumption II, we have that $\mathbf{J}'_{DD} \prec 0$ and $\mathbf{J}'_{DG}$ is full column rank.*

*Proof.* Note that the columns of $\mathbf{U}_D$ and $\mathbf{U}_G$ correspond to eigenvectors, and furthermore, the rows of $\mathbf{T}'_D$ and $\mathbf{T}'_G$ are the eigenvectors that correspond to zero eigenvalues. These eigenvectors correspond to a local subspace of equilibria and the above lemma considers a projection of the system to a space orthogonal to this subspace.

We first address a corner case where either $\mathbf{T}_D$ or $\mathbf{T}_G$ (the eigenvectors with non-zero eigenvalues) is empty. In the case that $\mathbf{T}_D$ is empty, it means that all discriminators in a neighborhood of the considered equilibrium are identically zero on the support of the true distribution (as proved in Lemma C.2). Then, for any generator, the discriminator update would be zero (because moving the discriminator in any direction locally does not result in a change in the objective). At the same time, the generator update would be zero too because these are all equilibrium discriminators. This means that the considered point is surrounded by a neighborhood of equilibria. Then, the system is trivially exponentially stable since any sufficiently close initialization is already at equilibrium.

Similarly when $\mathbf{T}_G$ is empty it means that all generators in a small neighborhood have the same distribution, namely the true underlying distribution (as proved in Lemma C.2). Then, the generator update for any discriminator would be zero (changing the generator slightly in any direction does not change the generated distribution, and hence the objective). Furthermore, since these are equilibrium generators, the discriminator updates would be zero too, for any discriminator. Thus, again we are situated in a neighborhood of equilibria and the system is trivially exponentially stable.

Now we handle the general case. First note that, the Jacobian block of the projected variables must be

$$\left( \begin{bmatrix} \mathbf{T_D} \\ \mathbf{T_G} \end{bmatrix} \mathbf{J} \begin{bmatrix} \mathbf{T_D}^T & \mathbf{T_G}^T \end{bmatrix} \right) = \begin{bmatrix} 2f''(0)\mathbf{T}_D\mathbf{K}_{DD}\mathbf{T}_D^T & f'(0)\mathbf{T}_D\mathbf{K}_{DG}\mathbf{T}_G^T \\ -f'(0)\mathbf{T}_G\mathbf{K}_{DG}^T\mathbf{T}_D^T & 0 \end{bmatrix}$$

where $\mathbf{J}$ is the Jacobian of the original system which we derived in Lemma C.1. Now note that, $\mathbf{T}_D\mathbf{K}_{DD}\mathbf{T}_D^T = \mathbf{T}_D\mathbf{U_D}\mathbf{\Lambda_D}\mathbf{U_D}^T\mathbf{T}_D^T = \mathbf{\Lambda}_D^{(+)}$ which is a diagonal matrix with only the positive eigenvalues. Therefore, since $f''(0) < 0$, $\mathbf{J}'_{DD} \prec 0$.

Next, in a similar manner we can show that $\mathbf{T}_G\mathbf{K}_{DG}^T\mathbf{K}_{DG}\mathbf{T}_G^T = \mathbf{\Lambda}_G^{(+)}$, which is a diagonal matrix with only positive eigenvalues. Thus, $\mathbf{K}_{DG}\mathbf{T}_G^T$ is full column rank. The non-trivial step here is to show that the matrix $\mathbf{T}_D\mathbf{K}_{DG}\mathbf{T}_G^T$ which has fewer rows is full column rank too. This will follow if we showed that for any $\mathbf{u}$ such that $\mathbf{u}^T\mathbf{K}_{DD} = 0$, $\mathbf{u}^T\mathbf{K}_{DG} = 0$ too. That is, the left null space of $\mathbf{K}_{DD}$ is a subset of the left null space of $\mathbf{K}_{DG}$ and therefore projecting to the row span of $\mathbf{K}_{DD}$ does not hurt the row rank of $\mathbf{K}_{DG}$.

To see why this is true, observe that from Lemma C.2 for any small perturbation along such a $\mathbf{u}$, since we are always at an equilibrium discriminator i.e., $\mathbf{D}_{\boldsymbol{\theta}_\mathbf{D}}(x) = 0$ for $x$ in the true support, it must be that $\mathbf{u}^T\nabla_{\boldsymbol{\theta}_\mathbf{D}}\mathbf{D}_{\boldsymbol{\theta}_\mathbf{D}}(x) = 0$. Furthermore, recall from our derivation of the Jacobian that $\nabla_{\boldsymbol{\theta}_\mathbf{G}}p_{\boldsymbol{\theta}_\mathbf{G}}(x) = 0$ for $x$ outside of this support. Then,

$$\mathbf{u}^T\mathbf{K}_{DG} = \int_{\mathcal{X}} \underbrace{\mathbf{u}^T\nabla_{\boldsymbol{\theta}_\mathbf{D}}D_{\boldsymbol{\theta}_\mathbf{D}}(x)}_{\text{0 inside supp}} \underbrace{\nabla_{\boldsymbol{\theta}_\mathbf{G}}^T p_{\boldsymbol{\theta}_\mathbf{G}}(x)}_{\text{0 outside supp}} dx \Bigg|_{\boldsymbol{\theta}_\mathbf{D}=\boldsymbol{\theta}_\mathbf{D}^\star, \boldsymbol{\theta}_\mathbf{G}=\boldsymbol{\theta}_\mathbf{G}^\star} = 0$$

Therefore, since $f'(0) \neq 0$, this means $f'(0)\mathbf{T}_D\mathbf{K}_{DG}\mathbf{T}_G^T$ is full column rank.

$\square$

The main theorem then follows from the above lemmas.

**Theorem 3.1.** *The dynamical system defined by the GAN objective in Equation 2 and the updates in Equation 3 is locally exponentially stable with respect to an equilibrium point $(\boldsymbol{\theta}_\mathbf{D}^\star, \boldsymbol{\theta}_\mathbf{G}^\star)$ when the Assumptions I, II, III, IV hold for $(\boldsymbol{\theta}_\mathbf{D}^\star, \boldsymbol{\theta}_\mathbf{G}^\star)$ and other equilibria in a small neighborhood around it. Furthermore, the rate of convergence is governed only by the eigenvalues $\lambda$ of the Jacobian $\mathbf{J}$ of the system at equilibrium with a strict negative real part upper bounded as:*

- *If* $\text{Im}(\lambda) = 0$, *then* $\text{Re}(\lambda) \leq \dfrac{2f''(0)f'^2(0)\lambda_{\min}^{(+)}(\mathbf{K}_{DD})\lambda_{\min}^{(+)}(\mathbf{K}_{DG}^T\mathbf{K}_{DG})}{4f''^2(0)\lambda_{\min}^{(+)}(\mathbf{K}_{DD})\lambda_{\max}(\mathbf{K}_{DD})+f'(0)^2\lambda_{\min}^{(+)}(\mathbf{K}_{DG}^T\mathbf{K}_{DG})}$

- *If* $\text{Im}(\lambda) \neq 0$, *then* $\text{Re}(\lambda) \leq f''(0)\lambda_{\min}^{(+)}(\mathbf{K}_{DD})$

*Proof.* We have from Lemma C.2 that the considered equilibrium point lies in a subspace of equilibria in a small neighborhood. Then, we have from Lemma C.3 that the Jacobian block corresponding to the subspace orthogonal to this, satsifies properties from Lemma G.2 which make it Hurwitz. We can then conclude exponential stability of the system from Theorem A.4. The eigenvalue bounds presented in the theorem follow from Lemma G.2. □

Finally, we show that we can indeed find a Lyapunov function that satisfies LaSalle's principle for the projected linearized system.

**Fact C.1.** *For the linearized projected system with the Jacobian* $\mathbf{J}'$, *we have that* $1/2\|\boldsymbol{\gamma}_D - \boldsymbol{\gamma}^\star{}_D\|^2 + 1/2\|\boldsymbol{\gamma}_G - \boldsymbol{\gamma}^\star{}_G\|^2$ *is a Lyapunov function such that for all non-equilbrium points, it either always decreases or only instantaneously remains constant.*

*Proof.* Note that the Lyapunov function is zero only at the equilibrium of the projected system. Furthermore, it is straightforward to verify that the rate at which this changes is given by $f''(0)(\boldsymbol{\gamma}_D - \boldsymbol{\gamma}^\star{}_D)^T\mathbf{T}_D^T\mathbf{K}_{DD}\mathbf{T}_D(\boldsymbol{\gamma}_D - \boldsymbol{\gamma}^\star{}_D)$. Observe that the generator terms have canceled out. Clearly this is zero only when $\boldsymbol{\gamma}_D = \boldsymbol{\gamma}^\star{}_D$ because $\mathbf{T}_D^T\mathbf{K}_{DD}\mathbf{T}_D$ is positive definite; otherwise it is strictly negative. Now, when this rate is indeed zero, we have that $\dot{\boldsymbol{\gamma}}_{\mathbf{D}} = f'(0)\mathbf{T}_D\mathbf{K}_{DG}\mathbf{T}_G^T(\boldsymbol{\gamma}_G - \boldsymbol{\gamma}^\star{}_G)$ because the other term in the update which is proportional to $\mathbf{K}_{DD}(\boldsymbol{\gamma}_D - \boldsymbol{\gamma}^\star{}_D)$ is zero. Now, again, this term is zero only when $\boldsymbol{\gamma}_G = \boldsymbol{\gamma}^\star{}_G$ because $\mathbf{T}_D\mathbf{K}_{DG}\mathbf{T}_G^T$ is full column rank. Thus, when we are not at equilibrium which means $\boldsymbol{\gamma}_G \neq \boldsymbol{\gamma}_{\mathbf{G}}^\star$, the update on the discriminator parameters is nonzero i.e., $\dot{\boldsymbol{\gamma}}_{\mathbf{D}} \neq 0$. In other words, it does not identically stay in the manifold $\boldsymbol{\gamma}_D = 0$ on which the energy does not decrease. □

### C.1   Realizable case with a relaxed assumption

In this section, we will relax Assumption IV and prove stability under certain conditions. Specifically, recall that originally we required the equilibrium generator to share the same support with any perturbation of the generator. Now, we will allow the generator to have different supports when perturbed, and instead impose conditions on the discriminator.

Our first condition is that the equilibrium discriminator must be zero not only on the support of $\boldsymbol{\theta}_G^\star$ but also on the supports of small perturbations of $\boldsymbol{\theta}_{\mathbf{G}}^\star$. If this were not true, $\boldsymbol{\theta}_{\mathbf{G}}^\star$ may not be at equilibrium as the slope of the discriminator function $D_{\boldsymbol{\theta}_{\mathbf{D}}^\star}(x)$ may be non-zero at the boundaries of $\text{supp}(p_{\boldsymbol{\theta}_G^\star})$ in $\mathcal{X}$, thus potentially encouraging the generator to push data points away from the true support.

To motivate our second condition, recall from Assumption III, we have that there could be directions along which we can perturb $\boldsymbol{\theta}_{\mathbf{D}}^\star$, while ensuring that the discriminator still outputs zero on $\text{supp}(p_{\boldsymbol{\theta}_{\mathbf{G}}^\star})$. The intention behind allowing this was that these directions could allow other equivalent equilibrium discriminators in the neighborhood of $\boldsymbol{\theta}_{\mathbf{D}}^\star$. However, under the relaxation of Assumption IV that we are now aiming for, these perturbations will correspond to equilibrium discriminators only if they satsify the above condition i.e., that they are zero on the support of perturbations of $\boldsymbol{\theta}_{\mathbf{G}}^\star$ too. We need to explicitly assume that this holds as we describe below. [6]

To state this assumption using the terminology we've developed so far, recall that imposing Property I on the function $\mathbb{E}_{p_{\text{data}}}[D_{\boldsymbol{\theta}_{\mathbf{D}}^\star}^2(x)]$ at $\boldsymbol{\theta}_{\mathbf{D}}^\star$ (where it attains its minimum of zero) implied that perturbations of $\boldsymbol{\theta}_{\mathbf{D}}^\star$ along the flat directions of the function retains the property that the discriminator is zero on the support of $p_{\text{data}}$ (i.e., $p_{\boldsymbol{\theta}_{\mathbf{G}}^\star}$). Extending this, we will assume that this property holds at $\boldsymbol{\theta}_{\mathbf{D}}^\star$ for the functions $\mathbb{E}_{p_{\boldsymbol{\theta}_G}}[D_{\boldsymbol{\theta}_{\mathbf{D}}}^2(x)]$ corresponding to every small perturbation $\boldsymbol{\theta}_{\mathbf{G}}$ of $\boldsymbol{\theta}_{\mathbf{G}}^\star$. Furthermore, the flat directions of all these functions must be identical so that perturbing $\boldsymbol{\theta}_{\mathbf{D}}^\star$ along these directions guarantees that all these functions are zero. Then, the output of the perturbed discriminator would be zero on the support of all perturbations of $\boldsymbol{\theta}_{\mathbf{G}}^\star$.

Formally, we can state these assumptions as follows:

**Assumption IV (Relaxed)** $\exists \epsilon_G, \epsilon_D > 0$ such that for all $\boldsymbol{\theta_G} \in B_{\epsilon_G}(\boldsymbol{\theta_G^\star})$:

1. for all $x \in \text{supp}(p_{\boldsymbol{\theta_G}})$, $D_{\boldsymbol{\theta_D^\star}}(x) = 0$.

2. at $(\boldsymbol{\theta_D^\star}, \boldsymbol{\theta_G})$, the function $\mathbb{E}_{p_{\boldsymbol{\theta_G}}}[D_{\boldsymbol{\theta_D}}^2(x)]$ satisfies Property I in the discriminator space and furthermore, $\text{Null}\left(\nabla_{\boldsymbol{\theta}_D}^2 \mathbb{E}_{p_{\text{data}}}[D_{\boldsymbol{\theta_D}}^2(x)]\big|_{\boldsymbol{\theta_D}=\boldsymbol{\theta_D^\star}}\right) = \text{Null}\left(\nabla_{\boldsymbol{\theta}_D}^2 \mathbb{E}_{p_{\boldsymbol{\theta_G}}}[D_{\boldsymbol{\theta_D}}^2(x)]\big|_{\boldsymbol{\theta_D}=\boldsymbol{\theta_D^\star}}\right).$

**Examples.** It is useful to illustrate simple examples that satisfy or break the two conditions above, for a clearer picture of what these assumptions imply. First, as an example that satisfies these conditions (and not the original Assumption IV), consider a system where $p_{\text{data}}$ is uniform over $[-1, 1]$ ($\mathcal{X} = \mathbb{R}$), the generator is a uniform distribution over an interval parametrized as $[-\theta_G, \theta_G]$, and the discriminator is any polynomial, for example, a linear function $\theta_D x$. Note that at equilibrium $\theta_D = 0$. Then, it can be verified that for this system the Hessian of $\mathbb{E}_{p_{\text{data}}}[D_{\boldsymbol{\theta_D}}^2(x)]$ is positive definite at equilibrium, thus trivially satisfying the second assumption.

As a simple example that breaks these assumptions, specifically condition (2) above[7], consider a system where $p_{\text{data}}$ is just a point mass at 0 ($\mathcal{X} = \mathbb{R}$), the generator is also a point mass at $\theta_G$ and the discriminator is a linear function $\theta_D x$. Again, at equilibrium $\theta_D = 0$ and $\theta_G = 0$. Surprisingly, even though this is a unique equilibrium, the Hessian of $\mathbb{E}_{p_{\text{data}}}[D_{\boldsymbol{\theta_D}}^2(x)]$ at equilibrium turns out to be zero. Thus, the null space of $\nabla_{\boldsymbol{\theta}_D}^2 \mathbb{E}_{p_{\text{data}}}[D_{\boldsymbol{\theta_D}}^2(x)]$ at equilibrium corresponds to the whole parameter space. On the other hand, at equilibrium $\nabla_{\boldsymbol{\theta}_D}^2 \mathbb{E}_{p_{\boldsymbol{\theta_G}}}[D_{\boldsymbol{\theta_D}}^2(x)] = 2\theta_G^2$, which is non-zero for any $\theta_G$ arbitrarily close to equilibrium. Thus, in the second condition above, while we have a null space for the first Hessian, there is no null space for the second Hessian, thereby breaking the condition. It can be shown that this system which breaks the condition is in fact not locally exponentially stable!

We now show that if these conditions hold, local exponentially stability holds too.

*Proof.* Most of the original proof holds as it is because all we needed was that the equilibrium discriminator be identically zero on the true support. We will prove only parts of the proof that required more than just this.

First, we extend Lemma C.2 for this assumption. First, observe that any vector $\mathbf{u} \in \text{Null}(\mathbf{K_{DD}})$, also satisfies $\mathbf{u} \in \text{Null}\left(\nabla_{\boldsymbol{\theta}_D}^2 \mathbb{E}_{p_{\boldsymbol{\theta_G}}}[D_{\boldsymbol{\theta_D}}^2(x)]\big|_{\boldsymbol{\theta_D}=\boldsymbol{\theta_D^\star}}\right)$ for all $\boldsymbol{\theta_G} \in B_\epsilon(\boldsymbol{\theta_G^\star})$ by the second condition in Assumption IV. Then for any $\boldsymbol{\theta_D} = \boldsymbol{\theta_D^\star} + \epsilon \mathbf{u}$, $D_{\boldsymbol{\theta_D}}(x) = 0$ for all $x$ in the support of $p_{\boldsymbol{\theta_G}}$ where $\boldsymbol{\theta_G} \in B_\epsilon(\boldsymbol{\theta_G^\star})$. Then, we can show that any perturbation of the discriminator within the null space of $\mathbf{K}_{DD}$ is an 'equilibrium discriminator' which when paired with any generator in small neighborhood around $\boldsymbol{\theta_G^\star}$, results in zero updates on the generator. To prove this, recall that $\dot{\boldsymbol{\theta}}_{\mathbf{G}}$ consists of two terms integrated over $\mathcal{X}$, $D_{\boldsymbol{\theta_D}}(x)$ and $\nabla_{\boldsymbol{\theta_G}} p_{\boldsymbol{\theta_G}}(x)$. In our previous proof under the original version of Assumption IV, we used an intricate fact about these two terms. In particular, we said that for a generator within a radius of $\epsilon_G/2$ from equilibrium (where $\epsilon_G$ is as defined in the original version of Assumption IV), i) the support of $p_{\boldsymbol{\theta_G}}$ is the same as $p_{\text{data}}$ and therefore $D_{\boldsymbol{\theta_D}}(x) = 0$ for all $x$ in the true support and ii) for all $x$ not in the true support, and for any generator $\boldsymbol{\theta_G} \in B_{\epsilon_G/2}(\boldsymbol{\theta_G^\star})$, $\nabla_{\boldsymbol{\theta_G}} p_{\boldsymbol{\theta_G}}(x) = 0$.

In this case, we only have a weaker guarantee that for a generator within a perturbation of $\epsilon_G/2$ from $\boldsymbol{\theta_G^\star}$, the support is contained in the combined support $\bigcup_{\boldsymbol{\theta_G} \in B_{\epsilon_G}(\boldsymbol{\theta_G^\star})} \text{supp}(p_{\boldsymbol{\theta_G}})$. But then, i) for all $x$ in the combined support we have that $D_{\boldsymbol{\theta_D}}(x) = 0$ and ii) for all $x$ not in the combined support and for any generator $\boldsymbol{\theta_G} \in B_{\epsilon_G/2}(\boldsymbol{\theta_G^\star})$, $\nabla_{\boldsymbol{\theta_G}} p_{\boldsymbol{\theta_G}}(x) = 0$. Then, the generator updates are:

$$\dot{\boldsymbol{\theta}}_{\mathbf{G}} = -\int_{\mathcal{X}} f(-\underbrace{D_{\boldsymbol{\theta_D}}(x)}_{\substack{0 \text{ inside} \\ \text{combined supp}}}) \underbrace{\nabla_{\boldsymbol{\theta_G}} p_{\boldsymbol{\theta_G}}(x)}_{\substack{0 \text{ outside} \\ \text{combined supp}}} dx = -f(0) \nabla_{\boldsymbol{\theta_G}} \int_{\bigcup_{\boldsymbol{\theta_G} \in B_{\epsilon_G}(\boldsymbol{\theta_G^\star})} \text{supp}(p_{\boldsymbol{\theta_G}})} p_{\boldsymbol{\theta_G}}(x) dx$$

$$= -f(0) \nabla_{\boldsymbol{\theta_G}} 1 = 0$$

The second part of Lemma C.2 holds similarly.

We need to make a similar argument to prove that the generator's Hessian $\mathbf{J}_{GG} = 0$ at equilibrium.

$$\mathbf{v}^T \left.\frac{\partial \dot{\boldsymbol{\theta}}_{\mathbf{G}}}{\partial \boldsymbol{\theta}_{\mathbf{G}}}\right|_{\boldsymbol{\theta}_{\mathbf{D}}=\boldsymbol{\theta}_{\mathbf{D}}^\star,\boldsymbol{\theta}_{\mathbf{G}}=\boldsymbol{\theta}_{\mathbf{G}}^\star} = \mathbf{v}^T \left.\frac{\partial \dot{\boldsymbol{\theta}}_{\mathbf{G}}\big|_{\boldsymbol{\theta}_{\mathbf{D}}=\mathbf{v}^T\boldsymbol{\theta}_{\mathbf{D}}^\star}}{\partial \boldsymbol{\theta}_{\mathbf{G}}}\right|_{\boldsymbol{\theta}_{\mathbf{G}}=\boldsymbol{\theta}_{\mathbf{G}}^\star} = -\lim_{\substack{\boldsymbol{\theta}_{\mathbf{G}}-\boldsymbol{\theta}_{\mathbf{G}}^\star=\epsilon\mathbf{v}\\\epsilon\to 0}} \frac{\int_{\mathcal{X}} f(-\overbrace{D_{\boldsymbol{\theta}_{\mathbf{D}}^\star}(x)}^{\substack{0\text{ inside}\\\text{combined supp}}})\,\overbrace{\nabla_{\boldsymbol{\theta}_{\mathbf{G}}}^T p_{\boldsymbol{\theta}_{\mathbf{G}}}(x)}^{\substack{0\text{ outside}\\\text{combined supp}}}\,dx}{\epsilon}$$

$$= -f(0)\lim_{\substack{\boldsymbol{\theta}_{\mathbf{G}}-\boldsymbol{\theta}_{\mathbf{G}}^\star=\epsilon\mathbf{v}\\\epsilon\to 0}} \frac{\int_{\bigcup_{\boldsymbol{\theta}_{\mathbf{G}}\in B_{\epsilon_G}(\boldsymbol{\theta}_{\mathbf{G}}^\star)}\text{supp}(p_{\boldsymbol{\theta}_{\mathbf{G}}})} \nabla_{\boldsymbol{\theta}_{\mathbf{G}}}^T p_{\boldsymbol{\theta}_{\mathbf{G}}}(x)dx}{\epsilon}$$

$$= -f(0)\lim_{\substack{\boldsymbol{\theta}_{\mathbf{G}}-\boldsymbol{\theta}_{\mathbf{G}}^\star=\epsilon\mathbf{v}\\\epsilon\to 0}} \frac{\nabla_{\boldsymbol{\theta}_{\mathbf{G}}}^T \int_{\bigcup_{\boldsymbol{\theta}_{\mathbf{G}}\in B_{\epsilon_G}(\boldsymbol{\theta}_{\mathbf{G}}^\star)}\text{supp}(p_{\boldsymbol{\theta}_{\mathbf{G}}})} p_{\boldsymbol{\theta}_{\mathbf{G}}}(x)dx}{\epsilon}$$

$$= -f(0)\lim_{\substack{\boldsymbol{\theta}_{\mathbf{G}}-\boldsymbol{\theta}_{\mathbf{G}}^\star=\epsilon\mathbf{v}\\\epsilon\to 0}} \frac{\nabla_{\boldsymbol{\theta}_{\mathbf{G}}}^T 1}{\epsilon} = 0$$

A similar modification of the proof can be done for Lemma C.3 where we show that $\mathbf{T}_D \mathbf{K}_{DG} \mathbf{T}_G^T$ has the same column rank as $\mathbf{K}_{DG} \mathbf{T}_G^T$. The rest of the proof follows as it did. $\square$

### C.2 The non-realizable case

In this section, we extend our results about local stability of GANs to the case in which the true distribution can not be represented by any generator in the generator space. While this is a hard problem in general, we consider a specific case in which the discriminator is linear in its parameters and show that the system is locally stable at any equilibrium and its surrounding equilibria (none of which may correspond to the true distribution). More formally, consider a discriminator of the form:

$$D_{\boldsymbol{\theta}_D}(x) = \boldsymbol{\theta}_D^T \boldsymbol{\phi}(x)$$

where $\boldsymbol{\phi}$ is any feature mapping. For example, $\boldsymbol{\phi}(x)$ could be a polynomial basis or the representation learned by a neural network (which we assume is not trained during the updates near equilibrium). Thus, the objective in this case is:

$$V(D_{\boldsymbol{\theta}_{\mathbf{D}}}, G_{\boldsymbol{\theta}_{\mathbf{G}}}) = \mathbb{E}_{p_{\text{data}}}[f(\boldsymbol{\theta}_D^T \boldsymbol{\phi}(x))] + \mathbb{E}_{p_{\boldsymbol{\theta}_{\mathbf{G}}}}[f(\boldsymbol{\theta}_D^T \boldsymbol{\phi}(x))]$$

We consider a generator space that does not necessarily contain the true distribution, but however contains a generator $\boldsymbol{\theta}_{\mathbf{G}}^\star$ that is an equilibrium point when paired with a discriminator that is zero on the support of the true data and the generated data. It must be noted that $\boldsymbol{\theta}_{\mathbf{D}}^\star = \mathbf{0}$ is not necessarily the only equilibrium discriminator. Especially, if $\boldsymbol{\phi}$ lies in a lower dimensional manifold, there could be a subspace of all-zero discriminators. Now, for such a generator to exist, we need:

$$\nabla_{\boldsymbol{\theta}_{\mathbf{D}}} V(D_{\boldsymbol{\theta}_{\mathbf{D}}}, G_{\boldsymbol{\theta}_{\mathbf{G}}})|_{(\boldsymbol{\theta}_{\mathbf{D}}^\star, \boldsymbol{\theta}_{\mathbf{G}}^\star)} = 0$$
$$\implies \mathbb{E}_{p_{\text{data}}}[\boldsymbol{\phi}(\mathbf{x})] = \mathbb{E}_{p_{\boldsymbol{\theta}_{\mathbf{G}}^\star}}[\boldsymbol{\phi}(\mathbf{x})]$$

In other words, we want the means of the generated distribution and the true distribution in the representation $\boldsymbol{\phi}$ to be identical. For a given generator space, this essentially is a restriction on the representation $\boldsymbol{\phi}$ that has been learned/chosen for the discriminator. If $\boldsymbol{\phi}$ was a richer representation that computes many higher order moments of the data, we may never find an equilibrium generator.

We now prove Theorem 3.1 for the non-realizable case. Our main idea is identical to that of the proof in the realizable case. However, we need to be careful in a number of steps. We first prove a result similar to Lemma C.1 that derives the Jacobian of the system at equilibrium.

**Lemma C.4.** *For the dynamical system defined by the GAN objective in Equation 2 and the updates in Equation 3, the Jacobian at an equilibrium point $(\boldsymbol{\theta}_{\mathbf{D}}^\star, \boldsymbol{\theta}_{\mathbf{G}}^\star)$, under the Assumptions I (for the non-realizable case) and IV is:*

$$\mathbf{J} = \begin{bmatrix} \mathbf{J}_{DD} & \mathbf{J}_{DG} \\ -\mathbf{J}_{DG}^T & \mathbf{J}_{GG} \end{bmatrix} = \begin{bmatrix} 2f''(0)\mathbf{K}_{DD} & f'(0)\mathbf{K}_{DG} \\ -f'(0)\mathbf{K}_{DG}^T & 0 \end{bmatrix}$$

*where*

$$2\mathbf{K}_{DD} \triangleq \mathbb{E}_{p_{\text{data}}}[(\nabla_{\boldsymbol{\theta}_{\mathbf{D}}}D_{\boldsymbol{\theta}_{\mathbf{D}}}(x))(\nabla_{\boldsymbol{\theta}_{\mathbf{D}}}D_{\boldsymbol{\theta}_{\mathbf{D}}}(x))^T] + \mathbb{E}_{p_{\boldsymbol{\theta}_{\mathbf{G}}^\star}}[(\nabla_{\boldsymbol{\theta}_{\mathbf{D}}}D_{\boldsymbol{\theta}_{\mathbf{D}}}(x))(\nabla_{\boldsymbol{\theta}_{\mathbf{D}}}D_{\boldsymbol{\theta}_{\mathbf{D}}}(x))^T]\Big|_{\boldsymbol{\theta}_{\mathbf{D}}^\star} \succeq 0$$

*and*

$$\mathbf{K}_{DG} \triangleq \int_{\mathcal{X}} \nabla_{\boldsymbol{\theta}_{\mathbf{D}}}D_{\boldsymbol{\theta}_{\mathbf{D}}}(x)\nabla_{\boldsymbol{\theta}_{\mathbf{G}}}^T p_{\boldsymbol{\theta}_{\mathbf{G}}}(x)dx\Big|_{\boldsymbol{\theta}_{\mathbf{D}}=\boldsymbol{\theta}_{\mathbf{D}}^\star, \boldsymbol{\theta}_{\mathbf{G}}=\boldsymbol{\theta}_{\mathbf{G}}^\star}$$

*Proof.* Recall that,

$$V(D_{\boldsymbol{\theta}_{\mathbf{D}}}, G_{\boldsymbol{\theta}_{\mathbf{G}}}) = \mathbb{E}_{p_{data}}[f(D_{\boldsymbol{\theta}_{\mathbf{D}}}(x))] + \mathbb{E}_{p_{\boldsymbol{\theta}_{\mathbf{G}}}}[f(-D_{\boldsymbol{\theta}_{\mathbf{D}}}(x))]$$

$$\dot{\boldsymbol{\theta}}_{\mathbf{D}} = \mathbb{E}_{p_{data}}[f'(D_{\boldsymbol{\theta}_{\mathbf{D}}}(x))\nabla_{\boldsymbol{\theta}_{\mathbf{D}}}D_{\boldsymbol{\theta}_{\mathbf{D}}}(x)] - \mathbb{E}_{p_{\boldsymbol{\theta}_{\mathbf{G}}}}[f'(-D_{\boldsymbol{\theta}_{\mathbf{D}}}(x))\nabla_{\boldsymbol{\theta}_{\mathbf{D}}}D_{\boldsymbol{\theta}_{\mathbf{D}}}(x)]$$

$$\dot{\boldsymbol{\theta}}_{\mathbf{G}} = -\int_{\mathcal{X}} \nabla_{\boldsymbol{\theta}_{\mathbf{G}}}^T p_{\boldsymbol{\theta}_{\mathbf{G}}} f(-D_{\boldsymbol{\theta}_{\mathbf{D}}}(x))dx$$

First we show that $\mathbf{J}_{DD}$ has a similar form which is still negative semi-definite when $f''(0) < 0$:

$$\mathbf{J}_{DD} = \nabla_{\boldsymbol{\theta}_{\mathbf{D}}}^2 V(G_{\boldsymbol{\theta}_{\mathbf{G}}}, D_{\boldsymbol{\theta}_{\mathbf{D}}})\big|_{(\boldsymbol{\theta}_{\mathbf{D}}^\star, \boldsymbol{\theta}_{\mathbf{G}}^\star)} = \frac{\partial \dot{\boldsymbol{\theta}}_{\mathbf{D}}}{\partial \boldsymbol{\theta}_{\mathbf{D}}}\Bigg|_{\boldsymbol{\theta}_{\mathbf{D}}=\boldsymbol{\theta}_{\mathbf{D}}^\star, \boldsymbol{\theta}_{\mathbf{G}}=\boldsymbol{\theta}_{\mathbf{G}}^\star} = \frac{\partial \dot{\boldsymbol{\theta}}_{\mathbf{D}}\big|_{\boldsymbol{\theta}_{\mathbf{G}}=\boldsymbol{\theta}_{\mathbf{G}}^\star}}{\partial \boldsymbol{\theta}_{\mathbf{D}}}\Bigg|_{\boldsymbol{\theta}_{\mathbf{D}}=\boldsymbol{\theta}_{\mathbf{D}}^\star}$$

$$= \frac{\partial \left(\mathbb{E}_{p_{data}}[f'(D_{\boldsymbol{\theta}_{\mathbf{D}}}(x))\nabla_{\boldsymbol{\theta}_{\mathbf{D}}}D_{\boldsymbol{\theta}_{\mathbf{D}}}(x)] - \mathbb{E}_{p_{\boldsymbol{\theta}_{\mathbf{G}}^\star}}[f'(-D_{\boldsymbol{\theta}_{\mathbf{D}}}(x))\nabla_{\boldsymbol{\theta}_{\mathbf{D}}}D_{\boldsymbol{\theta}_{\mathbf{D}}}(x)]\right)}{\partial \boldsymbol{\theta}_{\mathbf{D}}}\Bigg|_{\boldsymbol{\theta}_{\mathbf{D}}=\boldsymbol{\theta}_{\mathbf{D}}^\star}$$

$$= \left(\mathbb{E}_{p_{data}}\left[f''(D_{\boldsymbol{\theta}_{\mathbf{D}}}(x))\nabla_{\boldsymbol{\theta}_{\mathbf{D}}}D_{\boldsymbol{\theta}_{\mathbf{D}}}(x)\nabla_{\boldsymbol{\theta}_{\mathbf{D}}}^T D_{\boldsymbol{\theta}_{\mathbf{D}}}(x)\right] + \mathbb{E}_{p_{data}}\left[f'(D_{\boldsymbol{\theta}_{\mathbf{D}}}(x))\nabla_{\boldsymbol{\theta}_{\mathbf{D}}}^2 D_{\boldsymbol{\theta}_{\mathbf{D}}}(x)\right]\right)\big|_{\boldsymbol{\theta}_{\mathbf{D}}=\boldsymbol{\theta}_{\mathbf{D}}^\star}$$

$$+ \left(\mathbb{E}_{p_{\boldsymbol{\theta}_{\mathbf{G}}^\star}}\left[f''(-D_{\boldsymbol{\theta}_{\mathbf{D}}}(x))\nabla_{\boldsymbol{\theta}_{\mathbf{D}}}D_{\boldsymbol{\theta}_{\mathbf{D}}}(x)\nabla_{\boldsymbol{\theta}_{\mathbf{D}}}^T D_{\boldsymbol{\theta}_{\mathbf{D}}}(x)\right] - \mathbb{E}_{p_{\boldsymbol{\theta}_{\mathbf{G}}^\star}}\left[f'(-D_{\boldsymbol{\theta}_{\mathbf{D}}}(x))\nabla_{\boldsymbol{\theta}_{\mathbf{D}}}^2 D_{\boldsymbol{\theta}_{\mathbf{D}}}(x)\right]\right)\big|_{\boldsymbol{\theta}_{\mathbf{D}}=\boldsymbol{\theta}_{\mathbf{D}}^\star}$$

$$= \left(\mathbb{E}_{p_{\text{data}}}\left[f''(0)\nabla_{\boldsymbol{\theta}_{\mathbf{D}}}D_{\boldsymbol{\theta}_{\mathbf{D}}}(x)\nabla_{\boldsymbol{\theta}_{\mathbf{D}}}^T D_{\boldsymbol{\theta}_{\mathbf{D}}}(x)\right] + \mathbb{E}_{p_{\text{data}}}\left[f'(0)\underbrace{\nabla_{\boldsymbol{\theta}_{\mathbf{D}}}^2 D_{\boldsymbol{\theta}_{\mathbf{D}}}(x)}_{=0}\right]\right)\Bigg|_{\boldsymbol{\theta}_{\mathbf{D}}=\boldsymbol{\theta}_{\mathbf{D}}^\star}$$

$$+ \left(\mathbb{E}_{p_{\boldsymbol{\theta}_{\mathbf{G}}^\star}}\left[f''(0)\nabla_{\boldsymbol{\theta}_{\mathbf{D}}}D_{\boldsymbol{\theta}_{\mathbf{D}}}(x)\nabla_{\boldsymbol{\theta}_{\mathbf{D}}}^T D_{\boldsymbol{\theta}_{\mathbf{D}}}(x)\right] - \mathbb{E}_{p_{\boldsymbol{\theta}_{\mathbf{G}}^\star}}\left[f'(0)\underbrace{\nabla_{\boldsymbol{\theta}_{\mathbf{D}}}^2 D_{\boldsymbol{\theta}_{\mathbf{D}}}(x)}_{=0}\right]\right)\Bigg|_{\boldsymbol{\theta}_{\mathbf{D}}=\boldsymbol{\theta}_{\mathbf{D}}^\star}$$

$$= f''(0)\left(\mathbb{E}_{p_{\text{data}}}\left[\nabla_{\boldsymbol{\theta}_{\mathbf{D}}}D_{\boldsymbol{\theta}_{\mathbf{D}}}(x)\nabla_{\boldsymbol{\theta}_{\mathbf{D}}}^T D_{\boldsymbol{\theta}_{\mathbf{D}}}(x)\right] + \mathbb{E}_{p_{\boldsymbol{\theta}_{\mathbf{D}}^\star}}\left[\nabla_{\boldsymbol{\theta}_{\mathbf{D}}}D_{\boldsymbol{\theta}_{\mathbf{D}}}(x)\nabla_{\boldsymbol{\theta}_{\mathbf{D}}}^T D_{\boldsymbol{\theta}_{\mathbf{D}}}(x)\right]\right)\big|_{\boldsymbol{\theta}_{\mathbf{D}}=\boldsymbol{\theta}_{\mathbf{D}}^\star}$$

$$= 2f''(0)\mathbf{K}_{DD}$$

The most crucial step here is that we were able to ignore the terms corresponding to $\nabla_{\boldsymbol{\theta}_{\mathbf{D}}}^2 D_{\boldsymbol{\theta}_{\mathbf{D}}}(x)$ because the discriminator is linear in its parameters i.e., $\nabla_{\boldsymbol{\theta}_{\mathbf{D}}}D_{\boldsymbol{\theta}_{\mathbf{D}}}(x) = \boldsymbol{\phi}(x)$ and thus the Hessian is zero.

All other terms in the Jacobian are identical to the realizable case because we assume that at equilibrium the discriminator must be identically zero.

$\square$

Now, we again show that the equilibrium point in consideration lies in a subspace of equilibria.

**Lemma C.5.** *Under Assumptions I (Non-realizable), III, and IV there exists $\epsilon_D, \epsilon_G > 0$ such that for all $\epsilon'_D \leq \epsilon_D$ and $\epsilon'_G \leq \epsilon_G$, and for any unit vectors $\mathbf{u} \in \mathsf{Null}(\mathbf{K}_{DD})$, $\mathbf{v} \in \mathsf{Null}(\mathbf{K}_{DG})$, $(\boldsymbol{\theta}^\star_{\mathbf{D}} + \epsilon'_D\mathbf{u}, \boldsymbol{\theta}^\star_{\mathbf{G}} + \epsilon'_G\mathbf{v})$ is an equilibrium point.*

*Proof.* Our proof is only slightly different from that of Lemma C.2. Note that $4\mathbf{K}_{DD}$ is the Hessian of the function $\mathbb{E}_{p_{\text{data}}}[D^2_{\boldsymbol{\theta}_{\mathbf{D}}}(x)] + \mathbb{E}_{p_{\boldsymbol{\theta}^\star_{\mathbf{G}}}}[D^2_{\boldsymbol{\theta}_{\mathbf{D}}}(x)]$ at equilibrium.

Since this is the sum of two positive semi-definite matrices, any vector in the null space of $\mathsf{Null}(\mathbf{K}_{DD})$ is also in the null space of the Hessian of $\mathbb{E}_{p_{\text{data}}}[D^2_{\boldsymbol{\theta}_{\mathbf{D}}}(x)]$. Then, by Assumption III, $\mathbb{E}_{p_{\text{data}}}[D^2_{\boldsymbol{\theta}_{\mathbf{D}}}(x)]$ is locally constant along any unit vector $\mathbf{u} \in \mathsf{Null}(\mathbf{K}_{DD})$. That is, for sufficiently small $\epsilon$, if $\boldsymbol{\theta}_{\mathbf{D}} = \boldsymbol{\theta}^\star_{\mathbf{D}} + \epsilon\mathbf{u}$, $\mathbb{E}_{p_{\text{data}}}[D^2_{\boldsymbol{\theta}_{\mathbf{D}}}(x)]$ equals the value of the function at equilibrium, which is 0 because $D_{\boldsymbol{\theta}^\star_D}(x) = 0$ (according to Assumption I). Thus, we can conclude that for all $x$ in the support of $p_{\text{data}}$, $D_{\boldsymbol{\theta}_{\mathbf{D}}}(x) = 0$. Now from Assumption IV, the support of generators in a small neighborhood is identical to the support of the true distribution, therefore these discriminators are equilibrium discriminators i.e., when paired with any generator, the generator updates are zero.

Similarly, $2\mathbf{K}^T_{DG}\mathbf{K}_{DG}$ is the Hessian of the function $\left\| \mathbb{E}_{p_{\text{data}}}[\nabla_{\boldsymbol{\theta}_{\mathbf{D}}}D_{\boldsymbol{\theta}_{\mathbf{D}}}(x)] - \mathbb{E}_{p_{\boldsymbol{\theta}_{\mathbf{G}}}}[\nabla_{\boldsymbol{\theta}_{\mathbf{D}}}D_{\boldsymbol{\theta}_{\mathbf{D}}}(x)] \right\|^2$ at equilibrium. Then, by Assumption III, $\left\| \mathbb{E}_{p_{\text{data}}}[\nabla_{\boldsymbol{\theta}_{\mathbf{D}}}D_{\boldsymbol{\theta}_{\mathbf{D}}}(x)] - \mathbb{E}_{p_{\boldsymbol{\theta}_{\mathbf{G}}}}[\nabla_{\boldsymbol{\theta}_{\mathbf{D}}}D_{\boldsymbol{\theta}_{\mathbf{D}}}(x)] \right\|^2$ is locally constant along any unit vector $\mathbf{v} \in \mathsf{Null}(\mathbf{K}_{DG})$. That is, for sufficiently small $\epsilon'$, if $\boldsymbol{\theta}_{\mathbf{G}} = \boldsymbol{\theta}^\star_{\mathbf{G}} + \epsilon'\mathbf{v}$, $\left\| \mathbb{E}_{p_{\text{data}}}[\nabla_{\boldsymbol{\theta}_{\mathbf{D}}}D_{\boldsymbol{\theta}_{\mathbf{D}}}(x)] - \mathbb{E}_{p_{\boldsymbol{\theta}_{\mathbf{G}}}}[\nabla_{\boldsymbol{\theta}_{\mathbf{D}}}D_{\boldsymbol{\theta}_{\mathbf{D}}}(x)] \right\|^2$ equals the value of the function at equilibrium. Now, since this function is proportional to the magnitude of the equilibrium discriminator's update, it equals zero at equilibrium. Now, observe that

$$\mathbb{E}_{p_{\text{data}}}[\nabla_{\boldsymbol{\theta}_{\mathbf{D}}}D_{\boldsymbol{\theta}_{\mathbf{D}}}(x)] - \mathbb{E}_{p_{\boldsymbol{\theta}_{\mathbf{G}}}}[\nabla_{\boldsymbol{\theta}_{\mathbf{D}}}D_{\boldsymbol{\theta}_{\mathbf{D}}}(x)] = \mathbb{E}_{p_{\text{data}}}[\boldsymbol{\phi}(x)] - \mathbb{E}_{p_{\boldsymbol{\theta}_{\mathbf{G}}}}[\boldsymbol{\phi}(x)]$$

is independent of the discriminator variables (Here, we have used the fact that the discriminator is linear in its parameters.) . This means that for these generators along $\mathbf{v}$, the discriminator update must be zero. In other words, these generators are equilibrium generators in the non-realizable sense, that their $\boldsymbol{\phi}$ representation matches with the true distribution.

In summary, for all slight perturbations along $\mathbf{u} \in \mathsf{Null}(\mathbf{K}_{DD})$, $\mathbf{v} \in \mathsf{Null}(\mathbf{K}_{DG})$ we have established that the discriminator and generator individually satisfy the requirements of an equilibrium discriminator and generator pair, and therefore the system is itself is in equilibrium for these perturbations. $\square$

It turns out that given these two lemmas, Lemma C.3 follows as it did earlier, and therefore the main theorem follows too.

## D  Linear Quadratic GAN – Gaussian example

In order to illustrate our assumptions in Theorem 3.1, consider a simple GAN that learns an $n$-dimensional Gaussian distribution $\mathcal{N}(\boldsymbol{\mu}, \boldsymbol{\Sigma})$, where $\boldsymbol{\Sigma} \succ 0$. Let the latent variable be drawn from the standard normal, $\mathcal{N}(\mathbf{0}, \mathbf{I}_n)$. Consider a quadratic discriminator $D(\mathbf{x}) = \mathbf{x}^T\mathbf{W}_2\mathbf{x} + \mathbf{w}_1^T\mathbf{x}$, and a linear generator $G(z) = \mathbf{A}z + \mathbf{b}$. We call the resulting system LQ (linear-quadratic). Let $\boldsymbol{\Sigma}^{1/2}$ be the unique real positive definite matrix such that $\left( \boldsymbol{\Sigma}^{1/2} \right)^2 = \boldsymbol{\Sigma}$. Then we have the following:

**Theorem D.1.** *In LQ, $\mathbf{A} = \boldsymbol{\Sigma}^{1/2}, \mathbf{b} = \boldsymbol{\mu}$ and $\mathbf{W}_2 = 0, \mathbf{w}_1 = 0$ corresponds to an equilibrium that is locally exponentially stable provided $f''(0) < 0$ and $f'(0) \neq 0$.*

*Proof.* Since the system consists of parameters arranged in the form of matrices, we will need vectorization calculus [Magnus et al., 1995] to arrange these parameters as a vector and differentiate them/with respect to them.

To verify that the given point is indeed an equilibrium, let us look at the GAN objective:

$$\begin{aligned} V(G, D) &= \mathbb{E}_{\mathbf{x}\sim\mathcal{N}(\boldsymbol{\mu},\boldsymbol{\Sigma})}[f(\mathbf{x}^T\mathbf{W}_2\mathbf{x} + \mathbf{w}_1^T\mathbf{x})] \\ &\quad + \mathbb{E}_{\mathbf{z}\sim\mathcal{N}(\mathbf{0},\mathbf{I}_n)}[f(-(\mathbf{A}\mathbf{z} + \mathbf{b})^T\mathbf{W}_2(\mathbf{A}\mathbf{z} + \mathbf{b}) - \mathbf{w}_1^T(\mathbf{A}\mathbf{z} + \mathbf{b}))] \end{aligned}$$

The updates in Equation 3 for LQ can be written as :

$$\dot{\mathbf{W}}_2 = \mathbb{E}_{\mathbf{x}\sim\mathcal{N}(\boldsymbol{\mu},\boldsymbol{\Sigma})}[\mathbf{x}\mathbf{x}^T f'(\mathbf{x}^T\mathbf{W}_2\mathbf{x} + \mathbf{w}_1^T\mathbf{x})]$$
$$- \mathbb{E}_{\mathbf{z}\sim\mathcal{N}(\mathbf{0},\mathbf{I}_n)}[(\mathbf{A}\mathbf{z}+\mathbf{b})(\mathbf{A}\mathbf{z}+\mathbf{b})^T f'(-(\mathbf{A}\mathbf{z}+\mathbf{b})^T\mathbf{W}_2(\mathbf{A}\mathbf{z}+\mathbf{b}) - \mathbf{w}_1^T(\mathbf{A}\mathbf{z}+\mathbf{b}))]$$

$$\dot{\mathbf{w}}_1 = \mathbb{E}_{\mathbf{x}\sim\mathcal{N}(\boldsymbol{\mu},\boldsymbol{\Sigma})}[\mathbf{x} f'(\mathbf{x}^T\mathbf{W}_2\mathbf{x} + \mathbf{w}_1^T\mathbf{x})]$$
$$- \mathbb{E}_{\mathbf{z}\sim\mathcal{N}(\mathbf{0},\mathbf{I}_n)}[(\mathbf{A}\mathbf{z}+\mathbf{b}) f'(-(\mathbf{A}\mathbf{z}+\mathbf{b})^T\mathbf{W}_2(\mathbf{A}\mathbf{z}+\mathbf{b}) - \mathbf{w}_1^T(\mathbf{A}\mathbf{z}+\mathbf{b}))]$$

$$\dot{\mathbf{A}} = \mathbb{E}_{\mathbf{z}\sim\mathcal{N}(\mathbf{0},\mathbf{I}_n)}[((\mathbf{W}_2 + \mathbf{W}_2^T)\mathbf{A}\mathbf{z}\mathbf{z}^T + (\mathbf{W}_2 + \mathbf{W}_2^T)\mathbf{b}\mathbf{z}^T + \mathbf{w}_1\mathbf{z}^T)$$
$$f'(-(\mathbf{A}\mathbf{z}+\mathbf{b})^T\mathbf{W}_2(\mathbf{A}\mathbf{z}+\mathbf{b}) - \mathbf{w}_1^T(\mathbf{A}\mathbf{z}+\mathbf{b}))]$$

$$\dot{\mathbf{b}} = \mathbb{E}_{\mathbf{z}\sim\mathcal{N}(\mathbf{0},\mathbf{I}_n)}[((\mathbf{W}_2 + \mathbf{W}_2^T)\mathbf{A}\mathbf{z} + (\mathbf{W}_2 + \mathbf{W}_2^T)\mathbf{b} + \mathbf{w}_1)$$
$$f'(-(\mathbf{A}\mathbf{z}+\mathbf{b})^T\mathbf{W}_2(\mathbf{A}\mathbf{z}+\mathbf{b}) - \mathbf{w}_1^T(\mathbf{A}\mathbf{z}+\mathbf{b}))]$$

Clearly, when $\mathbf{z} \sim \mathcal{N}(0, I)$, we have that $\boldsymbol{\Sigma}^{1/2}\mathbf{z} + \boldsymbol{\mu} \sim \mathcal{N}(\boldsymbol{\mu}, \boldsymbol{\Sigma})$, therefore at $\mathbf{A} = \boldsymbol{\Sigma}^{1/2}, \mathbf{b} = \boldsymbol{\mu}$ and $\mathbf{W}_2 = 0, \mathbf{w}_1 = 0$, all the above updates become zero, implying that it is an equilibrium for which the generator has converged to the true distribution. To prove that it is locally stable, we need to examine the Jacobian at that point. Note that since the Jacobian is a matrix with one cell for each pair of discriminator-generator parameters, we need to calculate second-order derivatives after vectorizing the parameter matrices $\mathbf{Q}$ and $\mathbf{A}$.

We first calculate the derivative of the discriminator updates with respect to the discriminator itself.

$$\left.\frac{\partial vec(\dot{\mathbf{W}}_2)}{\partial vec(\mathbf{W}_2)}\right|_{\substack{\mathbf{b}=\boldsymbol{\mu},\mathbf{W}_2=0,\\ \mathbf{w}_1=0,\mathbf{A}=\boldsymbol{\Sigma}^{1/2}}} = \left.\frac{\partial}{\partial vec(\mathbf{W}_2)}\left(\left. vec(\dot{\mathbf{W}}_2)\right|_{\mathbf{b}=\boldsymbol{\mu},\mathbf{A}=\boldsymbol{\Sigma}^{1/2},\mathbf{w}_1=0}\right)\right|_{\mathbf{W}_2=0}$$

$$= \left.\frac{\partial}{\partial vec(\mathbf{W}_2)}\mathbb{E}_{\mathbf{x}\sim\mathcal{N}(\boldsymbol{\mu},\boldsymbol{\Sigma})}[vec(\mathbf{x}\mathbf{x}^T)(f'(\mathbf{x}^T\mathbf{W}_2\mathbf{x}) - f'(-\mathbf{x}^T\mathbf{W}_2\mathbf{x}))]\right|_{\mathbf{W}_2=0}$$

$$= 2f''(0)\mathbb{E}_{\mathbf{x}\sim\mathcal{N}(\boldsymbol{\mu},\boldsymbol{\Sigma})}[(\mathbf{x}\otimes\mathbf{x})(\mathbf{x}\otimes\mathbf{x})^T]$$

$$\left.\frac{\partial vec(\dot{\mathbf{W}}_2)}{\partial \mathbf{w}_1}\right|_{\substack{\mathbf{b}=\boldsymbol{\mu},\mathbf{W}_2=0,\\ \mathbf{w}_1=0,\mathbf{A}=\boldsymbol{\Sigma}^{1/2}}} = \left.\frac{\partial}{\partial \mathbf{w}_1}\left(\left. vec(\dot{\mathbf{W}}_2)\right|_{\mathbf{b}=\boldsymbol{\mu},\mathbf{A}=\boldsymbol{\Sigma}^{1/2},\mathbf{W}_2=0}\right)\right|_{\mathbf{w}_1=0}$$

$$= \left.\frac{\partial}{\partial \mathbf{w}_1}\mathbb{E}_{\mathbf{x}\sim\mathcal{N}(\boldsymbol{\mu},\boldsymbol{\Sigma})}[vec(\mathbf{x}\mathbf{x}^T)(f'(\mathbf{w}_1^T\mathbf{x}) - f'(-\mathbf{w}_1^T\mathbf{x}))]\right|_{\mathbf{w}_1=0}$$

$$= 2f''(0)\mathbb{E}_{\mathbf{x}\sim\mathcal{N}(\boldsymbol{\mu},\boldsymbol{\Sigma})}[(\mathbf{x}\otimes\mathbf{x})\mathbf{x}^T]$$

$$\left.\frac{\partial \dot{\mathbf{w}}_1}{\partial \mathbf{w}_1}\right|_{\substack{\mathbf{b}=\boldsymbol{\mu},\mathbf{W}_2=0,\\ \mathbf{w}_1=0,\mathbf{A}=\boldsymbol{\Sigma}^{1/2}}} = \left.\frac{\partial}{\partial \mathbf{w}_1}\left(\left. \dot{\mathbf{w}}_1\right|_{\mathbf{b}=\boldsymbol{\mu},\mathbf{A}=\boldsymbol{\Sigma}^{1/2},\mathbf{W}_2=0}\right)\right|_{\mathbf{w}_1=0}$$

$$= \left.\frac{\partial}{\partial \mathbf{w}_1}\mathbb{E}_{\mathbf{x}\sim\mathcal{N}(\boldsymbol{\mu},\boldsymbol{\Sigma})}[\mathbf{x}(f'(\mathbf{w}_1^T\mathbf{x}) - f'(-\mathbf{w}_1^T\mathbf{x}))]\right|_{\mathbf{w}_1=0}$$

$$= 2f''(0)\mathbb{E}_{\mathbf{x}\sim\mathcal{N}(\boldsymbol{\mu},\boldsymbol{\Sigma})}[\mathbf{x}\mathbf{x}^T]$$

Then we calculate the derivative of the discriminator updates with respect to the generator parameters. Note that we will be using the constant matrix $\mathbf{T}_{n,n}$ which is a matrix of zeros and ones defined in vectorization algebra; this matrix is the vectorization equivalent of the transpose operator. That is, for any square matrix $\mathbf{V} \in \mathbb{R}^n$, $\mathbf{T}_{n,n}vec(\mathbf{V}) = vec(\mathbf{V}^T)$.

$$\left.\frac{\partial vec(\dot{\mathbf{W}}_2)}{\partial vec(\mathbf{A})}\right|_{\substack{\mathbf{b}=\boldsymbol{\mu},\mathbf{W}_2=0,\\ \mathbf{w}_1=0,\mathbf{A}=\boldsymbol{\Sigma}^{1/2}}} = \left.\frac{\partial}{\partial vec(\mathbf{A})}\left(\left. vec(\dot{\mathbf{W}}_2)\right|_{\mathbf{b}=\boldsymbol{\mu},\mathbf{W}_2=0,\mathbf{w}_1=0}\right)\right|_{\mathbf{A}=\boldsymbol{\Sigma}^{1/2}}$$

$$= -\frac{\partial}{\partial vec(\mathbf{A})} vec\left(\mathbb{E}_{\mathbf{z}\sim\mathcal{N}(\mathbf{0},\mathbf{I}_n)}[(\mathbf{A}\mathbf{z}+\boldsymbol{\mu})(\mathbf{A}\mathbf{z}+\boldsymbol{\mu})^T f'(0)]\right)\Big|_{\mathbf{A}=\boldsymbol{\Sigma}^{1/2}}$$

$$= -\frac{\partial}{\partial vec(\mathbf{A})} vec\left(\mathbb{E}_{\mathbf{z}\sim\mathcal{N}(\mathbf{0},\mathbf{I}_n)}[(\mathbf{A}\mathbf{z}\mathbf{z}^T\mathbf{A}^T + \mathbf{A}\mathbf{z}\boldsymbol{\mu}^T + \boldsymbol{\mu}\mathbf{z}^T\mathbf{A}^T)f'(0)]\right)\Big|_{\mathbf{A}=\boldsymbol{\Sigma}^{1/2}}$$

$$= -\frac{\partial}{\partial vec(\mathbf{A})} vec(\mathbf{A}\mathbf{A}^T)f'(0)\Big|_{\mathbf{A}=\boldsymbol{\Sigma}^{1/2}} = -(\mathbf{I}_{n^2} + \mathbf{T}_{n,n})(\boldsymbol{\Sigma}^{1/2} \times \mathbf{I}_n)f'(0)$$

$$\frac{\partial vec(\dot{\mathbf{W}}_2)}{\partial \mathbf{b}}\Big|_{\substack{\mathbf{b}=\boldsymbol{\mu},\mathbf{W}_2=0, \\ \mathbf{w}_1=0,\mathbf{A}=\boldsymbol{\Sigma}^{1/2}}} = \frac{\partial}{\partial \mathbf{b}}\left(vec(\dot{\mathbf{W}}_2)\Big|_{\mathbf{w}_1=0,\mathbf{W}_2=0,\mathbf{A}=\boldsymbol{\Sigma}^{1/2}}\right)\Big|_{\mathbf{b}=\boldsymbol{\mu}}$$

$$= -\frac{\partial}{\partial \mathbf{b}} vec\left(\mathbb{E}_{\mathbf{z}\sim\mathcal{N}(\mathbf{0},\mathbf{I}_n)}[(\boldsymbol{\Sigma}^{1/2}\mathbf{z}+\mathbf{b})(\boldsymbol{\Sigma}^{1/2}\mathbf{z}+\mathbf{b})^T f'(0)]\right)$$

$$= -f'(0)\frac{\partial}{\partial \mathbf{b}} vec(\mathbf{b}\mathbf{b}^T)\Big|_{\mathbf{b}=\boldsymbol{\mu}}$$

$$= -f'(0)(\boldsymbol{\mu} \otimes \mathbf{I}_n + \mathbf{I}_n \otimes \boldsymbol{\mu})$$

$$\frac{\partial \dot{\mathbf{w}}_1}{\partial vec(\mathbf{A})} = \frac{\partial}{\partial vec(\mathbf{A})}\left(\dot{\mathbf{w}}_1|_{\mathbf{w}_1=0,\mathbf{W}_2=0,\mathbf{b}=\boldsymbol{\mu}}\right)\Big|_{\mathbf{A}=\boldsymbol{\Sigma}^{1/2}}$$

$$= -\frac{\partial}{\partial vec(\mathbf{A})}\mathbb{E}_{\mathbf{z}\sim\mathcal{N}(\mathbf{0},\mathbf{I}_n)}[(\mathbf{A}\mathbf{z}+\mathbf{b})f'(0)] = 0$$

$$\frac{\partial \dot{\mathbf{w}}_1}{\partial \mathbf{b}} = \frac{\partial}{\partial \mathbf{b}}\left(\dot{\mathbf{w}}_1|_{\mathbf{w}_1=0,\mathbf{W}_2=0,\mathbf{A}=\boldsymbol{\Sigma}^{1/2}}\right)\Big|_{b=\boldsymbol{\mu}} = -\frac{\partial}{\partial \mathbf{b}}\mathbb{E}_{\mathbf{z}\sim\mathcal{N}(\mathbf{0},\mathbf{I}_n)}[(\boldsymbol{\Sigma}^{1/2}\mathbf{z}+\mathbf{b})f'(0)]$$

$$= -\mathbf{I}f'(0)$$

Recall that the Jacobian can then be written as:

$$\begin{bmatrix} \mathbf{J}_{DD} & \mathbf{J}_{DG} \\ -\mathbf{J}_{DG}^T & 0 \end{bmatrix}$$

where

$$\mathbf{J}_{DD} = \begin{bmatrix} \frac{\partial vec(\dot{\mathbf{W}}_2)}{\partial vec(\mathbf{W}_2)}\Big|_{\text{eqbm}} & \frac{\partial vec(\dot{\mathbf{W}}_2)}{\partial \mathbf{w}_1}\Big|_{\text{eqbm}} \\ \frac{\partial \dot{\mathbf{w}}_1}{\partial vec(\mathbf{W}_2)}\Big|_{\text{eqbm}} & \frac{\partial \dot{\mathbf{w}}_1}{\partial \mathbf{w}_1}\Big|_{\text{eqbm}} \end{bmatrix} =$$

$$= \begin{bmatrix} \mathbb{E}_{\mathbf{x}\sim\mathcal{N}(\boldsymbol{\mu},\boldsymbol{\Sigma})}[(\mathbf{x} \otimes \mathbf{x})(\mathbf{x} \otimes \mathbf{x})^T] & \mathbb{E}_{\mathbf{x}\sim\mathcal{N}(\boldsymbol{\mu},\boldsymbol{\Sigma})}[(\mathbf{x} \otimes \mathbf{x})\mathbf{x}^T] \\ \left(\mathbb{E}_{\mathbf{x}\sim\mathcal{N}(\boldsymbol{\mu},\boldsymbol{\Sigma})}[(\mathbf{x} \otimes \mathbf{x})\mathbf{x}^T]\right)^T & \mathbb{E}_{\mathbf{x}\sim\mathcal{N}(\boldsymbol{\mu},\boldsymbol{\Sigma})}[\mathbf{x}\mathbf{x}^T] \end{bmatrix} 2f''(0)$$

and

$$\mathbf{J}_{DG} = \begin{bmatrix} \frac{\partial vec(\dot{\mathbf{W}}_2)}{\partial vec(\mathbf{A})}\Big|_{\text{eqbm}} & \frac{\partial vec(\dot{\mathbf{W}}_2)}{\partial \mathbf{b}}\Big|_{\text{eqbm}} \\ \frac{\partial \dot{\mathbf{w}}_1}{\partial vec(\mathbf{A})}\Big|_{\text{eqbm}} & \frac{\partial \dot{\mathbf{w}}_1}{\partial \mathbf{b}}\Big|_{\text{eqbm}} \end{bmatrix} =$$

$$= -\begin{bmatrix} (\mathbf{I}_{n^2} + \mathbf{T}_{n,n})(\boldsymbol{\Sigma}^{1/2} \otimes \mathbf{I}_n) & \boldsymbol{\mu} \otimes \mathbf{I}_n + \mathbf{I}_n \otimes \boldsymbol{\mu} \\ 0 & \mathbf{I}_n \end{bmatrix} f'(0)$$

We can show that $\mathbf{J}_{DD}$ is negative definite because it is a moment matrix with a negative multiplicative factor. This is proved in Theorem D.2. Recall that as long as $f''(0) < 0$, $f'(0) \neq 0$ and $\mathbf{J}_{DG}$ is full column rank (in this case full rank because $\mathbf{J}_{DG}$ is a square matrix), the matrix has eigenvalues whose real components are strictly negative.

To show that $\mathbf{J}_{DG}$ is full column rank, first observe that the last few columns corresponding to $\mathbf{b}$ are linearly independent because, if $\mathbf{y}$ belongs to its null space, then

$$\begin{bmatrix} \boldsymbol{\mu} \otimes \mathbf{I}_n + \mathbf{I}_n \otimes \boldsymbol{\mu} \\ \mathbf{I} \end{bmatrix} \mathbf{y} = \begin{bmatrix} (\boldsymbol{\mu} \otimes \mathbf{I}_n + \mathbf{I}_n \otimes \boldsymbol{\mu})\mathbf{y} \\ \mathbf{y} \end{bmatrix} = 0,$$

which implies that $\mathbf{y} = 0$.

To verify whether the first few columns corresponding to $\mathbf{A}$ are linearly independent or not, consider any $\mathbf{V} \neq 0$. Then, we want to verify whether the following term is always non-zero or not:

$$(\mathbf{I}_{n^2} + \mathbf{T}_{n,n})(\boldsymbol{\Sigma}^{1/2} \otimes \mathbf{I}_n)vec(\mathbf{V}) = (\mathbf{I}_{n^2} + \mathbf{T}_{n,n})vec(\mathbf{I}_n \mathbf{V}(\boldsymbol{\Sigma}^{1/2})^T)$$
$$= vec(\mathbf{V}(\boldsymbol{\Sigma}^{1/2})^T + \boldsymbol{\Sigma}^{1/2}\mathbf{V}^T),$$

which is equivalent to testing whether $\mathbf{V}(\boldsymbol{\Sigma}^{1/2})^T + \boldsymbol{\Sigma}^{1/2}\mathbf{V}^T$ is non-zero.

Now, we will show that if $\mathbf{V}(\boldsymbol{\Sigma}^{1/2})^T + \boldsymbol{\Sigma}^{1/2}\mathbf{V}^T = 0$, then $\mathbf{V} = 0$. Recall that $\boldsymbol{\Sigma}^{1/2} = \mathbf{U}\boldsymbol{\Lambda}^{1/2}\mathbf{U}^T$. Then,

$$\mathbf{V}(\boldsymbol{\Sigma}^{1/2})^T = -\boldsymbol{\Sigma}^{1/2}\mathbf{V}^T$$
$$\implies \mathbf{V}\mathbf{U}\boldsymbol{\Lambda}^{1/2}\mathbf{U}^T = -\mathbf{U}\boldsymbol{\Lambda}^{1/2}\mathbf{U}^T\mathbf{V}^T$$
$$\mathbf{U}^T\mathbf{V}\mathbf{U}\boldsymbol{\Lambda}^{1/2}\mathbf{U}^T\mathbf{V}\mathbf{U} = -\boldsymbol{\Lambda}^{1/2}\mathbf{U}^T\mathbf{V}^T\mathbf{V}\mathbf{U}$$

Observe that the left hand side is positive semi-definite while the right hand side is negative semi-definite. Therefore these terms must be equal to zero, which would then imply that $\mathbf{V}^T\mathbf{V} = 0$ i.e., $\mathbf{V} = 0$. Thus the Jacobian is indeed Hurwitz.

In summary, this means that Assumption III holds trivially because there are no zero eigenvalues for the matrices involved in the Jacobian. This further means that there are no other equilibria in a small neighborhood around the considered equilibrium. Therefore, Assumption I is also satisfied. Finally, since the support of the distribution is $\mathbb{R}^n$, Assumption IV is also trivially satisfied. Thus, if Assumption II holds, the system is exponentially stable.

$\square$

We now prove that $\mathbf{J}_{DD}$ is negative definite.

**Theorem D.2.** *The matrix*
$$\begin{bmatrix} \mathbb{E}_{\mathbf{x}\sim\mathcal{N}(\boldsymbol{\mu},\Sigma)}[(\mathbf{x} \otimes \mathbf{x})(\mathbf{x} \otimes \mathbf{x})^T] & \mathbb{E}_{\mathbf{x}\sim\mathcal{N}(\boldsymbol{\mu},\Sigma)}[(\mathbf{x} \otimes \mathbf{x})\mathbf{x}^T] \\ \left(\mathbb{E}_{\mathbf{x}\sim\mathcal{N}(\boldsymbol{\mu},\Sigma)}[(\mathbf{x} \otimes \mathbf{x})\mathbf{x}^T]\right)^T & \mathbb{E}_{\mathbf{x}\sim\mathcal{N}(\boldsymbol{\mu},\Sigma)}[\mathbf{x}\mathbf{x}^T] \end{bmatrix}$$
*is positive definite.*

*Proof.* Let $\mathbf{U}$ be any arbitrary matrix and $\mathbf{v}$ be an arbitrary vector. Then,

$$\begin{bmatrix} vec(\mathbf{U}) \\ \mathbf{v} \end{bmatrix}^T \begin{bmatrix} \mathbb{E}_{\mathbf{x}\sim\mathcal{N}(\boldsymbol{\mu},\Sigma)}[(\mathbf{x} \otimes \mathbf{x})(\mathbf{x} \otimes \mathbf{x})^T] & \mathbb{E}_{\mathbf{x}\sim\mathcal{N}(\boldsymbol{\mu},\Sigma)}[(\mathbf{x} \otimes \mathbf{x})\mathbf{x}^T] \\ \left(\mathbb{E}_{\mathbf{x}\sim\mathcal{N}(\boldsymbol{\mu},\Sigma)}[(\mathbf{x} \otimes \mathbf{x})\mathbf{x}^T]\right)^T & \mathbb{E}_{\mathbf{x}\sim\mathcal{N}(\boldsymbol{\mu},\Sigma)}[\mathbf{x}\mathbf{x}^T] \end{bmatrix} \begin{bmatrix} vec(\mathbf{U}) \\ \mathbf{v} \end{bmatrix} =$$

$$= \mathbb{E}_{\mathbf{x}\sim\mathcal{N}(\boldsymbol{\mu},\Sigma)} \left[ \left\| \begin{bmatrix} \mathbf{x} \otimes \mathbf{x} \\ \mathbf{x} \end{bmatrix}^T \begin{bmatrix} vec(\mathbf{U}) \\ \mathbf{v} \end{bmatrix} \right\|^2 \right]$$

$$= \mathbb{E}_{\mathbf{x}\sim\mathcal{N}(\boldsymbol{\mu},\Sigma)} \left[ \left( \mathbf{x}^T\mathbf{U}\mathbf{x} + \mathbf{x}^T\mathbf{v} \right)^2 \right]$$

Now, $\left( \mathbf{x}^T\mathbf{U}\mathbf{x} + \mathbf{x}^T\mathbf{v} \right)^2 = 0$ forms a quadric $n-1$-dimensional hypersurface in $n$ dimensions, and therefore is of measure zero. For all other points, $\left( \mathbf{x}^T\mathbf{U}\mathbf{x} + \mathbf{x}^T\mathbf{v} \right)^2 > 0$ and therefore the above expectation is strictly positive.

$\square$

# E WGANs are not necessarily asymptotically stable

We consider a specific case of the LQ WGAN that learns a zero mean gaussian distribution, and show that there exists points near certain equilibria such that if the system is initialized to that point, it will periodically come back to that initial point rather than converge to the equilibrium.

**Theorem E.1.** *The* LQ *WGAN system for learning a zero mean Gaussian distribution* $\mathcal{N}(\mathbf{0}, \boldsymbol{\Sigma})$ *($\boldsymbol{\Sigma} \succ 0$) is not asymptotically stable at the equilibrium corresponding to* $\mathbf{A} = \boldsymbol{\Sigma}^{1/2}, \mathbf{b} = \mathbf{0}$ *and* $\mathbf{W}_2 = 0, \mathbf{w}_1 = 0$.

*Proof.* In order to show that the system is not asymptotically stable, we show that there are initializations of the system that are arbitrarily close to the equilibrium such that the system goes orbits around the equilibrium forever. For simplicity, we first prove this for the one-dimensional gaussian $\mathcal{N}(0, \sigma)$ and later extend it to the multi-dimensional case. Let the quadratic discriminator be $D(x) = w_2^2 x + w_1 x$ and the linear generator be $az + b$. Then the WGAN objective in Equation 2 for the LQ system is:

$$V(G, D) = \mathbb{E}_{x \sim \mathcal{N}(0,\sigma)}[w_2 x^2 + w_1 x] - \mathbb{E}_{\mathbf{z} \sim \mathcal{N}(0,1)}[w_2(az + b)^2 + w_1(az + b)]$$
$$= w_2(\sigma^2) - w_2(a^2 + b^2) - w_1 b$$

The updates in Equation 3 for LQ simplify as follows:

$$\dot{w}_2 = \sigma^2 - a^2 - b^2$$
$$\dot{w}_1 = -b$$
$$\dot{a} = 2w_2 a$$
$$\dot{b} = 2w_2 b + w_1$$

The system has two equilibria, $w_2 = 0, w_1 = 0, a = \pm\sigma, b = 0$. We will assume that the system is initialized with $w_1 = b = 0$, which means that the system will forever have $w_1 = b = 0$ because the respective updates are zero too. Hence, we only need to focus on the variables $w_2$ and $a$.

Now, it can be shown that if $a$ is initialized to $a_0 \geq 0$, $a$ never becomes negative (and similarly for $a \leq 0$). Therefore, we will focus on the equilibrium where $a = \sigma$, and assuming $a \geq 0$ examine how the distance from the equilibrium $w_2^2 + (a - \sigma)^2$ changes with time. The rate of change of this quantity is given by $2(w_2 \dot{w}_2 + (a - \sigma)\dot{a}) = 2w_2(a - \sigma)^2$. Observe that when $w_2 > 0$, this term is non-negative i.e., the system never gets closer to the equilibrium. Thus, when the system is in the "bad" half-space $w_2 > 0$, the only hope for it to converge is to exit this half-space so that $w_2$ becomes negative. However, we show that there exists initializations that are close to the equilibrium such that even if it does exit the bad half-space it eventually re-enters it, going in a perpetual loop.

More specifically, let $(w_2(t), a(t))$ denote the system at time $t$. Let the initialization satisfy $w_2(0) = 0$ and $a(0) \in (0, \sigma)$. We will now analyze the trajectory of this system. First note that $\dot{w}_2(0) > 0$, which means the system enters the bad half-space after immediately $t > 0$. Thus, if the system had to converge to the considered equilibrium, it would have to reach $w_2 = 0$ again at some time $T$. First observe that at this time $a(T) > \sigma$ because we need $\dot{w}_2(T) < 0$ at this time. (In fact we can say that $a(T) - \sigma \geq \sigma - a(0)$ because we know that the radius never decreased until time $T$.) Now, we claim that the system simply retraces back its path along $a$ and reaches $a(0)$ at time $2T$. More clearly, we claim that the system at time $T + t$ can be described in terms of what it was at time $T - t$ as $(w_2(T + t), a(T + t)) = (-w_2(T - t), a(T - t))$.

To prove this observe that this statement is true for $t = 0$ because $w_2(T) = 0$. Then we only need to show that at any $t$, if $(w_2(T + t), a(T + t)) = (-w_2(T - t), a(T - t))$, then $\dot{w}_2(T + t) = \dot{w}_2(T - t)$ and $\dot{a}(T + t) = -\dot{a}(T - t)$. This is indeed true because $\dot{w}_2(T + t) = \sigma^2 - a^2(T + t) = \sigma^2 - a^2(T - t) = \dot{w}_2(T - t)$ and $\dot{a}_2(T + t) = 2w_2(T + t)a(T + t) = 2(-w_2(T - t))a(T - t) = -\dot{a}(T - t)$. Therefore, applying $t = T$, we get $(w_2(2T), a(2T)) = (-w_2(0), a(0)) = (0, a(0))$ i.e., the system has looped back to its original state by following its old path mirrored across the line $w_2 = 0$. Since this holds for initializations that are arbitrarily close to the equilibrium (i.e., $a(0)$ can be arbitrarily close to $\sigma$), the system is not asymptotically stable.

We extend this argument to the higher dimensional case as follows. Again, we initialize the system so that $\mathbf{w}_1 = \mathbf{0}$ and $\mathbf{b} = \mathbf{0}$, then we can only focus on the updates on $\mathbf{W}_2$ and $\mathbf{A}$:

$$\dot{\mathbf{W}}_2 = \boldsymbol{\Sigma} - \mathbf{A}\mathbf{A}^T$$
$$\dot{\mathbf{A}} = (\mathbf{W}_2 + \mathbf{W}_2^T)\mathbf{A}$$

As before, we initialize $\mathbf{W}_2 = 0$. We will also consider a more sophisticated initialization compared to $a \in (0, \sigma)$. Since $\boldsymbol{\Sigma}$ is positive definite, let $\boldsymbol{\Sigma} = \mathbf{U}\boldsymbol{\Lambda}\mathbf{U}^T$. We initialize $\mathbf{A} = \mathbf{U}\boldsymbol{\Lambda}_A(0)\mathbf{U}^T$ such that $\boldsymbol{\Lambda}_A(0)$ has at least one diagonal element that is positive but strictly less than the corresponding diagonal element in $\boldsymbol{\Lambda}^{1/2}$ (where $\boldsymbol{\Lambda}^{1/2} \succ 0$).

Now, we first establish that all the updates and the variables in the system remain in the eigenspace defined by $\mathbf{U}$. That is, at any point in time $t$, the variables can be expressed as $\mathbf{W}_2(t) = \mathbf{U}\boldsymbol{\Lambda}_W(t)\mathbf{U}^T$ and $\mathbf{A}(t) = \mathbf{U}\boldsymbol{\Lambda}_A(t)\mathbf{U}^T$ for some real diagonal matrices $\boldsymbol{\Lambda}_W(t)$ and $\boldsymbol{\Lambda}_A(t)$. Clearly, this is true for time $t = 0$. Assuming this is true for arbitrary time $t$, observe that the updates are

$$\dot{\mathbf{W}}_2(t) = \mathbf{U}(\boldsymbol{\Lambda} - \boldsymbol{\Lambda}_A^2(t))\mathbf{U}^T$$
$$\dot{\mathbf{A}}(t) = 2\mathbf{U}\boldsymbol{\Lambda}_W\mathbf{U}^T\mathbf{U}\boldsymbol{\Lambda}_A\mathbf{U}^T = 2\mathbf{U}\boldsymbol{\Lambda}_W\boldsymbol{\Lambda}_A\mathbf{U}^T$$

Thus this is true for any time $t$. Therefore, we can analyze the system in terms of $\boldsymbol{\Lambda}_A, \boldsymbol{\Lambda}_W$ and the constant $\boldsymbol{\Lambda}$ as though there are $n$ independent 1-dimensional Gaussian systems. Then, the orbiting systems from the 1-dimensional updates must manifest here too. More specifically, these cycles would correspond to the diagonal in $\boldsymbol{\Lambda}_A$ which was initialized to be less than $\boldsymbol{\Lambda}^{1/2}$. $\qquad\square$

# F   Gradient-based regularization

In Section F.1, we prove how our gradient-based regularizer stabilizes the both the GAN and the WGAN system. Besides this property, in Section F.2 we provide an alternative mathematical intuition that is based on arg-max differentiation, to motivate our regularization term. Finally, in Section F.3, we discuss how our regularizer addresses mode collapse and 1-unrolled GAN updates.

## F.1   Local stability of gradient-regularized GANs

We first restate our main result below.

**Theorem 3.2.** *The dynamical system defined by the GAN objective in Equation 2 and the updates in Equation 4, is locally exponentially stable at the equilibrium, under the same conditions as in Theorem 3.1, if $\eta < \frac{1}{2\lambda_{\max}(-\mathbf{J}_{DD})}$. Further, under appropriate conditions similar to these, the WGAN system is locally exponentially stable at the equilibrium for any $\eta$. The rate of convergence for the WGAN is governed only by the eigenvalues $\lambda$ of the Jacobian at equilibrium with a strict negative real part upper bounded as:*

- *If $\mathrm{Im}(\lambda) = 0$, then $\mathrm{Re}(\lambda) \leq -\frac{2f'^2(0)\eta\lambda_{\min}^{(+)}(\mathbf{K}_{DG}^T\mathbf{K}_{DG})}{4f'^2(0)\eta^2\lambda_{\max}(\mathbf{K}_{DG}^T\mathbf{K}_{DG})+1}$*

- *If $\mathrm{Im}(\lambda) \neq 0$, then $\mathrm{Re}(\lambda) \leq -\eta f'^2(0)\lambda_{\min}^{(+)}(\mathbf{K}_{DG}^T\mathbf{K}_{DG})$.*

To prove this result, we first present the Jacobian of the system at equilibrium in the presence of the gradient penalty. Recall that the penalty basically adds an extra $-\nabla_{\boldsymbol{\theta}_\mathbf{G}}\|\nabla_{\boldsymbol{\theta}_D}V(D_{\boldsymbol{\theta}_\mathbf{D}}, G_{\boldsymbol{\theta}_\mathbf{G}})\|^2$ to the generator's update.

**Lemma F.1.** *For the dynamical system defined by the GAN objective in Equation 2 and the updates in Equation 4, the Jacobian at an equilibrium point $(\boldsymbol{\theta}_\mathbf{D}^\star, \boldsymbol{\theta}_\mathbf{G}^\star)$, under the Assumptions I and IV is:*

$$\mathbf{J} = \begin{bmatrix} \mathbf{J}_{DD} & \mathbf{J}_{DG} \\ -\mathbf{J}_{DG}^T(\mathbf{I} + 2\eta\mathbf{J}_{DD}) & -2\eta\mathbf{J}_{DG}^T\mathbf{J}_{DG} \end{bmatrix}$$

*where $\mathbf{J}_{DD}$ and $\mathbf{J}_{DG}$ are terms in the Jacobian corresponding to the original updates, as described in Theorem 3.1.*

*Proof.* Note that the only change to the Jacobian would be in the rows corresponding to the generator parameters. Therefore, we will focus only on the additional terms in these rows.

The additional term added to $-\mathbf{J}_{DG}^T$ is:

$$
-\left.\frac{\partial \eta \nabla_{\boldsymbol{\theta}_{\mathbf{G}}}\|\nabla_{\boldsymbol{\theta}_D} V(D_{\boldsymbol{\theta}_{\mathbf{D}}},G_{\boldsymbol{\theta}_{\mathbf{G}}})\|^2}{\partial \boldsymbol{\theta}_{\mathbf{D}}}\right|_{\boldsymbol{\theta}_{\mathbf{D}}^\star,\boldsymbol{\theta}_{\mathbf{G}}^\star} = -\eta\left(\left.\frac{\partial \nabla_{\boldsymbol{\theta}_{\mathbf{D}}}\|\nabla_{\boldsymbol{\theta}_D} V(D_{\boldsymbol{\theta}_{\mathbf{D}}},G_{\boldsymbol{\theta}_{\mathbf{G}}})\|^2}{\partial \boldsymbol{\theta}_{\mathbf{G}}}\right|_{\boldsymbol{\theta}_{\mathbf{D}}^\star,\boldsymbol{\theta}_{\mathbf{G}}^\star}\right)^T
$$

$$
= -\eta\left(\left.\frac{\partial\left(2\nabla_{\boldsymbol{\theta}_{\mathbf{D}}}^2 V(D_{\boldsymbol{\theta}_{\mathbf{D}}},G_{\boldsymbol{\theta}_{\mathbf{G}}})\nabla_{\boldsymbol{\theta}_{\mathbf{D}}} V(D_{\boldsymbol{\theta}_{\mathbf{D}}},G_{\boldsymbol{\theta}_{\mathbf{G}}})\right)}{\partial \boldsymbol{\theta}_{\mathbf{G}}}\right)^T\right|_{\boldsymbol{\theta}_{\mathbf{D}}^\star,\boldsymbol{\theta}_{\mathbf{G}}^\star}
$$

$$
= -2\eta\left(\frac{\partial\nabla_{\boldsymbol{\theta}_{\mathbf{D}}}^2 V(D_{\boldsymbol{\theta}_{\mathbf{D}}},G_{\boldsymbol{\theta}_{\mathbf{G}}})}{\partial \boldsymbol{\theta}_{\mathbf{G}}}\underbrace{\nabla_{\boldsymbol{\theta}_{\mathbf{D}}} V(D_{\boldsymbol{\theta}_{\mathbf{D}}},G_{\boldsymbol{\theta}_{\mathbf{G}}})}_{\text{0 at eqbm}} + \frac{\partial\nabla_{\boldsymbol{\theta}_{\mathbf{G}}} V(D_{\boldsymbol{\theta}_{\mathbf{D}}},G_{\boldsymbol{\theta}_{\mathbf{G}}})}{\partial \boldsymbol{\theta}_{\mathbf{D}}}\nabla_{\boldsymbol{\theta}_{\mathbf{D}}}^2 V(D_{\boldsymbol{\theta}_{\mathbf{D}}},G_{\boldsymbol{\theta}_{\mathbf{G}}})\right)^T\Bigg|_{\boldsymbol{\theta}_{\mathbf{D}}^\star,\boldsymbol{\theta}_{\mathbf{G}}^\star}
$$

$$
= -2\eta\mathbf{J}_{DG}^T\mathbf{J}_{DD}
$$

Now, the additional term added to $\mathbf{J}_{GG}$ is:

$$
-\left.\frac{\partial \eta \nabla_{\boldsymbol{\theta}_{\mathbf{G}}}\|\nabla_{\boldsymbol{\theta}_D} V(D_{\boldsymbol{\theta}_{\mathbf{D}}},G_{\boldsymbol{\theta}_{\mathbf{G}}})\|^2}{\partial \boldsymbol{\theta}_{\mathbf{G}}}\right|_{\boldsymbol{\theta}_{\mathbf{D}}^\star,\boldsymbol{\theta}_{\mathbf{G}}^\star}
$$

$$
= -\eta\left.\frac{\partial}{\partial \boldsymbol{\theta}_{\mathbf{G}}}\left(2\frac{\partial\nabla_{\boldsymbol{\theta}_{\mathbf{G}}} V(D_{\boldsymbol{\theta}_{\mathbf{D}}},G_{\boldsymbol{\theta}_{\mathbf{G}}})}{\partial \boldsymbol{\theta}_{\mathbf{D}}}\nabla_{\boldsymbol{\theta}_{\mathbf{D}}} V(D_{\boldsymbol{\theta}_{\mathbf{D}}},G_{\boldsymbol{\theta}_{\mathbf{G}}})\right)\right|_{\boldsymbol{\theta}_{\mathbf{D}}^\star,\boldsymbol{\theta}_{\mathbf{G}}^\star}
$$

$$
= -2\eta\left(\frac{\partial\nabla_{\boldsymbol{\theta}_{\mathbf{G}}}^2 V(D_{\boldsymbol{\theta}_{\mathbf{D}}},G_{\boldsymbol{\theta}_{\mathbf{G}}})}{\partial \boldsymbol{\theta}_{\mathbf{D}}}\underbrace{\nabla_{\boldsymbol{\theta}_{\mathbf{D}}} V(D_{\boldsymbol{\theta}_{\mathbf{D}}},G_{\boldsymbol{\theta}_{\mathbf{G}}})}_{\text{0 at eqbm}} + \left(\frac{\partial\nabla_{\theta_G} V(D_{\boldsymbol{\theta}_{\mathbf{D}}},G_{\boldsymbol{\theta}_{\mathbf{G}}})}{\partial \boldsymbol{\theta}_{\mathbf{D}}}\right)^T\frac{\partial\nabla_{\theta_G} V(D_{\boldsymbol{\theta}_{\mathbf{D}}},G_{\boldsymbol{\theta}_{\mathbf{G}}})}{\partial \boldsymbol{\theta}_{\mathbf{D}}}\right)\Bigg|_{\boldsymbol{\theta}_{\mathbf{D}}^\star,\boldsymbol{\theta}_{\mathbf{G}}^\star}
$$

$$
= -2\eta\mathbf{J}_{DG}^T\mathbf{J}_{DG}
$$

$\square$

Now, we will prove stability of the regularized system for conventional GANs. Observe that Lemmas C.2 regarding the subspace of equilibria holds in this case too. Again, we can project the system as follows:

**Lemma F.2.** *For the dynamical system defined by the GAN objective in Equation 2 and the updates in Equation 4, consider the eigenvalue decompositions $\mathbf{K}_{DD} = \mathbf{U_D}\mathbf{\Lambda_D}\mathbf{U_D}^T$ and $\mathbf{K}_{DG}^T\mathbf{K}_{DG} = \mathbf{U_G}\mathbf{\Lambda_G}\mathbf{U_G}^T$. Let $\mathbf{U_D} = [\mathbf{T}_D^T, \mathbf{T}_D'^T]$ and $\mathbf{U_G} = [\mathbf{T}_G^T, \mathbf{T}_G'^T]$ such that $\mathsf{Col}(\mathbf{T}_D'^T) = \mathsf{Null}(\mathbf{K}_{DD})$ and $\mathsf{Col}(\mathbf{T}_G'^T) = \mathsf{Null}(\mathbf{K}_{DG})$. Consider the projections, $\boldsymbol{\gamma_D} = \mathbf{T}_D\boldsymbol{\theta}_D$ and $\boldsymbol{\gamma_G} = \mathbf{T}_G\boldsymbol{\theta}_G$. Then, the block in the Jacobian at equilibrium that corresponds to the projected system has the form:*

$$
\mathbf{J}' = \begin{bmatrix} \mathbf{J}'_{DD} & \mathbf{J}'_{DG} \\ -\mathbf{J}'^T_{DG} & \mathbf{J}'_{GG} \end{bmatrix} = \begin{bmatrix} \mathbf{T}_D\mathbf{J}_{DD}\mathbf{T}_D^T & \mathbf{T}_D\mathbf{J}_{DG}\mathbf{T}_G^T \\ -\mathbf{T}_G\mathbf{J}_{DG}^T(\mathbf{I}+2\eta\mathbf{J}_{DD})\mathbf{T}_D^T & -2\eta\mathbf{T}_G\mathbf{J}_{DG}^T\mathbf{J}_{DG}\mathbf{T}_G^T \end{bmatrix}
$$

*Under Assumption II, we have that $\mathbf{J}'_{DD} \prec 0$ and $\mathbf{J}'_{DG}$ is full column rank and $\mathbf{J}'_{GG} \prec 0$.*

It is straightforward to extend the proof of Lemma C.3 to prove this lemma. Now, recall from Theorem A.4 that if we show $\mathbf{J}'$ is Hurwitz the original system is exponentially stable. In the non-regularized system, we showed this by making use of the structure of the matrix. For this system, we will design a quadratic Lyapunov function that strictly decreases at non-equilibria points.

**Lemma F.3.** *For the dynamical system defined by the GAN objective in Equation 2 and the updates in Equation 4, if $\eta < \frac{1}{2\lambda_{\max}(-\mathbf{J_{DD}})}$ the linearization of the system projected to a subspace orthogonal to the subspace of equilibria is exponentially stable with the Lyapunov function $\mathbf{x}^T\mathbf{P}\mathbf{x}$ where,*

$$
\mathbf{P} = \begin{bmatrix} \mathbf{T}_D(\mathbf{I}+2\eta\mathbf{J}_{DD})\mathbf{T}_D^T & 0 \\ 0 & \mathbf{I} \end{bmatrix}
$$

and $\mathbf{x}^T$ is $[\boldsymbol{\gamma_D}^T \boldsymbol{\gamma_G}^T] - [\boldsymbol{\gamma_D^\star}^T \boldsymbol{\gamma_G^\star}^T]$. *The function strictly decreases with time except at the equilibrium* $[\boldsymbol{\gamma_D^\star}^T \boldsymbol{\gamma_G^\star}^T]^T$.

*Proof.* Note that when $\eta < \frac{1}{2\lambda_{\max}(-\mathbf{J_{DD}})}$, $\mathbf{P} = \mathbf{P}^T \succ 0$ therefore the Lyapunov function is indeed positive definite. Furthermore, note that the rate of decrease is given by $\mathbf{x}^T \mathbf{Q} \mathbf{x}$ where $\mathbf{Q} = (\mathbf{J'}^T \mathbf{P} + \mathbf{P} \mathbf{J'})$. To show that this is strictly decreasing, we only need to show that $\mathbf{J'}^T \mathbf{P} + \mathbf{P} \mathbf{J'} \prec 0$. First of all, note that $\mathbf{Q} =$

$$
\begin{bmatrix} \mathbf{T}_D \\ \mathbf{T}_G \end{bmatrix} \left( \begin{bmatrix} \mathbf{J}_{DD}(\mathbf{I} + 2\eta\mathbf{J}_{DD}) & -(\mathbf{I} + 2\eta\mathbf{J}_{DD})\mathbf{J}_{DG} \\ \mathbf{J}_{DG}^T \mathbf{T}_D^T \mathbf{T}_D (\mathbf{I} + 2\eta\mathbf{J}_{DD}) & -2\eta\mathbf{J}_{DG}^T \mathbf{J}_{DG} \end{bmatrix} + \right.
$$
$$
\left. \begin{bmatrix} (\mathbf{I} + 2\eta\mathbf{J}_{DD})\mathbf{J}_{DD} & (\mathbf{I} + 2\eta\mathbf{J}_{DD})\mathbf{T}_D^T \mathbf{T}_D \mathbf{J}_{DG} \\ -\mathbf{J}_{DG}^T(\mathbf{I} + 2\eta\mathbf{J}_{DD}) & -2\eta\mathbf{J}_{DG}^T \mathbf{J}_{DG} \end{bmatrix} \right) \begin{bmatrix} \mathbf{T}_D \\ \mathbf{T}_G \end{bmatrix}^T
$$

Here, the off-diagonal terms are $\mathbf{T}_D(\mathbf{I} + 2\eta\mathbf{J}_{DD})(\mathbf{I} - \mathbf{T}_D^T \mathbf{T}_D)\mathbf{J}_{DG}\mathbf{T}_G^T$ and its negative. This can be equated to zero because,

$$
\begin{aligned}
\mathbf{T}_D(\mathbf{I} + 2\eta\mathbf{J}_{DD})(\mathbf{I} - \mathbf{T}_D^T \mathbf{T}_D) &= \mathbf{T}_D - \overbrace{\mathbf{T}_D\mathbf{T}_D^T}^{\mathbf{I}}\mathbf{T}_D + 2\eta\mathbf{T}_D\mathbf{J}_{DD} - 2\eta\mathbf{T}_D\mathbf{J}_{DD}\mathbf{T}_D^T \mathbf{T}_D \\
&= 2\eta(\mathbf{T}_D\mathbf{J}_{DD} - \mathbf{T}_D\mathbf{J}_{DD}\mathbf{T}_D^T \mathbf{T}_D) \\
&= 2\eta(\mathbf{T}_D\mathbf{T}_D^T \boldsymbol{\Lambda}_D \mathbf{T}_D - \mathbf{T}_D\mathbf{T}_D^T \boldsymbol{\Lambda}_D \mathbf{T}_D\mathbf{T}_D^T \mathbf{T}_D) \\
&= 2\eta(\boldsymbol{\Lambda}_D \mathbf{T}_D - \boldsymbol{\Lambda}_D \mathbf{T}_D\mathbf{T}_D^T \mathbf{T}_D) = 0
\end{aligned}
$$

Then, the above matrix is equal to the diagonal matrix:

$$
\begin{bmatrix} \mathbf{T_D}\left[\mathbf{J}_{DD}(\mathbf{I} + 2\eta\mathbf{J}_{DD}) + (\mathbf{I} + 2\eta\mathbf{J}_{DD})\mathbf{J}_{DD}\right]\mathbf{T_D}^T & 0 \\ 0 & -4\eta\mathbf{T}_G\mathbf{J}_{DG}^T\mathbf{J}_{DG}\mathbf{T}_G^T \end{bmatrix}
$$

Note that by our choice of $\eta$, $(\mathbf{I} + 2\eta\mathbf{J}_{DD}) \succ 0$. Therefore, $\mathbf{J}_{DD}$ and $\mathbf{I} + 2\eta\mathbf{J}_{DD}$ share the same set of eigenvectors. Thus, the null space of $\mathbf{J}_{DD}$ and the term $\mathbf{J}_{DD}(\mathbf{I} + 2\eta\mathbf{J}_{DD}) + (\mathbf{I} + 2\eta\mathbf{J}_{DD})\mathbf{J}_{DD}$ are the same, specifically orthogonal to $\mathbf{T}_D$. In other words, the top-left block above is a diagonal matrix with strictly negative eigenvalues. Similarly, we also know that $-2\eta\mathbf{T_G}\mathbf{J}_{DG}^T\mathbf{J}_{DG}\mathbf{T_G}^T$ is a diagonal matrix with negative values. Hence, the above matrix is negative definite. $\square$

### F.1.1 Exponential stability of gradient-regularized WGAN

We now proceed to the Wasserstain GAN scenario. First we lay down equivalent assumptions for the WGAN under which we can guarantee exponential stability in the regularized case. Note that even under these conditions, the unregularized update does not ensure asymptotic stability.

First, we note that due to the linearity of the loss function, it is not necessary that the discriminator be only identically zero on the support for the system to be at equilibrium — it could also be constant on the support. Thus, we relax Assumption I for this case to accommodate this.

**Assumption I. (WGAN, Realizable)** $p_{\boldsymbol{\theta_G^\star}} = p_{\text{data}}$ and $D_{\boldsymbol{\theta_D^\star}}(x) = c, \forall\, x \in \text{supp}(p_{\text{data}})$ for some $c \in \mathbb{R}$.

Next, we state an assumption equivalent to Assumption III. Recall that earlier we wanted $\mathbb{E}_{p_{\text{data}}}[D_{\boldsymbol{\theta_D}}^2(x)]$ to satisfy Property I in the discriminator space. Instead of this function, we will now require that the magnitude of the generator updates satisfy Property I in the discriminator space. Note that the Hessian of this function at equilibrium is $\mathbf{K}_{DG}\mathbf{K}_{DG}^T$.

**Assumption III. (WGAN)** At an equilibrium $(\boldsymbol{\theta_D^\star}, \boldsymbol{\theta_G^\star})$, the functions
$$
\left\| \int_{\mathcal{X}} \nabla_{\boldsymbol{\theta_G}} p_{\boldsymbol{\theta_G}}(x) D_{\boldsymbol{\theta_D}}(x) \right\|^2 \Big|_{\boldsymbol{\theta_G} = \boldsymbol{\theta_G^\star}} \quad \text{and} \quad \left\| \mathbb{E}_{p_{\text{data}}}[\nabla_{\boldsymbol{\theta_D}} D_{\boldsymbol{\theta_D}}(x)] - \mathbb{E}_{p_{\boldsymbol{\theta_G}}}[\nabla_{\boldsymbol{\theta_D}} D_{\boldsymbol{\theta_D}}(x)] \right\|^2 \Big|_{\boldsymbol{\theta_D} = \boldsymbol{\theta_D^\star}}
$$
must satisfy Property I in the discriminator and generator space respectively.

Note that in effect, we get rid of the assumption on the other function and introduce a different function here; in either case, the original system is not asymptotically stable due to zero diagonal blocks in its Jacobian. Next, we retain Assumption IV as it is. These are the only three assumptions we will need.

We will now begin with a lemma similar to Lemma C.2

**Lemma F.4.** *For the dynamical system defined by the WGAN objective in Equation 2 and the updates in Equation 4, under Assumptions I and III under the WGAN case, there exists $\epsilon_D, \epsilon_G > 0$ such that for all $\epsilon'_D \leq \epsilon_D$ and $\epsilon'_G \leq \epsilon_G$, and for any unit vectors $\mathbf{u} \in \mathsf{Null}(\mathbf{K}_{DG}^T), \mathbf{v} \in \mathsf{Null}(\mathbf{K}_{DG})$, $(\boldsymbol{\theta}_{\mathbf{D}}^{\star} + \epsilon'_D \mathbf{u}, \boldsymbol{\theta}_{\mathbf{G}}^{\star} + \epsilon'_G \mathbf{v})$ is an equilibrium point.*

*Proof.* Note that $2\mathbf{K}_{DG}\mathbf{K}_{DG}^T$ is the Hessian of the function $\left\| \int_{\mathcal{X}} \nabla_{\boldsymbol{\theta}_{\mathbf{G}}} p_{\boldsymbol{\theta}_{\mathbf{G}}}(x) D_{\boldsymbol{\theta}_{\mathbf{D}}}(x) \right\|^2$ at equilibrium, namely the magnitude of the generator update. Then, by Assumption III, this function is locally constant along any unit vector $\mathbf{u} \in \mathsf{Null}(\mathbf{K}_{DG}^T)$. That is, for sufficiently small $\epsilon$, if $\boldsymbol{\theta}_{\mathbf{D}} = \boldsymbol{\theta}_{\mathbf{D}}^{\star} + \epsilon \mathbf{u}$, the function value is equal to the value at equilibrium which is zero, because by definition at equilibrium the generator update is zero. Now at $(\boldsymbol{\theta}_{\mathbf{D}}, \boldsymbol{\theta}_{\mathbf{G}}^{\star})$, the discriminator update is zero too since the generator matches the true distribution. Then by Assumption I, it means that $D_{\boldsymbol{\theta}_{\mathbf{D}}}$ is identical over the true support. Then, it is an equilibrium discriminator such that the update for any generator would be zero.

Similarly, as we saw, $2\mathbf{K}_{DG}^T\mathbf{K}_{DG}$ is the Hessian of the function $\left\| \mathbb{E}_{p_{\text{data}}}[\nabla_{\boldsymbol{\theta}_{\mathbf{D}}} D_{\boldsymbol{\theta}_{\mathbf{D}}}(x)] - \mathbb{E}_{p_{\boldsymbol{\theta}_{\mathbf{G}}}}[\nabla_{\boldsymbol{\theta}_{\mathbf{D}}} D_{\boldsymbol{\theta}_{\mathbf{D}}}(x)] \right\|^2$ at equilibrium, namely the magnitude of the discriminator update. We also saw that for sufficiently small $\epsilon'$, if $\boldsymbol{\theta}_{\mathbf{G}} = \boldsymbol{\theta}_{\mathbf{G}}^{\star} + \epsilon' \mathbf{v}$, this function is zero. Thus, at $(\boldsymbol{\theta}_{\mathbf{D}}^{\star}, \boldsymbol{\theta}_{\mathbf{G}})$, the discriminator update is zero. Furthermore, the generator update is zero too because the discriminator is constant throughout the support. Thus, $(\boldsymbol{\theta}_{\mathbf{D}}^{\star}, \boldsymbol{\theta}_{\mathbf{G}})$ is an equilibrium point and from Assumption I we can conclude that $p_{\boldsymbol{\theta}_{\mathbf{G}}} = p_{\text{data}}$. Thus, it is an equilibrium generator such that the update for any discriminator would be zero.

In summary, for all slight perturbations along $\mathbf{u} \in \mathsf{Null}(\mathbf{K}_{DG}^T), \mathbf{v} \in \mathsf{Null}(\mathbf{K}_{DG})$ we have established that the discriminator and generator individually satisfy the requirements of an equilibrium discriminator and generator pair, and therefore the system is itself is in equilibrium for these perturbations. □

Now, we show that this system can again be projected to a subspace orthogonal the equilibrium subspace such that the resulting Jacobian of the reduced system is Hurwitz. While earlier we chose $\mathbf{T}_D$ based on the matrix $\mathbf{K}_{DD}$ now we will choose it based on $\mathbf{K}_{DG}^T$.

**Lemma F.5.** *For the dynamical system defined by the GAN objective in Equation 2 and the updates in Equation 4, consider the eigenvalue decompositions $\mathbf{K}_{DG}\mathbf{K}_{DG}^T = \mathbf{U_D}\boldsymbol{\Lambda_D}\mathbf{U_D}^T$ and $\mathbf{K}_{DG}^T\mathbf{K}_{DG} = \mathbf{U_G}\boldsymbol{\Lambda_G}\mathbf{U_G}^T$. Let $\mathbf{U_D} = [\mathbf{T}_D^T, \mathbf{T}_D'^T]$ and $\mathbf{U_G} = [\mathbf{T}_G^T, \mathbf{T}_G'^T]$ such that $\mathsf{Col}(\mathbf{T}_D'^T) = \mathsf{Null}(\mathbf{K}_{DD})$ and $\mathsf{Col}(\mathbf{T}_G'^T) = \mathsf{Null}(\mathbf{K}_{DG})$. Consider the projections, $\boldsymbol{\gamma_D} = \mathbf{T}_D \boldsymbol{\theta}_D$ and $\boldsymbol{\gamma_G} = \mathbf{T}_G \boldsymbol{\theta}_G$. Then, the block in the Jacobian at equilibrium that corresponds to the projected system has the form:*

$$\mathbf{J}' = \begin{bmatrix} \mathbf{J}'_{DD} & \mathbf{J}'_{DG} \\ -\mathbf{J}'^T_{DG} & \mathbf{J}'_{GG} \end{bmatrix} = \begin{bmatrix} 0 & \mathbf{T}_D \mathbf{J}_{DG} \mathbf{T}_G^T \\ -\mathbf{T}_G \mathbf{J}_{DG}^T \mathbf{T}_D^T & -2\eta \mathbf{T}_G \mathbf{J}_{DG}^T \mathbf{J}_{DG} \mathbf{T}_G^T \end{bmatrix}$$

*Furthermore $\mathbf{J}'_{GG} \prec 0$ and $\mathbf{J}'^T_{DG}$ is full column rank.*

*Proof.* Observe that the form of $\mathbf{J}'$ follows from Lemma F.1 by substituting $\mathbf{J}_{DD} = 0$. Furthermore, like we have seen before, observe that $\mathbf{T}_G \mathbf{J}_{DG}^T \mathbf{J}_{DG} \mathbf{T}_G^T$ is a diagonal matrix with positive eigenvalues and therefore $\mathbf{J}'_{GG} \prec 0$. Similarly, $\mathbf{J}_{DG}^T \mathbf{T}_D$ is a full column rank matrix because we have projected it to the subspace orthogonal to its null space. However, we need to show that $\mathbf{T}_G^T \mathbf{J}_{DG}^T \mathbf{T}_D$ which may have fewer rows, did not reduce in its rank. This is indeed true, since this is effectively a projection onto the subspace orthogonal to its left null space. □

We now compile the above lemmas to prove our main result in Theorem 3.2.

*Proof.* The first part of the theorem statement for the conventional GAN follows from Lemma F.1, F.2, F.3.

To prove the second part it is sufficient to show that the projected Jacobian of the linearized system in Lemma F.5 is Hurwitz, from which exponential stability of the original system follows from Theorem A.4. The fact that this is Hurwitz follows as usual from Lemma G.2 after we flip the discriminator and generator variables:

$$\begin{bmatrix} \mathbf{J}'_{GG} & -\mathbf{J}'^{T}_{DG} \\ \mathbf{J}'_{DG} & 0 \end{bmatrix}.$$

The Jacobian is thus Hurwitz because $\mathbf{J}'_{GG}$ is negative definite and $-\mathbf{J}'^{T}_{DG}$ is full column rank. Now, for the eigenvalue bounds we have from Lemma G.2 that:

- If $\mathrm{Im}(\lambda) = 0$, then $\mathrm{Re}(\lambda) \leq -\frac{2f'^2(0)\eta\lambda^{(+)}_{\min}(\mathbf{K}_{DG}\mathbf{K}^{T}_{DG})\lambda^{(+)}_{\min}(\mathbf{K}^{T}_{DG}\mathbf{K}_{DG})}{4f'^2(0)\eta^2\lambda_{\max}(\mathbf{K}_{DG}\mathbf{K}^{T}_{DG})\lambda^{(+)}_{\min}(\mathbf{K}_{DG}\mathbf{K}^{T}_{DG})+\lambda^{(+)}_{\min}(\mathbf{K}^{T}_{DG}\mathbf{K}_{DG})}$

- If $\mathrm{Im}(\lambda) \neq 0$, then $\mathrm{Re}(\lambda) \leq -\eta f'^2(0)\lambda^{(+)}_{\min}(\mathbf{K}_{DG}\mathbf{K}^{T}_{DG})$

However, this can be further simplified to arrive at the given bound by noting that all the non-zero eigenvalues of any matrix $\mathbf{AB}$ is also equal to the non-zero eigenvalues of the matrix $\mathbf{BA}$. Therefore, we can replace every occurrence of $\mathbf{K}_{DG}\mathbf{K}^{T}_{DG}$ with $\mathbf{K}^{T}_{DG}\mathbf{K}_{DG}$ in the above inequality.

□

Additionally, we show that we can find a Lyapunov function that satisfies LaSalle's principle for the projected linearized system.

**Fact F.1.** *For the linearized projected system with the Jacobian $\mathbf{J}'$, we have that $1/2\|\gamma_D - \gamma^{\star}{}_D\|^2 + 1/2\|\gamma_G - \gamma^{\star}{}_G\|^2$ is a Lyapunov function such that for all non-equilbrium points, it either always decreases or only instantaneously remains constant.*

*Proof.* Note that the Lyapunov function is zero only at the equilibrium of the projected system. Furthermore, it is straightforward to verify that the rate at which this changes is given by $-2\eta(\gamma_{\mathbf{G}} - \gamma^{\star}_{\mathbf{G}})^T\mathbf{T}_G\mathbf{K}^{T}_{DG}\mathbf{K}_{DG}\mathbf{T}^{T}_G(\gamma_{\mathbf{G}} - \gamma^{\star}_{\mathbf{G}})$ which is non-positive. Clearly this is zero only when $\gamma_{\mathbf{G}} = \gamma^{\star}_{\mathbf{G}}$ because $\mathbf{T}_G\mathbf{K}^{T}_{DG}\mathbf{K}_{DG}\mathbf{T}^{T}_G$ is positive definite. When this rate is indeed zero, we have that for the linearized system, $\dot{\gamma}_{\mathbf{G}} = \mathbf{T}_G\mathbf{K}^{T}_{DG}\mathbf{T}^{T}_D(\gamma_D - \gamma^{\star}{}_D)$ because the other term becomes zero. For the system to identically stay on the manifold $\gamma_{\mathbf{G}} = \gamma^{\star}_{\mathbf{G}}$ we need $\dot{\gamma}_{\mathbf{G}} = 0$, which happens only when $\gamma_D = \gamma^{\star}{}_D$ because $\mathbf{T}_G\mathbf{K}^{T}_{DG}\mathbf{T}^{T}_D$ is full column rank. When that is the case, we are at equilibrium.

□

### F.2 Intuition based on arg-max differentiation

In an ideal world, an optimizer would hope to have access to a function $\theta^{\star}_{\mathbf{D}}(\theta_{\mathbf{G}}) = \arg\max_{\theta_{\mathbf{D}}} V(D_{\theta_{\mathbf{D}}}, G_{\theta_{\mathbf{G}}})$, which is basically the optimal discriminator as a function of the generator; given this, the optimizer should be able to update the generator with respect to that. Then, the update can be shown to be the following (for clarity we use the superscript $t$ and $t+1$ to denote the current and the updated parameters):

$$\theta^{(t+1)}_G := \theta^{(t)}_G - \alpha \underbrace{\nabla_{\theta_{\mathbf{G}}} V(D_{\theta^{\star}_{\mathbf{D}}(\theta^{(t)}_G)}, G_{\theta_{\mathbf{G}}})}_{\text{conventional update}} -$$

$$\alpha \left(\frac{\partial\theta^{\star}_{\mathbf{D}}(\theta_{\mathbf{G}})}{\partial\theta_{\mathbf{G}}}\right)^T\bigg|_{\theta_{\mathbf{G}}=\theta^{(t)}_G} \nabla_{\theta_{\mathbf{D}}} V(D_{\theta_{\mathbf{D}}}, G_{\theta^{(t)}_G})\bigg|_{\theta_{\mathbf{D}}=\theta^{\star}_{\mathbf{D}}(\theta^{(t)}_G)}$$

Observe that the last term is zero because, for the optimal discriminator $\nabla_{\theta_{\mathbf{D}}} V(D_{\theta_{\mathbf{D}}}, G_{\theta^{(t)}_G}) = 0$. However, in practice, we would not be at the optimal discriminator and therefore this term may be non-zero. Our hypothesis is that, instead of ignoring this term like it is done for the conventional updates,

retaining this term may prove to be useful. To do so, we simply plug in the current discriminator for $\boldsymbol{\theta}_D^\star(\boldsymbol{\theta}_G^\star)$ while computing this term, and furthermore, estimate the value of $\nabla_{\boldsymbol{\theta_G}}\boldsymbol{\theta}_\mathbf{D}^\star(\boldsymbol{\theta_G})$ using the following equation:

$$0 = \nabla_{\boldsymbol{\theta_D}} V(\boldsymbol{\theta_D}, \boldsymbol{\theta_G^{(t)}})\Big|_{\boldsymbol{\theta_D}=\boldsymbol{\theta}_\mathbf{D}^\star(\boldsymbol{\theta_G}^{(t)})} \implies$$

$$0 = \frac{\partial}{\partial \boldsymbol{\theta_G}}\left(\nabla_{\boldsymbol{\theta_D}} V(D_{\boldsymbol{\theta_D}}, G_{\boldsymbol{\theta_G}})|_{\boldsymbol{\theta_D}=\boldsymbol{\theta}_\mathbf{D}^\star(\boldsymbol{\theta_G}^{(t)})}\right)\Big|_{\boldsymbol{\theta_G}=\boldsymbol{\theta_G}^{(t)}}$$

$$= \frac{\partial \nabla_{\boldsymbol{\theta_D}} V(D_{\boldsymbol{\theta_D}}, G_{\boldsymbol{\theta_G}})}{\partial \boldsymbol{\theta_G}}\Bigg|_{\substack{\boldsymbol{\theta_D}=\boldsymbol{\theta}_\mathbf{D}^\star(\boldsymbol{\theta_G}),\\ \boldsymbol{\theta_G}=\boldsymbol{\theta_G}^{(t)}}} + \nabla_{\boldsymbol{\theta_D}}^2 V(D_{\boldsymbol{\theta_D}}, D_{\boldsymbol{\theta_G}^{(t)}})|_{\boldsymbol{\theta_D}=\boldsymbol{\theta}_\mathbf{D}^\star(\boldsymbol{\theta_G})} \frac{\partial \boldsymbol{\theta}_\mathbf{D}^\star(\boldsymbol{\theta_G})}{\partial \boldsymbol{\theta_G}}\Bigg|_{\boldsymbol{\theta_G}=\boldsymbol{\theta_G}^{(t)}}$$

$$(8)$$

In the second step above, we apply the chain rule. Rearranging, we get:

$$\frac{\partial \boldsymbol{\theta}_\mathbf{D}^\star(\boldsymbol{\theta_G})}{\partial \boldsymbol{\theta_G}}\Bigg|_{\boldsymbol{\theta_G}=\boldsymbol{\theta_G^{(t)}}} = -\left(\nabla_{\boldsymbol{\theta_D}}^2 V(D_{\boldsymbol{\theta_D}}, D_{\boldsymbol{\theta_G}^{(t)}})|_{\boldsymbol{\theta_D}=\boldsymbol{\theta}_\mathbf{D}^\star(\boldsymbol{\theta_G})}\right)^{-1} \frac{\partial \nabla_{\boldsymbol{\theta_D}} V(D_{\boldsymbol{\theta_D}}, G_{\boldsymbol{\theta_G}})}{\partial \boldsymbol{\theta_G}}\Bigg|_{\substack{\boldsymbol{\theta_D}=\boldsymbol{\theta}_\mathbf{D}^\star(\boldsymbol{\theta_G}),\\ \boldsymbol{\theta_G}=\boldsymbol{\theta_G}^{(t)}}}$$

$$(9)$$

Since we hope the objective to be concave in the discriminator parameters, we can approximate the Hessian as $\nabla_{\boldsymbol{\theta_D}}^2 V(D_{\boldsymbol{\theta_D}}, D_{\boldsymbol{\theta_G}^{(t)}}) = -\mathbf{I}/\eta$. Plugging this into the update equation of $\boldsymbol{\theta}_G^{(t+1)}$ and also replacing the optimal discriminator with the current discriminator, we get the following update rule which is equivalent to the original one presented in Equation 4:

$$\boldsymbol{\theta}_G^{(t+1)} := \boldsymbol{\theta}_G^{(t)} - \alpha \nabla_{\boldsymbol{\theta_G}} V(D_{\boldsymbol{\theta_D}}, G_{\boldsymbol{\theta_G}})|_{\boldsymbol{\theta_G}=\boldsymbol{\theta_G^{(t)}}} - \alpha\eta \left(\frac{\partial \nabla_{\boldsymbol{\theta_D}} V(D_{\boldsymbol{\theta_D}}, G_{\boldsymbol{\theta_G}})}{\partial \boldsymbol{\theta_G}}\right)^T\Bigg|_{\boldsymbol{\theta_G}=\boldsymbol{\theta_G}^{(t)}} \nabla_{\boldsymbol{\theta_D}} V(D_{\boldsymbol{\theta}}, G_{\boldsymbol{\theta}_G^{(t)}})$$

$$(10)$$

### F.3  Mode Collapse and Relation to 1-unrolled updates

Our regularization term also has natural and intuitive connections to an important issue that arises in GAN optimization called mode collapse. Mode collapse is a situation where a GAN may enter an irrecoverable failure state where the generator incorrectly assigns all its probability mass to a small region in space. This arises because a globally optimal strategy for the generator is to push all its mass towards the single point that the discriminator is the most confident about being a real data point. To overcome this the generator needs more "foresight" – it must know that when it collapses all the mass, the discriminator will subsequently label the collapsed point as fake data. Our penalty indeed encodes this foresight, because the discriminator's ability to outdo the generator is quantified by the magnitude of the discriminator's gradient. More clearly, our generator seeks a state where it can spread data out enough, to make sure the discriminator has no obvious countermeasure (i.e., no big gradients).

In fact, we can show how our penalty term and 1-unrolled GANs have very similar structure because intuitively both provide a one-step lookahead to the generator. More precisely, we can arrive at 1-unrolled updates if we simplify our updates further and replace $\boldsymbol{\theta_D}$ by an "unrolled" $\boldsymbol{\theta_D} + \eta \nabla_{\boldsymbol{\theta_D}} \hat{V}(D_{\boldsymbol{\theta_D}}, G_{\boldsymbol{\theta_G}})$.

We begin by simplifying the 1-unrolled updates. The key idea of a 1-unrolled update is to allow the generator to *explicitly* foresee how the discriminator would react to its update, and optimize accordingly:

$$\boldsymbol{\theta_G}^{(t+1)} := \boldsymbol{\theta_G}^{(t)} - \alpha \nabla_{\boldsymbol{\theta_G}} V(D_{\underbrace{\boldsymbol{\theta_D} + \eta \nabla_{\boldsymbol{\theta_D}} V(D_{\boldsymbol{\theta_D}}, G_{\boldsymbol{\theta_G}})}_{\text{unrolling}}}, G_{\boldsymbol{\theta_G}})\Bigg|_{\boldsymbol{\theta_G}=\boldsymbol{\theta_G}^{(t)}}$$

$$= \boldsymbol{\theta_G}^{(t)} - \alpha \nabla_{\boldsymbol{\theta_G}} V(D_{\boldsymbol{\theta_D} + \eta \nabla_{\boldsymbol{\theta_D}} V(D_{\boldsymbol{\theta_D}}, G_{\boldsymbol{\theta_G}^{(t)}})}, G_{\boldsymbol{\theta_G}})\Big|_{\boldsymbol{\theta_G}=\boldsymbol{\theta_G}^{(t)}}$$

$$- \alpha \nabla_{\boldsymbol{\theta}_{\mathbf{G}}} V(D_{\boldsymbol{\theta}_{\mathbf{D}} + \eta \nabla_{\boldsymbol{\theta}_{\mathbf{D}}} V(D_{\boldsymbol{\theta}_{\mathbf{D}}}, G_{\boldsymbol{\theta}_{\mathbf{G}}})}, G_{\boldsymbol{\theta}_{\mathbf{G}}{}^{(t)}})\Big|_{\boldsymbol{\theta}_{\mathbf{G}} = \boldsymbol{\theta}_{\mathbf{G}}{}^{(t)}}$$

$$= \boldsymbol{\theta}_{\mathbf{G}}{}^{(t)} - \alpha \nabla_{\boldsymbol{\theta}_{\mathbf{G}}} V(D_{\boldsymbol{\theta}_{\mathbf{D}} + \eta \nabla_{\boldsymbol{\theta}_{\mathbf{D}}} V(D_{\boldsymbol{\theta}_{\mathbf{D}}}, G_{\boldsymbol{\theta}_{\mathbf{G}}{}^{(t)}})}, G_{\boldsymbol{\theta}_{\mathbf{G}}})\Big|_{\boldsymbol{\theta}_{\mathbf{G}} = \boldsymbol{\theta}_{\mathbf{G}}{}^{(t)}}$$

$$- \alpha \eta \left( \frac{\partial \nabla_{\boldsymbol{\theta}_{\mathbf{D}}} V(D_{\boldsymbol{\theta}_{\mathbf{D}}}, G_{\boldsymbol{\theta}_{\mathbf{G}}})}{\partial \boldsymbol{\theta}_{\mathbf{G}}} \right)^{T} \Bigg|_{\boldsymbol{\theta}_{\mathbf{G}} = \boldsymbol{\theta}_{\mathbf{G}}{}^{(t)}} \nabla_{\boldsymbol{\theta}'_{\mathbf{D}}} V(D_{\boldsymbol{\theta}'_{\mathbf{D}}}, G_{\boldsymbol{\theta}_{\mathbf{G}}{}^{(t)}})\Big|_{\boldsymbol{\theta}'_{\mathbf{D}} = \boldsymbol{\theta}_{\mathbf{D}} + \eta \nabla_{\boldsymbol{\theta}_{\mathbf{D}}} V(D_{\boldsymbol{\theta}_{\mathbf{D}}}, G_{\boldsymbol{\theta}_{\mathbf{G}}{}^{(t)}})}$$

In the first step, we compute gradient with respect to $\boldsymbol{\theta}_{\mathbf{G}}$ as the sum of the gradients with respect to the two instances of $\boldsymbol{\theta}_{\mathbf{G}}$ that occur in $V(D_{\boldsymbol{\theta}_{\mathbf{D}} + \eta \nabla_{\boldsymbol{\theta}_{\mathbf{D}}} V(D_{\boldsymbol{\theta}_{\mathbf{D}}}, G_{\boldsymbol{\theta}_{\mathbf{G}}})}, G_{\boldsymbol{\theta}_{\mathbf{G}}})$, the first one that occurs as the second argument to $V(\cdot, \cdot)$, and the second one that occurs in the unrolled update of the first argument. In the second step, we apply the chain rule on the second gradient.

We can compare our updates in Equation 10 with the above to show how our updates are more flexible in terms of using the lookahead. While both have two similar terms, a crucial difference is that in the latter, every occurrence of the discriminator parameters (except one) has an additional unrolled update, namely $\eta \nabla_{\boldsymbol{\theta}_{\mathbf{D}}} V(D_{\boldsymbol{\theta}_{\mathbf{D}}}, G_{\boldsymbol{\theta}_{\mathbf{G}}})$. Clearly, this should provide more power to the latter; however in practice, we observe that our technique can be more powerful than 1-unrolled or even 10-unrolled updates (which are in fact much slower to run). The reason is that the unrolled updates constrain $\eta$ to be small, typically of the order $10^{-4}$ which is the step size. It would not be possible to increase $\eta$ to greater magnitudes as it would be equivalent to a coarse step size in the unrolling. Our method on the other hand, allows for larger $\eta$ because the discriminator is retained as it is; in some sense, our penalty provides a way of extracting and leveraging the unrolled update more flexibly.

# G    Eigenvalue bounds

In this section, we prove one of the most useful lemmas that we used in our proofs, that matrices of the form $\begin{bmatrix} -\mathbf{Q} & \mathbf{P}; & -\mathbf{P}^{T} & 0 \end{bmatrix}$ are Hurwitz when $Q \succ 0$ and $P$ is full column rank. We also prove eigenvalue bounds for such a matrix. To do so, we begin with a simple fact:

**Lemma G.1.** *For* $\mathbf{Q} \succeq 0$ *be a real symmetric matrix. If* $\mathbf{a}^{T}\mathbf{Q}\mathbf{a} = c$, *then* $\mathbf{a}^{T}\mathbf{Q}^{T}\mathbf{Q}\mathbf{a} \in [\lambda_{\min}(\mathbf{Q})c, \lambda_{\max}(\mathbf{Q})c, ]$.

*Proof.* Let $\mathbf{Q} = \mathbf{U}\boldsymbol{\Lambda}\mathbf{U}^{T}$ be the eigenvalue decomposition of $\mathbf{Q}$. Let $\mathbf{x} = \mathbf{U}\mathbf{a}$. Then, $c = \mathbf{x}\boldsymbol{\Lambda}\mathbf{x}$ or in other words, $c = \sum x_i^2 \lambda_i$. Similarly, $\mathbf{a}^{T}\mathbf{Q}^{T}\mathbf{Q}\mathbf{a} = \sum x_i^2 \lambda_i^2$ which differs from $c$ by a multiplicative factor within $[\lambda_{\min}(\mathbf{Q}), \lambda_{\max}(\mathbf{Q})]$. □

We now prove our main result.

**Lemma G.2.** *Let*

$$\mathbf{J} = \begin{bmatrix} -\mathbf{Q} & \mathbf{P} \\ -\mathbf{P}^{T} & 0 \end{bmatrix},$$

*where* $\mathbf{Q}$ *is a symmetric real positive definite matrix and* $\mathbf{P}$ *is a full column rank matrix. Then,* $\mathrm{Re}(\lambda) < 0$ *for every eigenvalue* $\lambda$ *of* $\mathbf{J}$. *In fact,*

- *When* $\mathrm{Im}(\lambda) = 0$,

$$\mathrm{Re}(\lambda) \leq -\frac{\lambda_{\min}(\mathbf{Q})\lambda_{\min}(\mathbf{P}^{T}\mathbf{P})}{\lambda_{\max}(\mathbf{Q})\lambda_{\min}(\mathbf{Q}) + \lambda_{\min}(\mathbf{P}^{T}\mathbf{P})}$$

- *When* $\mathrm{Im}(\lambda) \neq 0$,

$$\mathrm{Re}(\lambda) \leq -\frac{\lambda_{\min}(\mathbf{Q})}{2}$$

*Proof.* We consider a generic eigenvector equation and equate the real and complex parts together so as to arrive at our bounds. Consider the following eigenvector equation:

$$\begin{bmatrix} -\mathbf{Q} & \mathbf{P} \\ -\mathbf{P}^T & 0 \end{bmatrix} \begin{bmatrix} \mathbf{a}_1 + i\mathbf{a}_2 \\ \mathbf{b}_1 + i\mathbf{b}_2 \end{bmatrix} = (\lambda_1 + i\lambda_2) \begin{bmatrix} \mathbf{a}_1 + i\mathbf{a}_2 \\ \mathbf{b}_1 + i\mathbf{b}_2 \end{bmatrix},$$

where $\mathbf{a}_i, \mathbf{b}_i, \lambda_i$ are all real-valued. We assume that the vector is normalized i.e., $\mathbf{a}_1^2 + \mathbf{a}_2^2 + \mathbf{b}_1^2 + \mathbf{b}_2^2 = 1$. So, in case $\lambda_2 = 0$, we assume that $\mathbf{a}_1^2 + \mathbf{b}_1^2 = 1$. We want to show that $\lambda_1 < 0$. Let us first rewrite the above equation as follows:

$$\begin{bmatrix} -\mathbf{Q}\mathbf{a}_1 + \mathbf{P}\mathbf{b}_1 + i(-\mathbf{Q}\mathbf{a}_2 + \mathbf{P}\mathbf{b}_2) \\ -\mathbf{P}^T\mathbf{a}_1 + i(-\mathbf{P}^T\mathbf{a}_2) \end{bmatrix} = \begin{bmatrix} \lambda_1\mathbf{a}_1 - \lambda_2\mathbf{a}_2 + i(\lambda_1\mathbf{a}_2 + \lambda_2\mathbf{a}_1) \\ \lambda_1\mathbf{b}_1 - \lambda_2\mathbf{b}_2 + i(\lambda_1\mathbf{b}_2 + \lambda_2\mathbf{b}_1) \end{bmatrix}$$

We can then equate the real and imaginary parts.

$$-\mathbf{Q}\mathbf{a}_1 + \mathbf{P}\mathbf{b}_1 = \lambda_1\mathbf{a}_1 - \lambda_2\mathbf{a}_2 \tag{11}$$

$$-\mathbf{P}^T\mathbf{a}_1 = \lambda_1\mathbf{b}_1 - \lambda_2\mathbf{b}_2 \tag{12}$$

$$-\mathbf{Q}\mathbf{a}_2 + \mathbf{P}\mathbf{b}_2 = \lambda_1\mathbf{a}_2 + \lambda_2\mathbf{a}_1 \tag{13}$$

$$-\mathbf{P}^T\mathbf{a}_2 = \lambda_1\mathbf{b}_2 + \lambda_2\mathbf{b}_1 \tag{14}$$

We now multiply the above equations by $\mathbf{a}_1^T, \mathbf{b}_1^T, \mathbf{a}_2^T, \mathbf{b}_2^T$ respectively and add them:

$$\begin{aligned} \mathbf{a}_1^T(-\mathbf{Q}\mathbf{a}_1 + \mathbf{P}\mathbf{b}_1) - \mathbf{b}_1^T\mathbf{P}^T\mathbf{a}_1 &= \mathbf{a}_1^T(\lambda_1\mathbf{a}_1 - \lambda_2\mathbf{a}_2) + \mathbf{b}_1^T(\lambda_1\mathbf{b}_1 - \lambda_2\mathbf{b}_2) \\ +\mathbf{a}_2^T(-\mathbf{Q}\mathbf{a}_2 + \mathbf{P}\mathbf{b}_2) - \mathbf{b}_2^T\mathbf{P}^T\mathbf{a}_2 & \quad +\mathbf{a}_2^T(\lambda_1\mathbf{a}_2 + \lambda_2\mathbf{a}_1) + \mathbf{b}_2^T(\lambda_1\mathbf{b}_2 + \lambda_2\mathbf{b}_1) \end{aligned}$$

As a result, only square terms and $\lambda_1$ terms remain:

$$-\mathbf{a}_1^T\mathbf{Q}\mathbf{a}_1 - \mathbf{a}_2^T\mathbf{Q}\mathbf{a}_2 = \lambda_1(\mathbf{a}_1^T\mathbf{a}_1 + \mathbf{a}_2^T\mathbf{a}_2 + \mathbf{b}_1^T\mathbf{b}_1 + \mathbf{b}_2^T\mathbf{b}_2) = \lambda_1$$

**Proof for $\lambda_1 < 0$.** Now observe that $-\mathbf{a}_1^T\mathbf{Q}\mathbf{a}_1 - \mathbf{a}_2^T\mathbf{Q}\mathbf{a}_2 \leq 0$ because $\mathbf{Q} \succ 0$. If $-\mathbf{a}_1^T\mathbf{Q}\mathbf{a}_1 - \mathbf{a}_2^T\mathbf{Q}\mathbf{a}_2 < 0$, it would immediately imply that $\lambda_1 < 0$.

However this may not be true, and that would happen only when $\mathbf{a}_1 = 0$ and $\mathbf{a}_2 = 0$ because $\mathbf{Q}_1, \mathbf{Q}_2 \prec 0$. We will show that this case would not occur. First of all, this would force $\lambda_1 = 0$ to ensure the above equality. By applying the Equations 12 and 14, we can conclude that $\lambda_2\mathbf{b}_2 = 0$ and $\lambda_2\mathbf{b}_1 = 0$. Since one of $\mathbf{b}_1, \mathbf{b}_2 \neq 0$ this implies that $\lambda_2 = 0$ too. Now, by applying Equation 11 and 13, we have that $\mathbf{P}\mathbf{b}_1 = 0$ and $\mathbf{P}\mathbf{b}_2 = 0$. Since one of $\mathbf{b}_1, \mathbf{b}_2 \neq 0$ (if they were both zero, our eigenvector would itself be zero), this implies that $\mathbf{P}$ is not a full column rank matrix, which is a contradiction of our assumption. Therefore, it cannot be the case that both $\mathbf{a}_1 = 0$ and $\mathbf{a}_2 = 0$.

**Stricter bound.** Now, we prove our bounds on $\lambda_1$. (Note that an easy lower bound follows as $\lambda_1 \geq -\lambda_{\max}(\mathbf{Q})(\|\mathbf{a}_1\|^2 + \|\mathbf{a}_2\|^2) \geq -\lambda_{\max}(\mathbf{Q})$ but we are interested in an upper bound). In order to prove the upper bound, we multiply Equations 11 and Equations 13 by $-\mathbf{a}_2^T$ and $\mathbf{a}_1^T$ respectively and sum them up, and Equations 12 and Equations 14 by $-\mathbf{b}_2^T$ and $\mathbf{b}_1^T$ respectively and sum them up.

$$\begin{aligned} \mathbf{a}_2^T\mathbf{Q}\mathbf{a}_1 - \mathbf{a}_2^T\mathbf{P}\mathbf{b}_1 - \mathbf{a}_1^T\mathbf{Q}\mathbf{a}_2 + \mathbf{a}_1^T\mathbf{P}\mathbf{b}_2 &= -\mathbf{a}_2^T\lambda_1\mathbf{a}_1 + \mathbf{a}_2^T\lambda_2\mathbf{a}_2 + \mathbf{a}_1^T\lambda_1\mathbf{a}_2 + \mathbf{a}_1^T\lambda_2\mathbf{a}_1 \\ \implies -\mathbf{a}_2^T\mathbf{P}\mathbf{b}_1 + \mathbf{a}_1^T\mathbf{P}\mathbf{b}_2 &= \lambda_2(\|\mathbf{a}_2\|^2 + \|\mathbf{a}_1^2\|) \\ \mathbf{b}_2^T\mathbf{P}^T\mathbf{a}_1 - \mathbf{b}_1^T\mathbf{P}^T\mathbf{a}_2 &= -\mathbf{b}_2^T\lambda_1\mathbf{b}_1 + \mathbf{b}_2^T\lambda_2\mathbf{b}_2 + \mathbf{b}_1^T\lambda_1\mathbf{b}_2 + \mathbf{b}_1^T\lambda_2\mathbf{b}_1 \\ \implies \mathbf{b}_2^T\mathbf{P}^T\mathbf{a}_1 - \mathbf{b}_1^T\mathbf{P}^T\mathbf{a}_2 &= \lambda_2(\|\mathbf{b}_2\|^2 + \|\mathbf{b}_1\|^2) \end{aligned}$$

As a consequence,

$$\lambda_2(\|\mathbf{a}_2\|^2 + \|\mathbf{a}_1^2\|) = \lambda_2(\|\mathbf{b}_2\|^2 + \|\mathbf{b}_1\|^2)$$

From the above we have that either $\lambda_2 = 0$ or $\|\mathbf{b}_2\|^2 + \|\mathbf{b}_1\|^2 = \|\mathbf{a}_2\|^2 + \|\mathbf{a}_1^2\| = 1/2$. Now, if $\lambda_2 \neq 0$, since $-\mathbf{a}_1^T\mathbf{Q}\mathbf{a}_1 - \mathbf{a}_2^T\mathbf{Q}\mathbf{a}_2 = \lambda_1$, we immediately get a bound $\lambda_1 \leq -\lambda_{\min}(\mathbf{Q})/2$.

In the former case, since the imaginary part of the eigenvalue is zero i.e., $\lambda_2 = 0$, the imaginary part of the eigenvector must be zero too i.e., $\mathbf{a}_2 = \mathbf{b}_2 = 0$. Then, we have the equations:

$$-\mathbf{Q}\mathbf{a}_1 + \mathbf{P}\mathbf{b}_1 = \lambda_1 \mathbf{a}_1$$
$$-\mathbf{P}^T \mathbf{a}_1 = \lambda_1 \mathbf{b}_1$$

Rearranging and squaring the first equation we get:

$$\mathbf{b}_1^T \mathbf{P}^T \mathbf{P} \mathbf{b}_1 = \mathbf{a}_1^T (\lambda_1 \mathbf{I} + \mathbf{Q})^T (\lambda_1 \mathbf{I} + \mathbf{Q}) \mathbf{a}_1$$
$$= \mathbf{a}_1^T (\lambda_1^2 I + 2\lambda_1 \mathbf{Q} + \mathbf{Q}^T \mathbf{Q}) \mathbf{a}_1$$

Then,

$$\implies \lambda_{\min}(\mathbf{P}^T \mathbf{P}) \|\mathbf{b}_1^2\| \leq \lambda_1^2 \|\mathbf{a}_1\|^2 - 2\lambda_1^2 + \mathbf{a}_1^T \mathbf{Q}^T \mathbf{Q}\mathbf{a}$$
$$\lambda_{\min}(\mathbf{P}^T \mathbf{P}) \leq (\lambda_1^2 + \lambda_{\min}(\mathbf{P}^T \mathbf{P})) \|\mathbf{a}_1\|^2 - 2\lambda_1^2 + \mathbf{a}_1^T \mathbf{Q}^T \mathbf{Q}\mathbf{a}$$
$$\leq (-\lambda_1^3 - \lambda_{\min}(\mathbf{P}^T \mathbf{P})\lambda_1) \frac{1}{\lambda_{\min}(\mathbf{Q})} - 2\lambda_1^2 - \lambda_1 \lambda_{\max}(\mathbf{Q})$$
$$\leq -\frac{\lambda_1}{\lambda_{\min}(\mathbf{Q})} \left( \lambda_1^2 + 2\lambda_{\min}(\mathbf{Q})\lambda_1 + \lambda_{\max}(\mathbf{Q})\lambda_{\min}(\mathbf{Q}) + \lambda_{\min}(\mathbf{P}^T \mathbf{P}) \right).$$

In the first step, we make use of the fact that $\lambda_1 = -\mathbf{a}_1^T \mathbf{Q}\mathbf{a}_1$. In the second step, we use $\|\mathbf{a}_1\|^2 + \|\mathbf{b}_1\|^2 = 1$. In the third step, we use Lemma G.1 i.e., $\mathbf{a}_1^T \mathbf{Q}^T \mathbf{Q}\mathbf{a}_1 \leq -\lambda_1 \lambda_{\max}(\mathbf{Q})$. We also use the fact that since $\lambda_1 = -\mathbf{a}_1^T \mathbf{Q}\mathbf{a}_1$, $\|\mathbf{a}_1\|^2 \leq \frac{-\lambda_1}{\lambda_{\min}(\mathbf{Q})}$.

How do we upper bound $\lambda_1$ using this inequality?

Let us examine the quadratic in $\lambda_1$ in the above expression. Since the discriminant of this quadratic is $4\lambda_{\min}^2(\mathbf{Q}) - 4\lambda_{\max}(\mathbf{Q})\lambda_{\min}(\mathbf{Q}) - 4\lambda_{\min}(\mathbf{P}^T \mathbf{P}) \leq -4\lambda_{\min}(\mathbf{P}^T \mathbf{P}) < 0$, the quadratic always takes the same sign, specifically positive. Next, note that the quadratic reaches its minimum at $-\lambda_{\min}(\mathbf{Q})$.

Now, $\lambda_1$ can either satisfy $\lambda_1 \leq -\lambda_{\min}(\mathbf{Q})$ or $0 \geq \lambda_1 > -\lambda_{\min}(\mathbf{Q})$. Since the former is already an upper bound, we will derive an upper bound in the latter case. Now, in the interval $(-\lambda_{\min}(Q), 0]$, the quadratic in $\lambda_1$ increases, and therefore for this interval, the above inequality can be rewritten by plugging in $\lambda_1 = 0$ inside the quadratic. On plugging it, we will get:

$$\lambda_{\min}(\mathbf{P}^T \mathbf{P}) \leq -\frac{\lambda_1}{\lambda_{\min}(\mathbf{Q})} \left( \lambda_{\max}(\mathbf{Q})\lambda_{\min}(\mathbf{Q}) + \lambda_{\min}(\mathbf{P}^T \mathbf{P}) \right)$$
$$\lambda_1 \leq -\lambda_{\min}(\mathbf{Q}) \frac{\lambda_{\min}(\mathbf{P}^T \mathbf{P})}{\lambda_{\max}(\mathbf{Q})\lambda_{\min}(\mathbf{Q}) + \lambda_{\min}(\mathbf{P}^T \mathbf{P})}$$

Observe that the term on the right here lies in $(-\lambda_{\min}(\mathbf{Q}), 0)$. This is because the fraction that is besides $-\lambda_{\min}(\mathbf{Q})$ in this term lies in $(0, 1)$. Thus, we will use this term as our bound on $\lambda_1$. $\qquad \square$

Now, we provide a similar upper bound result, though only partially, for eigenvalues of matrices that have the same structural properties as the Jacobian of our regularized system. Note that we have upper bounds only for eigenvalues that are complex (we have not used them anywhere in the main paper though).

**Theorem G.3.** *Let*

$$\mathbf{J} = \begin{bmatrix} -\mathbf{Q} & \mathbf{P} \\ -\mathbf{P}^T(\mathbf{I} - \eta\mathbf{Q}) & -2\eta\mathbf{P}^T\mathbf{P} \end{bmatrix},$$

*where $\mathbf{Q}$ is a real symmetric positive definite matrix and $\mathbf{P}$ is a full column rank matrix. Let $\eta < \frac{1}{\lambda_{\max}(\mathbf{Q})}$.*

*Then, if $\mathrm{Im}(\lambda) \neq 0$ for any eigenvalue $\lambda$ of $\mathbf{J}$,*

$$\text{Re}(\lambda) \leq -\frac{1}{2}\frac{1 - \eta\lambda_{\max}(\mathbf{Q})}{1 - \eta\lambda_{\min}(\mathbf{Q})}\left(\lambda_{\min}(\mathbf{Q}) + \eta\lambda_{\min}(\mathbf{P}^T\mathbf{P})\right)$$

*Proof.* Consider the following eigenvector equation:

$$\begin{bmatrix} -\mathbf{Q} & \mathbf{P} \\ -\mathbf{P}^T(\mathbf{I} - \eta\mathbf{Q}) & -2\eta\mathbf{P}^T\mathbf{P} \end{bmatrix}\begin{bmatrix} \mathbf{a}_1 + i\mathbf{a}_2 \\ \mathbf{b}_1 + i\mathbf{b}_2 \end{bmatrix} = (\lambda_1 + i\lambda_2)\begin{bmatrix} \mathbf{a}_1 + i\mathbf{a}_2 \\ \mathbf{b}_1 + i\mathbf{b}_2 \end{bmatrix},$$

where $u_i, v_i, \lambda_i$ are all real-valued. We want to show that $\lambda_1 < 0$. Let us first rewrite the above equation as follows:

$$\begin{bmatrix} -\mathbf{Q}\mathbf{a}_1 + \mathbf{P}\mathbf{b}_1 + i(-\mathbf{Q}\mathbf{a}_2 + \mathbf{P}\mathbf{b}_2) \\ -\mathbf{P}^T(\mathbf{I} - \eta\mathbf{Q})\mathbf{a}_1 - \eta\mathbf{P}^T\mathbf{P}\mathbf{b}_1 + i(-\mathbf{P}^T(\mathbf{I} - \eta\mathbf{Q})\mathbf{a}_2 - 2\eta\mathbf{P}^T\mathbf{P}\mathbf{b}_2) \end{bmatrix} = $$
$$\begin{bmatrix} \lambda_1\mathbf{a}_1 - \lambda_2\mathbf{a}_2 + i(\lambda_1\mathbf{a}_2 + \lambda_2\mathbf{a}_1) \\ \lambda_1\mathbf{b}_1 - \lambda_2\mathbf{b}_2 + i(\lambda_1\mathbf{b}_2 + \lambda_2\mathbf{b}_1) \end{bmatrix}$$

We can then equate the real and imaginary parts.

$$-\mathbf{Q}\mathbf{a}_1 + \mathbf{P}\mathbf{b}_1 = \lambda_1\mathbf{a}_1 - \lambda_2\mathbf{a}_2 \tag{15}$$
$$-\mathbf{P}^T(\mathbf{I} - \eta\mathbf{Q})\mathbf{a}_1 - 2\eta\mathbf{P}^T\mathbf{P}\mathbf{b}_1 = \lambda_1\mathbf{b}_1 - \lambda_2\mathbf{b}_2 \tag{16}$$
$$-\mathbf{Q}\mathbf{a}_2 + \mathbf{P}\mathbf{b}_2 = \lambda_1\mathbf{a}_2 + \lambda_2\mathbf{a}_1 \tag{17}$$
$$-\mathbf{P}^T(\mathbf{I} - \eta\mathbf{Q})\mathbf{a}_2 - 2\eta\mathbf{P}^T\mathbf{P}\mathbf{b}_2 = \lambda_1\mathbf{b}_2 + \lambda_2\mathbf{b}_1 \tag{18}$$
$$\tag{19}$$

We now multiply the above equations by $\mathbf{a}_1^T, \mathbf{b}_1^T, \mathbf{a}_2^T, \mathbf{b}_2^T$ respectively and add them:

$$\begin{aligned}\mathbf{a}_1^T(-\mathbf{Q}\mathbf{a}_1 + \mathbf{P}\mathbf{b}_1) - \mathbf{b}_1^T\mathbf{P}^T\mathbf{a}_1 - 2\eta\mathbf{b}_1\mathbf{P}^T\mathbf{P}\mathbf{b}_1 &= \mathbf{a}_1^T(\lambda_1\mathbf{a}_1 - \lambda_2\mathbf{a}_2) + \mathbf{b}_1^T(\lambda_1\mathbf{b}_1 - \lambda_2\mathbf{b}_2) \\ +\mathbf{a}_2^T(-\mathbf{Q}\mathbf{a}_2 + \mathbf{P}\mathbf{b}_2) - \mathbf{b}_2^T\mathbf{P}^T\mathbf{a}_2 - 2\eta\mathbf{b}_2\mathbf{P}^T\mathbf{P}\mathbf{b}_2 &\quad +\mathbf{a}_2^T(\lambda_1\mathbf{a}_2 + \lambda_2\mathbf{a}_1) + \mathbf{b}_2^T(\lambda_1\mathbf{b}_2 + \lambda_2\mathbf{b}_1) \\ +\eta\mathbf{b}_1^T\mathbf{P}^T\mathbf{Q}\mathbf{a}_1 + \eta\mathbf{b}_2^T\mathbf{P}^T\mathbf{Q}\mathbf{a}_2 & \end{aligned}$$

As a result, we get:

$$-\mathbf{a}_1^T\mathbf{Q}\mathbf{a}_1 - \mathbf{a}_2^T\mathbf{Q}\mathbf{a}_2 - 2\eta\mathbf{b}_1\mathbf{P}^T\mathbf{P}\mathbf{b}_1 - 2\eta\mathbf{b}_2\mathbf{P}^T\mathbf{P}\mathbf{b}_2 + \eta\mathbf{b}_1^T\mathbf{P}^T\mathbf{Q}\mathbf{a}_1 + \eta\mathbf{b}_2^T\mathbf{P}^T\mathbf{Q}\mathbf{a}_2 = \lambda_1$$

Above, we can substitute for $\mathbf{P}\mathbf{b}_1$ and $\mathbf{P}\mathbf{b}_2$ in $\eta\mathbf{b}_1^T\mathbf{P}^T\mathbf{Q}\mathbf{a}_1 + \eta\mathbf{b}_2^T\mathbf{P}^T\mathbf{Q}\mathbf{a}_2$ using the previous equations.

$$\begin{aligned} -\mathbf{a}_1^T\mathbf{Q}\mathbf{a}_1 - \mathbf{a}_2^T\mathbf{Q}\mathbf{a}_2 - 2\eta\mathbf{b}_1\mathbf{P}^T\mathbf{P}\mathbf{b}_1 - 2\eta\mathbf{b}_2\mathbf{P}^T\mathbf{P}\mathbf{b}_2 & \\ +\eta(\lambda_1\mathbf{a}_1^T\mathbf{Q}\mathbf{a}_1 - \lambda_2\mathbf{a}_1^T\mathbf{Q}\mathbf{a}_2 + \mathbf{a}_1^T\mathbf{Q}^T\mathbf{Q}\mathbf{a}_1) & \\ +\eta(\lambda_1\mathbf{a}_2^T\mathbf{Q}\mathbf{a}_2 + \lambda_2\mathbf{a}_2^T\mathbf{Q}\mathbf{a}_1 + \mathbf{a}_2^T\mathbf{Q}^T\mathbf{Q}\mathbf{a}_2) &= \lambda_1 \end{aligned}$$

$$\frac{-\mathbf{a}_1^T\mathbf{Q}\mathbf{a}_1 - \mathbf{a}_2^T\mathbf{Q}\mathbf{a}_2 - \eta\mathbf{b}_1\mathbf{P}^T\mathbf{P}\mathbf{b}_1 - \eta\mathbf{b}_2\mathbf{P}^T\mathbf{P}\mathbf{b}_2 + \eta\mathbf{a}_1^T\mathbf{Q}^T\mathbf{Q}\mathbf{a}_1 + \eta\mathbf{a}_2^T\mathbf{Q}^T\mathbf{Q}\mathbf{a}_2}{1 - \eta\left(\mathbf{a}_1^T\mathbf{Q}\mathbf{a}_1 + \mathbf{a}_2^T\mathbf{Q}\mathbf{a}_2\right)} = \lambda_1$$

We could do the above only because $\eta < \frac{1}{\lambda_{\max}(\mathbf{Q})}$ and therefore the denominator $1 - \eta(\mathbf{a}_1^T\mathbf{Q}\mathbf{a}_1 + \mathbf{a}_2^T\mathbf{Q}\mathbf{a}_2) \neq 0$.

In order to prove our upper bound, we first note the following inequality:

$$|\lambda_1| \geq \frac{(1 - \eta\lambda_{\max}(\mathbf{Q}))\lambda_{\min}(\mathbf{Q})(\|\mathbf{a}_1\|^2 + \|\mathbf{a}_2\|^2) + \eta\lambda_{\min}(\mathbf{P}^T\mathbf{P})(\|\mathbf{b}_1\| + \|\mathbf{b}_2\|^2)}{1 - \eta\lambda_{\min}(\mathbf{Q})(\|\mathbf{a}_1\|^2 + \|\mathbf{a}_2\|^2)} \tag{20}$$

We now multiply the first and third equations by $-\mathbf{a}_2^T$ and $\mathbf{a}_1^T$ respectively and sum them up, and second and fourth by $-\mathbf{b}_2^T$ and $\mathbf{b}_1^T$ respectively and sum them up. Then, we get:

$$-\mathbf{a}_2^T \mathbf{P} \mathbf{b}_1 + \mathbf{a}_1^T \mathbf{P} \mathbf{b}_2 = \lambda_2(\|\mathbf{a}_2\|^2 + \|\mathbf{a}_1^2\|)$$
$$\mathbf{b}_2^T \mathbf{P}^T (\mathbf{I} - \eta\mathbf{Q})\mathbf{a}_1 - \mathbf{b}_1^T \mathbf{P}^T (\mathbf{I} - \eta\mathbf{Q})\mathbf{a}_2 = \lambda_2(\|\mathbf{b}_2\|^2 + \|\mathbf{b}_1\|^2)$$

Using the above,

$$\begin{aligned}
\lambda_2(\|\mathbf{b}_2\|^2 + \|\mathbf{b}_1\|^2) - \lambda_2(\|\mathbf{a}_2\|^2 + \|\mathbf{a}_1\|^2) &= -\eta\mathbf{b}_2^T \mathbf{P}^T \mathbf{Q} \mathbf{a}_1 + \eta\mathbf{b}_1^T \mathbf{P}^T \mathbf{Q} \mathbf{a}_2 \\
&= -\eta\mathbf{a}_1^T \mathbf{Q}^T (\lambda_1\mathbf{a}_2 + \lambda_2\mathbf{a}_1 + \mathbf{Q}\mathbf{a}_2) + \\
&\quad \eta\mathbf{a}_2^T \mathbf{Q}^T (\lambda_1\mathbf{a}_1 - \lambda_2\mathbf{a}_2 + \mathbf{Q}\mathbf{a}_1) \\
&= -\eta\lambda_2\mathbf{a}_1^T \mathbf{Q}^T \mathbf{a}_1 - \eta\lambda_2\mathbf{a}_2^T \mathbf{Q}^T \mathbf{a}_2
\end{aligned}$$

Then, either $\lambda_2 = 0$ or when $\lambda_2 \neq 0$, we have $\|\mathbf{b}_2\|^2 + \|\mathbf{b}_1\|^2 = \|\mathbf{a}_2\|^2 + \|\mathbf{a}_1\|^2 - \eta\lambda_2\mathbf{a}_1^T \mathbf{Q}^T \mathbf{a}_1 - \eta\lambda_2\mathbf{a}_2^T \mathbf{Q}^T \mathbf{a}_2$. This translates to the inequality:

$$(1 - \eta\lambda_{\max}(\mathbf{Q}))(\|\mathbf{a}_2\|^2 + \|\mathbf{a}_1\|^2) \le \|\mathbf{b}_2\|^2 + \|\mathbf{b}_1\|^2 \le (1 - \eta\lambda_{\min}(\mathbf{Q}))(\|\mathbf{a}_2\|^2 + \|\mathbf{a}_1^2\|),$$

By adding $\|\mathbf{a}_2\|^2 + \|\mathbf{a}_1\|^2$ everywhere and using the fact that $\|\mathbf{a}_2\|^2 + \|\mathbf{a}_1\|^2 + \|\mathbf{b}_2\|^2 + \|\mathbf{b}_1\|^2 = 1$, the above inequality becomes:

$$\frac{1}{2 - \eta\lambda_{\min}(\mathbf{Q})} \le \|\mathbf{a}_2\|^2 + \|\mathbf{a}_1\|^2 \le \frac{1}{2 - \eta\lambda_{\max}(\mathbf{Q})}$$
$$\frac{1 - \eta\lambda_{\max}(\mathbf{Q})}{2 - \eta\lambda_{\max}(\mathbf{Q})} \le \|\mathbf{b}_2\|^2 + \|\mathbf{b}_1\|^2 \le \frac{1 - \eta\lambda_{\min}(\mathbf{Q})}{2 - \eta\lambda_{\min}(\mathbf{Q})}$$

The above inequalities yield an immediate lower bound on the magnitude in the latter case by plugging them in Equation 20:

$$\begin{aligned}
|\lambda_1| &\ge \frac{(1 - \eta\lambda_{\max}(\mathbf{Q}))\lambda_{\min}(\mathbf{Q})}{\frac{1}{(\|\mathbf{a}_1\|^2 + \|\mathbf{a}_2\|^2)} - \eta\lambda_{\min}(\mathbf{Q})} + \frac{\eta\lambda_{\min}(\mathbf{P}^T\mathbf{P})(\|\mathbf{b}_1\| + \|\mathbf{b}_2\|^2)}{1 - \eta\lambda_{\min}(\mathbf{Q})(\|\mathbf{a}_1\|^2 + \|\mathbf{a}_2\|^2)} \\
&\ge \frac{(1 - \eta\lambda_{\max}(\mathbf{Q}))\lambda_{\min}(\mathbf{Q})}{2(1 - \eta\lambda_{\min}(\mathbf{Q}))} + \frac{\eta\lambda_{\min}(\mathbf{P}^T\mathbf{P})(1 - \eta\lambda_{\max}(\mathbf{Q}))}{2(1 - \eta\lambda_{\min}(\mathbf{Q}))} \\
&\ge \frac{1}{2}\frac{1 - \eta\lambda_{\max}(\mathbf{Q})}{1 - \eta\lambda_{\min}(\mathbf{Q})}\left(\lambda_{\min}(\mathbf{Q}) + \eta\lambda_{\min}(\mathbf{P}^T\mathbf{P})\right)
\end{aligned}$$

Since, we know $\lambda_1$ is negative, this implies an upper bound on $\lambda_1$.

$\square$

## Footnotes

[5] We can consider only $\epsilon_G/2$ perturbations and not $\epsilon_G$ perturbations because for $\boldsymbol{\theta}_{\mathbf{G}}$ that is $\epsilon_G$ away from $\boldsymbol{\theta}_{\mathbf{G}}^{\star}$, perturbing it a little further might potentially change its support as a result of which $\nabla_{\boldsymbol{\theta}_{\mathbf{G}}} p_{\boldsymbol{\theta}_{\mathbf{G}}}(x)$ may not necessarily be zero for all $x \notin \text{supp}(p_{\boldsymbol{\theta}_{\mathbf{G}}^{\star}})$

[6]Thanks to Lars Mescheder for identifying that such a condition was missing in earlier versions of this paper.

[7]Thanks to Lars Mescheder for identifying this example.