[Reviews · NeurIPS 2017]

Reviewer 1



The authors present a dynamical system based analysis of simultaneous gradient descent updates for GANs, by considering the limit dynamical system that corresponds to the discrete updates. They show that under a series of assumptions, an equilibrium point of the dynamical system is locally asymptotically stable, implying convergence to the equilibrium if the system is initialized in a close neighborhood of it. Then they show how some types of GANs fail to satisfy some of their conditions and propose a fix to the gradient updates that re-instate local stability. They give experimental evidence that the local-stability inspired fix yields improvements in practice on MNIST digit generation and simple multi-modal distributions. However, I do think that these drawbacks are remedied by the fact that their modification, based on local asymptotic theory, did give noticeable improvements. Moreover, the local stability proof is a bit non-trivial from theoretical point of view. I think this paper provides some nice theoretical results in GAN training and gives interesting insights to GAN training through the dynamical system lens. The most promising part of the paper is the fact that their theory inspired fix to gradient descent yields experimental improvements. The theory part of the paper has some drawbacks: the assumptions that under equilibrium the discriminator outputs always zero is fairly strong and hard to believe in practice. In particular, it's hard to believe that the generator will be outputting a completely indistinguishable distribution from the real data, which is the only case where this could be an equilibrium for the discriminator. Also the fact that this is only a local stability without any arguments about the size of the region of attraction, might make the result irrelevant in practical training, where initializing in a tiny ball around the equilibrium would be impossible. Finally, it is not clear that local stability of the limit dynamical system will also imply local stability of the discrete system with stochastic gradients: if the variance in the stochastic gradient is large and the region of attraction is tiny, then a tiny step outside of the equilibrium together with a high variance gradient step could push the system outside of the region of attraction.

Reviewer 2



The authors show that the gradient descent method for training GANs is locally asymptotically stable in a more realistic setting than prior theoretical work. The proof idea is straightforward and accessible: the authors cast (stochastic) gradient descent as a stochastic dynamical system and appeal to standard results to conclude that the stability of gradient descent is equivalent to the stability of a deterministic dynamical system, for which a sufficient condition is having all negative eigenvalues in its Jacobian matrix. The rest of the paper states some technical assumptions, and uses them to compute bounds on the Jacobian of the deterministic dynamical system corresponding to the gradient descent method for training GANs. Using these bounds, the author conclude that ordinary GANs are locally stable while Wasserstein GANs are not. Finally, the authors apply their eigenvalue bounds to construct a new regularizer, which can make Wasserstein GANs stable. They present a few preliminary, but intuitively compelling experimental results with their regularizer. The paper is generally easy to read and the logic is clear. It clearly relates its result to that of recent related work. The implications of the theorems are well explained. One weakness of this paper seems to be an essential and restrictive dependence on realizability and the generator and the true distribution having the same support. While authors do allude to alternative results that relax these assumptions (at the cost of other restrictive ones), I think the implications and dependences on these assumptions should be discussed more prominently, and the results obtained using alternative assumptions better compared.

Reviewer 3



This paper presents a detailed theoretical analysis of the local asymptotic stability of GAN optimizations. The analysis provides insights into the nature of GAN optimizations. It highlights that the GAN objective can be a concave-concave objective with respect to both the discriminator and the generator parameters, especially at the regions close to the equilibrium. It later shows that the LQ WGAN system for learning a zero mean Gaussian distribution is not asymptotically stable at its equilibria. Motivated by this analysis, the authors propose a gradient-based regularization to stabilize both traditional GAN and the WGANs and improves convergence speed. It was not fully clear until much later (Section 4) about the effect these analyses have on the optimization convergence speed. Showing some more experiments with some quantification of why the optimization convergence is slow would be very useful (might be hard to show, though). Also, as the authors noted in Section 5, there are some limitations to the current analysis. Some empirical analysis (using different datasets and models) might shed light on the impact such limitations have on practical situations. Overall, this is a well written paper. The background description, however, can be improved to describe the main problem and issues with the existing work such that the reader does not have to review prior papers. Please fix the following typos in the final version: - of"foresight" -> of "foresight" - we use the analyze -> we use to analyze - guarantee locally stability -> guarantee local stability - is locally stable. provided f is -> is locally stable provided f is